# FINITE-TIME ANALYSIS OF ON-POLICY HETEROGENEOUS FEDERATED REINFORCEMENT LEARNING

**Chenyu Zhang**
Data Science Institute
Columbia University
New York, NY 10025, USA
`cz2736@columbia.edu`

**Han Wang**
Department of Electrical Engineering
Columbia University
New York, NY 10025, USA
`hw2786@columbia.edu`

**Aritra Mitra**
Department of Electrical and Computer Engineering
NC State University
Raleigh, NC 27695, USA
`amitra2@ncsu.edu`

**James Anderson**
Department of Electrical Engineering
Columbia University
New York, NY 10025, USA
`james.anderson@columbia.edu`

## ABSTRACT

Federated reinforcement learning (FRL) has emerged as a promising paradigm for reducing the sample complexity of reinforcement learning tasks by exploiting information from different agents. However, when each agent interacts with a potentially different environment, little to nothing is known theoretically about the non-asymptotic performance of FRL algorithms. The lack of such results can be attributed to various technical challenges and their intricate interplay: Markovian sampling, linear function approximation, multiple local updates to save communication, heterogeneity in the reward functions and transition kernels of the agents' MDPs, and continuous state-action spaces. Moreover, in the on-policy setting, the behavior policies vary with time, further complicating the analysis. In response, we introduce FedSARSA, a novel federated on-policy reinforcement learning scheme, equipped with linear function approximation, to address these challenges and provide a comprehensive finite-time error analysis. Notably, we establish that FedSARSA converges to a policy that is near-optimal for all agents, with the extent of near-optimality proportional to the level of heterogeneity. Furthermore, we prove that FedSARSA leverages agent collaboration to enable linear speedups as the number of agents increases, which holds for both fixed and adaptive step-size configurations.

## 1 INTRODUCTION

Federated reinforcement learning (FRL) (Qi et al., 2021; Nadiger et al., 2019; Zhuo et al., 2019), a distributed learning framework that unites the principles of reinforcement learning (RL) (Sutton & Barto, 2018) and federated learning (FL) (McMahan et al., 2017), is rapidly gaining prominence for its wide range of real-world applications, spanning areas such as edge computing (Wang et al., 2019), robot autonomous navigation (Liu et al., 2019), and Internet of Things (Lim et al., 2020). This paper poses an FRL problem, where multiple agents independently explore their own environments and collaborate to find a near-optimal universal policy accounting for their differing environmental models. FRL leverages the collaborative nature of FL to address the data efficiency and exploration challenges of RL. Specifically, we expect *linear speedups* in the convergence rate and increased overall exploration ability due to federated collaboration. We use FRL in autonomous driving (Liang et al., 2022) as a simple example to demonstrate our motivations and associated theoretical challenges. In this scenario, the objective is to determine a strategy (policy) that minimizes collision probability. In contrast to the single-agent setting, where a policy is found by letting one vehicle interact with its environment, the federating setting coordinates multiple vehicles to interact with their distinct environments—comprising different cities and traffic patterns. Despite their

aligned objectives, the environmental heterogeneity will produce distinct optimal strategies for each vehicle. Our goal is to find a universal robust strategy that performs well across all environments.

Tailored for such tasks, we propose a novel algorithm, FedSARSA, integrating SARSA, a classic on-policy temporal difference (TD) control algorithm (Rummery & Niranjan, 1994; Singh & Sutton, 1996), into a federated learning framework. On one hand, we want to leverage the power of federated collaboration to collect more comprehensive information and expedite the learning process. On the other hand, we want to utilize the robustness and adaptability of on-policy methods. To elaborate, within off-policy methods, such as Q-learning, agents select their actions according to a fixed *behavior* policy while seeking the optimal policy. In contrast, on-policy methods, such as SARSA, employ learned policies as behavior policies and constantly update them. By doing so, on-policy methods tend to learn safer policies, as they collect feedback through interaction following learned policies, and are more robust to environmental changes compared to off-policy methods (see Sutton & Barto (2018, Chapter 6)). Additionally, when equipped with different *policy improvement operators*, on-policy SARSA is more versatile and can learn a broader range of goals than off-policy Q-learning (see Section 4 and Appendix C). Formally analyzing our federated learning algorithm poses several multi-faceted challenges. We outline the most significant below.

- *Time-varying behavior policies.* In off-policy FRL with Markovian sampling (Woo et al., 2023; Khodadadian et al., 2022; Wang et al., 2023a), agents' observations are not i.i.d.; they are generated from a *time-homogeneous* ergodic Markov chain as agents follow a *fixed* behavior policy. Such an ergodic Markov chain converges rapidly to a steady-state distribution, enabling off-policy methods to inherit the theoretical guarantees for i.i.d. and mean-path cases (Bhandari et al., 2018; Wang et al., 2023a). In contrast, on-policy methods update agents' behavior policies dynamically, rendering their trajectories *nonstationary*. Therefore, previous analyses for off-policy methods, whether involving Markovian sampling or not, do not apply to our setting. Specifically, it remains unknown if the trajectories generated by on-policy FRL methods converge, and if they do, how this nonstationarity affects the convergence performance.

- *Environmental heterogeneity in on-policy planning.* In an FRL instance, it is impractical to assume that all agents share the same environment (Khodadadian et al., 2022; Woo et al., 2023). In a planning task, this heterogeneity results in agents having distinct optimal policies. Thus, to affirm the advantages of federated collaboration, it is crucial to precisely characterize the disparities in optimality. Only two FRL papers have considered heterogeneity: Jin et al. (2022) explored heterogeneity in transition dynamics without linear speedup, and Wang et al. (2023a) considered heterogeneity in a prediction task (policy evaluation). Beyond these studies, other research has addressed heterogeneity primarily within the domains of control design (Wang et al., 2023c) and system identification (Wang et al., 2023b). Unfortunately, neither the characterizations nor analyses of heterogeneity from the previous work apply to on-policy FRL. Specifically, heterogeneity in agents' optimal policies implies heterogeneity in the behavior policies, which could lead to drastically different local updates across agents, negating the benefits of collaboration.

- *Multiple local updates and client drift.* In the federated learning framework, agents communicate with a central server periodically to reduce communication cost, and conduct local updates between communication rounds. However, these local updates push agents to local solutions at the expense of the overall *federated* performance, a phenomenon known as *client drift* (Karimireddy et al., 2020). Uniquely within our setting, client drift and nonstationarity amplify each other.

- *Continuous state-action spaces and linear function approximation.* To better model real-world scenarios, we consider continuous state-action spaces and employ a linear approximation for the value function. Unfortunately, RL methods with linear function approximation (LFA) are known to exhibit less stable convergence when compared to tabular methods (Sutton & Barto, 2018; Gordon, 1996). Besides, the parameters associated with value function approximation no longer maintain an implicit magnitude bound. This concern is particularly relevant in on-policy FRL, where the client drift and the bias from nonstationarity both scale with the parameter magnitude.

Given these motivations and challenges, we ask

> *Can an agent expedite the process of learning its own near-optimal policy by leveraging information from other agents with potentially different environments?*

---

[1] Considered i.i.d. and Markovian sampling, but only established linear speedup result for the i.i.d. case.

Table 1: Comparison of finite-time analysis for value-based FRL methods. LSP and LFA represent linear speedup and linear function approximation under the Markovian sampling setting; Pred and Plan represent prediction (policy evaluation) and planning (policy optimization) tasks, respectively.

| Work | Hetero-geneity | LSP | LFA | Markovian Sampling | Task | Behavior Policy |
|---|---|---|---|---|---|---|
| Doan et al. (2019) | ✗ | ✗ | ✔ | ✗ | Pred | Fixed |
| Jin et al. (2022) | ✔ | ✗ | ✗ | ✗ | Plan | Fixed |
| Khodadadian et al. (2022) | ✗ | ✔ | ✔ | ✔ | Pred & Plan | Fixed |
| Shen et al. (2023) | ✗ | ✔[1] | ✔ | ✔ | Plan | Adaptive |
| Wang et al. (2023a) | ✔ | ✔ | ✔ | ✔ | Pred | Fixed |
| Woo et al. (2023) | ✗ | ✔ | ✗ | ✔ | Plan | Fixed |
| **Our work** | ✔ | ✔ | ✔ | ✔ | Pred & Plan | Adaptive |

We provide a complete non-asymptotic analysis of FedSARSA, resulting in the first positive answer to the above question. We situate our work with respect to prior work in Table 1. A summary of our contributions is provided below:

- *Heterogeneity in FRL optimal policies.* We formulate a practical FRL planning problem in which agents operate in heterogeneous environments, leading to heterogeneity in their optimal policies as agents pursue different goals. We provide an explicit bound on this heterogeneity in optimality, validating the benefits of collaboration (Theorem 1).

- *Federated SARSA and its finite-sample complexity.* We introduce the FedSARSA algorithm for the proposed FRL planning problem and establish a finite-time error bound achieving a state-of-the-art sample complexity (Theorem 2). At the time of writing, FedSARSA is the first provably sample-efficient on-policy algorithm for FRL problems.

- *Convergence region characterization and linear speedups via collaboration.* We demonstrate that when a constant step-size is used, federated learning enables FedSARSA to exponentially converge to a small region containing agents' optimal policies, whose radius tightens as the number of agents grows (Corollary 2.1). For a linearly decaying step-size, the learning process enjoys linear speedups through federated collaboration: the finite-time error reduces as the number of agents increases (Corollary 2.2). We validate these findings via numerical simulations.

## 2  RELATED WORK

**Federated reinforcement learning.**  A comprehensive review of FRL techniques and open problems was recently provided by Qi et al. (2021). FRL planning algorithms can be broadly categorized into two groups: policy- and value-based methods. In the first category, Jin et al. (2022); Xie & Song (2023) considered tabular methods but did not demonstrate any linear speedup. Fan et al. (2021) considered homogeneous environments and showed a *sublinear* speedup property. In the second category, Khodadadian et al. (2022); Woo et al. (2023) investigated federated Q-learning and demonstrated linear speedup under Markovian sampling. However, these studies did not examine the impact of environmental heterogeneity, a pivotal aspect in FRL. To bridge this gap, Wang et al. (2023a) presented a finite time analysis of federated TD(0) that can handle environmental heterogeneity. To take advantage of both policy- and value-based methods, Shen et al. (2023) analyzed distributed actor-critic algorithms, but only established the linear speedup result under i.i.d. sampling. Table 1 summarizes the key features of these value-based methods, including our work. There are also some works developed for studying the distributed version of RL algorithms: Doan et al. (2019) and Liu & Olshevsky (2023) provided a finite-time analysis of distributed variants of TD(0); however, their analysis is limited to the i.i.d sampling model.

**SARSA with linear function approximation.**  Single-agent SARSA is an on-policy TD control algorithm proposed by Rummery & Niranjan (1994) and Singh & Sutton (1996). To accommodate large or even continuous state-action spaces, Rummery & Niranjan (1994) proposed function

approximation. We refer to SARSA with and without LFA as linear SARSA and tabular SARSA respectively The asymptotic convergence result of tabular SARSA was first demonstrated by Singh et al. (2000). However, linear SARSA may suffer from chattering behavior within a region (Gordon, 1996; 2000; Bertsekas & Tsitsiklis, 1996). With a *smooth* policy improvement strategy, Perkins & Precup (2002) and Melo et al. (2008) established the asymptotic convergence guarantee for linear SARSA. Recently, the finite-time analysis for linear SARSA was provided by Zou et al. (2019).

## 3 PRELIMINARIES

### 3.1 FEDERATED LEARNING

Federated Learning (FL) is a distributed machine learning framework designed to train models using data from multiple clients while preserving privacy, reducing communication costs, and accommodating data heterogeneity. We adopt the server-client model with periodic aggregation, akin to well-known algorithms like FedAvg (McMahan et al., 2017) and FedProx (Sahu et al., 2018). Agents (clients) perform multiple *local updates* (iterations of a learning algorithm) between communication rounds with the central server. During a communication round, agents synchronize their local parameters with those aggregated by the server. However, this procedure introduces *client-drift* issues (Karimireddy et al., 2020; Charles & Konečný, 2021), which can hinder the efficacy of federated training. This problem is particularly pronounced in our on-policy FRL setting, where client drift is amplified due to the interplay with other factors.

### 3.2 MARKOV DECISION PROCESS AND ENVIRONMENTAL HETEROGENEITY

We consider $N$ agents that explore within the same state-action space but with potentially different environment models. Specifically, agent $i$'s environment model is characterized by a Markov decision process (MDP) denoted by $\mathcal{M}^{(i)} = \left(\mathcal{S}, \mathcal{A}, r^{(i)}, P^{(i)}, \gamma\right)$. Here, $\mathcal{S}$ denotes the state space, $\mathcal{A}$ is the action space, $r^{(i)} : \mathcal{S} \times \mathcal{A} \to [0, R]$ is a bounded reward function, $\gamma \in (0, 1)$ is the discount factor, and $P^{(i)}$ is the Markov transition kernel such that $P_a^{(i)}(s, s')$ is the probability of agent $i$'s transition from state $s$ to $s'$ following action $a$. While all agents share the same state-action space, their reward functions and state transition kernels can differ. Agents select actions based on their *policies*. A policy $\pi$ maps a state to a distribution over actions, $\pi(a|s)$ denotes the probability of an agent taking action $a$ at state $s$.

**Assumption 1** (Uniform ergodicity). For each $i \in [N]$, the Markov chain induced by any policy $\pi$ and state transition kernel $P^{(i)}$ is ergodic with a uniform mixing rate. In other words, for any MDP $\mathcal{M}^{(i)}$ and candidate policy $\pi$, there exists a steady-state distribution $\eta_\pi^{(i)}$, as well as constants $m_i \geq 1$ and $\rho_i \in (0, 1)$, such that

$$\sup_{s \in \mathcal{S}} \sup_\pi \left\| P_\pi \left( S_t^{(i)} = \cdot \,\middle|\, S_0^{(i)} = s \right) - \eta_\pi^{(i)} \right\|_{\mathrm{TV}} \leq m_i \rho_i^t,$$

where $\| \cdot \|_{\mathrm{TV}}$ is the total variation distance.[2]

Assumption 1 is a standard assumption in the RL literature needed to provide finite-time bounds under Markovian sampling (Bhandari et al., 2018; Zou et al., 2019; Srikant & Ying, 2019).

Agents operate in their own environments and may have their own goals. We collectively refer to the differences in the transition kernels and rewards as environmental heterogeneity. Intuitively, collaboration among agents is advantageous when the heterogeneity is small, but can become counterproductive when the heterogeneity is large. We now provide two natural definitions for measuring environmental heterogeneity.

**Definition 1** (Transition kernel heterogeneity). We capture the transition kernel heterogeneity using the total variation induced norm:

$$\epsilon_p := \max_{i,j \in [N]} \left\| P^{(i)} - P^{(j)} \right\|_{\mathrm{TV}},$$

---

[2]We use the functional-analytic definition of the total variation, which is twice the quantity $\sup_{A \in \mathcal{F}} |p(A)|$ for any signed measure $p$ on $\mathcal{F}$.

where with a slight abuse of notation, we define

$$\|P\|_{\mathrm{TV}} := \sup_{\substack{q \in \mathcal{P}(\mathcal{S} \times \mathcal{A}) \\ \|q\|_{\mathrm{TV}} = 1}} \|qP\|_{\mathrm{TV}} = \sup_{\substack{q \in \mathcal{P}(\mathcal{S} \times \mathcal{A}) \\ \|q\|_{\mathrm{TV}} = 1}} \left\| \int_{\mathcal{S} \times \mathcal{A}} q(s,a) P_a(s, \cdot) \mathrm{d}s \mathrm{d}a \right\|_{\mathrm{TV}},$$

where $\mathcal{P}(\mathcal{S} \times \mathcal{A})$ is the set of probability measures on $\mathcal{S} \times \mathcal{A}$. By the triangle inequality and the uniform bound on rewards, $R$, we have $\epsilon_p \leq 2$.

**Definition 2** (Reward heterogeneity). We capture the reward heterogeneity using the infinity norm:

$$\epsilon_r := \max_{i,j \in [N]} \frac{\left\| r^{(i)} - r^{(j)} \right\|_\infty}{R},$$

where $\|r\|_\infty = \sup_{s,a \in \mathcal{S} \times \mathcal{A}} |r(s,a)|$. By the triangle inequality, we have $\epsilon_r \leq 2$.

### 3.3 Value Function and SARSA

An RL planning task aims to maximize the expected *return*, defined as the accumulated reward of a trajectory. For a given policy $\pi$, the expected return of a state-action pair $(s, a)$ is captured by the Q-value function:

$$q_\pi(s,a) = \mathbb{E}_\pi \left[ \sum_{t=0}^\infty \gamma^t r(s_t, a_t) \Big| S_0 = s, A_0 = a \right] = \underbrace{r(s,a) + \gamma \mathbb{E}_\pi \left[ q_\pi(S_1, A_1) | S_0 = s, A_0 = a \right]}_{T_\pi q_\pi(s,a)}, \tag{1}$$

where the expectation is taken with respect to a transition kernel that follows the policy $\pi$ (except for the initial action, which is fixed to $a$). For any MDP, there exists an optimal policy $\pi_*$ such that $q_{\pi_*}(s,a) \geq q_\pi(s,a)$ for any other policy $\pi$ and state-action pair $(s,a)$. *This paper focuses on an FRL problem where all agents aim to find a universal policy that is near-optimal for all MDPs under a low-heterogeneity regime.*

To find such an optimal policy for a single agent, SARSA updates the estimated Q-value function based on (1) by sampling and bootstrapping. With the updated estimation of the value function, SARSA improves the policy via a policy improvement operator. By alternating policy evaluation and policy improvement, SARSA finds the optimal policy within the policy space. The tabular SARSA for a single agent can be described by the following update rules:

$$\begin{cases} Q(s_t, a_t) & \leftarrow Q(s_t, a_t) + \alpha \left( r(s_t, a_t) + \gamma Q(s_{t+1}, a_{t+1}) - Q(s_t, a_t) \right), \\ \pi(a_{t+1}|s_{t+1}) & \leftarrow \Gamma(Q(s_{t+1}, a_{t+1})), \end{cases} \tag{2}$$

where $Q$ is the estimated Q-value function, $\alpha$ is the learning step-size, and $\Gamma$ is the policy improvement operator. We provide further discussion on the policy improvement operator in Section 4.

### 3.4 Linear Function Approximation and Nonlinear Projected Bellman Equation

When the state-action space is large or continuous, tabular methods are intractable. Therefore, we employ a linear approximation for the Q-value function (Rummery & Niranjan, 1994). For a given feature extractor $\phi : \mathcal{S} \times \mathcal{A} \to \mathbb{R}^d$, we approximate the Q-value function as $Q_\theta(s,a) = \phi(s,a)^T \theta$, where $\theta \in \Theta \subseteq \mathbb{R}^d$ is a parameter vector to be learned. Without loss of generality, we assume that $\|\phi(s,a)\|_2 \leq 1$ for every state-action pair $(s,a)$. Linear function approximation translate the task of finding the optimal policy to that of identifying the optimal parameter $\theta$ that solves the nonlinear projected Bellman equation:

$$Q_\theta = \Pi_\pi T_\pi Q_\theta, \tag{3}$$

where $T_\pi$ is the Bellman operator defined by the right-hand side of (1), and $\Pi_\pi$ is the orthogonal projection onto the linear subspace spanned by the range of the $\phi$ using the inner product $\langle x, y \rangle_\pi = \mathbb{E}_{S \sim \eta_\pi, A \sim \pi(S)} [x(S, A)^T y(S, A)]$. Equation (3) reduces to the linear Bellman equation used in policy evaluation when the policy $\pi$ is fixed (Tsitsiklis & Van Roy, 1996; Bhandari et al., 2018), and to the Bellman optimality equation used in Q-learning when the policy improvement operator is the greedy selector (Watkins & Dayan, 1992; Melo et al., 2008).

## 4 ALGORITHM

We now develop FedSARSA; a federated version of linear SARSA. In FedSARSA, each agent explores its own environment and improves its policy using its observations, which we refer to as local updating. Periodically, agents send the parameter progress to the central server, where the parameters get aggregated and sent back to each agent. We present FedSARSA in Algorithm 1.

**Local update.** Locally, agent $i$ updates its parameter using the SARSA update rule. With linear function approximation, the Q-value function update in (2) becomes

$$\theta_{t+1}^{(i)} = \theta_t^{(i)} + \alpha_t g_t^{(i)} \left( \theta_t^{(i)}; s_t^{(i)}, a_t^{(i)} \right),$$

where $\alpha_t$ is the step-size[3] and $g_t^{(i)}$ is defined as

$$g_t^{(i)} (\theta; s, a) = \phi(s, a) r^{(i)}(s, a) + \phi(s, a) \left( \gamma \phi(s', a') - \phi(s, a) \right)^T \theta, \quad s' \sim P_a^{(i)}(s, \cdot),\ a' \sim \pi_{\theta_t^{(i)}}(\cdot|s'). \tag{4}$$

We refer to $g_t$ as a *semi-gradient* as it resembles a stochastic gradient but does not represent the true gradient of any static loss function (Barnard, 1993). Also, we introduce a subscript $t$ to the semi-gradient to indicate that it depends on the policy $\pi_{\theta_t^{(i)}}$ at time step $t$.

**Policy improvement.** We assume that all agents use the same policy improvement operator $\Gamma$, which returns a policy $\pi$ for any Q-value function. Since we consider linearly approximated Q-value functions, we can view the policy improvement operator as acting on the parameter space: $\Gamma : \theta \mapsto \pi$. We denote the policy resulting from the parameter $\theta$ as $\pi_\theta = \Gamma(\theta)$. To ensure the convergence of the algorithm, we need the following assumption on the policy improvement operator's smoothness.

**Assumption 2** (Lipschitz continuous policy improvement operator)**.** The policy improvement operator is Lipschitz continuous in TV distance with constant $L$:

$$\|\pi_{\theta_1}(\cdot|s) - \pi_{\theta_2}(\cdot|s)\|_{\text{TV}} \le L\|\theta_1 - \theta_2\|_2, \quad \forall \theta_1, \theta_2 \in \Theta, s \in \mathcal{S}.$$

Furthermore, $L \le w/(H\sigma)$, where $H$, $\sigma$, and $w$ are problem constants to be defined in Appendix.

When the action space is of finite measure, Assumption 2 is equivalent to that in Zou et al. (2019). This assumption is standard for linear SARSA (Zou et al., 2019; Perkins & Precup, 2002; Melo et al., 2008). As shown in (De Farias & Van Roy, 2000; Perkins & Pendrith, 2002; Zhang et al., 2022), linear SARSA with noncontinuous policy improvement may diverge.

An example policy improvement operator satisfying Assumption 2 is the softmax function with suitable temperature parameter (Gao & Pavel, 2017). In contrast, the deterministic greedy policy improvement employed in Q-learning is an illustrative case where Assumption 2 does not hold. Additionally, when the policy improvement operator maps to a fixed point $\pi$, SARSA reduces to TD learning, which evaluates the policy $\pi$. Generally, SARSA searches the *optimal* policy within the policy space $\Gamma(\Theta)$ determined by the policy improvement operator and the parameter space.

**Server side aggregation.** FedSARSA adds an additional aggregation step to parallelize linear SARSA. During this step, agents communicate with a central server by sending their parameters or parameter progress over a given period. The central server then aggregates these local parameters and returns the updated parameters to the agents. Intuitively, if the agents' MDPs are similar, i.e., the level of heterogeneity is low, then exchanging information via the server should benefit each agent. This is precisely the rationale behind the server-aggregation step. In general, $K$ is selected to strike a balance between the communication cost and the accuracy in FL.

Besides averaging, we add a projection step to ensure stability of the parameter sequence. This technique is commonly used in the literature on stochastic approximation and RL (Zou et al., 2019; Bhandari et al., 2018; Qiu et al., 2021; Wang et al., 2023a). In practice, it is anticipated that an *implicit* bound on the parameters exists without requiring explicit projection.

---

[3]For ease of presentation, we assume all agents share the same step-size. Our analysis handles agents using their own step-size schedule, as long as each agent's step-size falls within the specified range.

---

**Algorithm 1:** FedSARSA

---

**input** *Initial parameter* $\theta_0^{(i)} = \bar{\theta}_0$

**for** $t = 0, \ldots, T-1$ **do**

  **for** *each agent* $i = 1, \ldots, N$ **do in parallel**

    $\pi_t^{(i)} = \Gamma(\theta_t^{(i)})$                  `// policy improvement`

    Sample observation $(s_t^{(i)}, a_t^{(i)}, r_t^{(i)}, s_{t+1}^{(i)}, a_{t+1}^{(i)})$ following policy $\pi_t^{(i)}$

    $\theta_{t+1}^{(i)} = \theta_t^{(i)} + \alpha_t g_t^{(i)}$, where $g_t^{(i)}$ is defined in (4)     `// local update`

  **if** $t + 1 \equiv 0 \pmod{K}$ **then**           `// every K iterations`

    $\bar{\theta}_{t+1} = \Pi_{\bar{G}}\left(\frac{1}{N}\sum_{i=1}^{N}\theta_{t+1}^{(i)}\right)$        `// federated aggregation`

    Set $\theta_{t+1}^{(i)} = \bar{\theta}_{t+1}$ for each agent $i \in [N]$

---

## 5    ANALYSIS

We begin our analysis of FedSARSA by establishing a perturbation bound on the solution to (3), which captures the near-optimality of the solution under reward and transition heterogeneity. We then provide a finite-time error bound of FedSARSA, which enjoys the linear speedup achieved by the federated collaboration. Building on this, we discuss the parameter selection of our algorithm.

### 5.1    NEAR OPTIMALITY UNDER HETEROGENEITY

We consider an FRL task where all agents collaborate to find a universal policy. However, due to environmental heterogeneity, each agent has a potentially different optimal policy. Therefore, it is essential to determine the convergence region of our algorithm, and how it relates to the optimal parameters of the agents. To show that we find a near-optimal parameter for all agents, we need to characterize the difference between the optimal parameters of agents. Given the operator $\Gamma$, we denote by $\theta_*^{(i)}$ the unique solution to (3) for MDP $\mathcal{M}^{(i)}$. The next theorem bounds the distance between agents' optimal parameters as a function of reward- and transition kernel heterogeneity.

**Theorem 1** (Perturbation bounds on SARSA fixed points). *There exist positive problem dependent constants $w$, $H$, and $\sigma$ such that*

$$\max_{i,j\in[N]}\left\{\left\|\theta_*^{(i)} - \theta_*^{(j)}\right\|_2\right\} \leq \frac{R\epsilon_r + H\sigma\epsilon_p}{w} =: \frac{\Lambda(\epsilon_p, \epsilon_r)}{w},$$

*where $\epsilon_p$ and $\epsilon_r$ are the perturbation bounds on environmental models defined in Definitions 1 and 2.*

We explicitly define the constants in Theorem 1 and show that $w = O(1-\gamma)$ in Appendix I. In the next subsection, we demonstrate that there exists a parameter $\theta_*$ such that $\|\theta_*^{(i)} - \theta_*\| \leq \Lambda(\epsilon_p, \epsilon_r)/w$, and Algorithm 1 converges to a neighborhood of $\theta_*$ whose radius is also of $O(\Lambda(\epsilon_p, \epsilon_r)/(1-\gamma))$. Since $\Lambda(\epsilon_p, \epsilon_r) = O(\epsilon_p + \epsilon_r)$, when the environmental heterogeneity is small, these results guarantee that $\theta_*$ is near-optimal for all agents.

Theorem 1 is the first perturbation bound on nonlinear projected Bellman fixed points. Wang et al. (2023a) established similar perturbation bounds for linear projected Bellman fixed points using the perturbation theory of linear equations. However, it is crucial to note that their approach does not extend to our setting where (3) is nonlinear.

### 5.2    FINITE-TIME ERROR AND LINEAR SPEEDUP

We now provide the main theorem of the paper, which bounds the mean squared error of Algorithm 1 recursively, and directly gives several finite-time error bounds.

**Theorem 2** (One-step progress). *Let $\{\theta_t^{(i)}\}$ be the parameters returned by Algorithm 1 and $\bar{\theta}_t = \frac{1}{N}\sum_{i=1}^{N}\theta_t^{(i)}$. Then, there exist positive problem dependent constants $w, C_1, C_2, C_3, C_4$, and a parameter $\theta_*$ such that $\max_{i\in[N]}\|\theta_*^{(i)} - \theta_*\| \leq \Lambda(\epsilon_p, \epsilon_r)/w$, and for any $t \in \mathbb{N}$, it holds that*

$$\mathbb{E}\left\|\bar{\theta}_{t+1} - \theta_*\right\|^2 \leq (1 - \alpha_t w)\mathbb{E}\left\|\bar{\theta}_t - \theta_*\right\|^2 + \alpha_t C_1 \Lambda^2(\epsilon_p, \epsilon_r) + \alpha_t^2 C_2/N + \alpha_t^3 C_3 + \alpha_t^4 C_4. \quad (5)$$

*Explicit definitions of the constants are provided in Appendix J.*

On the right-hand side of (5), the first term is a contractive term that inherits its contractivity from the projected Bellman operator; the second term accounts for heterogeneity; the third term captures the effect of noise where the variance gets scaled down by a factor of $N$ (linear speedup) due to collaboration among agents; the last two terms represent higher-order terms, which are negligible, compared to other terms. In the following two corollaries, we study the effects of using constant and decaying step-sizes in the above bound.

**Corollary 2.1** (Finite-time error bound for constant step-size)**.** *With a constant step-size $\alpha_t \equiv \alpha_0 \leq w/(2120(2K + 8 + \ln(m/(\rho w))))$, for any $T \in \mathbb{N}$, we have*

$$\mathbb{E}\left\|\bar{\theta}_T - \theta_*^{(i)}\right\|^2 \leq 4e^{-\alpha_0 w T}\left\|\theta_0 - \theta_*^{(i)}\right\|^2 + \frac{1}{w}\left(\left(C_1 + \frac{6}{w}\right)\Lambda^2(\epsilon_p, \epsilon_r) + \alpha_0\frac{C_2}{N} + \alpha_0^2 C_3 + \alpha_0^3 C_4\right).$$

**Corollary 2.2** (Finite-time error bound for decaying step-size)**.** *With a linearly decaying step-size $\alpha_t = 4/(w(1 + t + a))$, where $a > 0$ is to guarantee that $\alpha_0 \leq \min\{1/(8K), w/64\}$, there exists a convex combination $\widetilde{\theta}_T$ of $\{\bar{\theta}_t\}_{t=0}^T$ such that*

$$\mathbb{E}\left\|\widetilde{\theta}_T - \theta_*^{(i)}\right\|^2 = \frac{H^2}{(1-\gamma)^2}\cdot O\left(\frac{K^2 + \tau^5}{(1-\gamma)^2 T^2} + \frac{\tau}{NT} + \frac{\Lambda^2(\epsilon_p, \epsilon_r)}{H^2}\right) = \frac{H^2}{(1-\gamma)^2}\cdot\widetilde{O}\left(\frac{1}{NT} + \frac{\Lambda^2(\epsilon_p, \epsilon_r)}{H^2}\right).$$

We now discuss the implications of the above theoretical guarantees.

**Convergence region.** From Corollary 2.1, with a constant step-size $\alpha$, FedSARSA exponentially converges to a ball around the optimal parameter $\theta_i^*$ of each agent. The radius of this ball is governed by two objects: (i) the level of environmental heterogeneity; (ii) the inherent noise in our model. In the absence of heterogeneity, the above guarantee is precisely what one obtains for stochastic approximation algorithms with a constant step-size (Zou et al., 2019; Srikant & Ying, 2019; Bhandari et al., 2018). The presence of heterogeneity manifests itself in the $O(\Lambda(\epsilon_p, \epsilon_r)/(1-\gamma)) = O(\epsilon_p + \epsilon_r)$ term in the convergence region radius. Since the optimal parameters of the agents may not be identical (under heterogeneity), such a term is generally unavoidable.

**Linear speedup.** Turning our attention to Corollary 2.2 (where we use a decaying step-size), let us first consider the homogeneous case where $\epsilon_p = \epsilon_r = 0$. When $T \geq N$, the $O(1/(NT))$ rate we obtain in this case is the best one can hope for statistically: with $T$ data samples per agent and $N$ agents, one can reduce the variance of our noise model by at most $NT$. Thus, for a homogeneous setting, our rate is optimal, and clearly demonstrates an $N$-fold linear speedup over the single-agent sample-complexity of $O(1/T)$ in Zou et al. (2019). In this context, *our work provides the first such bound for a federated on-policy RL algorithm*, and complements results of a similar flavor for the off-policy setting in Khodadadian et al. (2022). When the agents' MDPs differ, via collaboration, each agent is still able to converge at the *expedited* rate of $O(1/NT)$ to a ball of radius $O(\epsilon_p + \epsilon_r)$ around the optimal parameter of each agent. The implication of this result is simple: by participating in federation, each agent can *quickly* (i.e., with an $N$-fold speedup) find an $O(\epsilon_p + \epsilon_r)$-approximate solution of its optimal parameter; using such an approximate solution as an initial condition, the agent can then fine-tune (personalize) - based on its own data - to converge to its own optimal parameter *exactly* (in mean-squared sense). *This is the first result of its kind for federated planning, and complements the plethora of analogous results in federated optimization* (Sahu et al., 2018; Khaled et al., 2019; Li et al., 2019; Koloskova et al., 2020; Woodworth et al., 2020; Pathak & Wainwright, 2020; Wang et al., 2020; Mitra et al., 2021; Mishchenko et al., 2022). Arriving at the above result, however, poses significant challenges relative to prior art. We now provide insights into these challenges and our strategies to overcome them.

## 5.3 PROOF SKETCH: ERROR DECOMPOSITION

Our main approach of proving Theorem 2 is to leverage the contraction property of the Bellman equation (3) to identify a primary "descent direction." Algorithm 1 then updates the parameters along this direction with multi-sourced stochastic bias. We provide an informal mean squared error decomposition (formalized in Appendix I.1 ) to illustrate this idea:

$$\mathbb{E}\left\|\bar{\theta}_{t+1} - \theta_*\right\|^2 \leq \text{recursion} + \text{descent direction} + \text{gradient heterogeneity} + \text{client drift}$$
$$+ \text{gradient progress} + \text{mixing} + \text{backtracking} + \text{gradient variance}.$$

Some of these terms commonly appears in an FRL analysis: the descent direction is given by the contraction property of the Bellman equation (3) when the policy improvement operator is sufficiently smooth (Appendix I.2); the client drift represents the deviation of agents' local parameters from the central parameter, which is controlled by the step-size and synchronization period (Appendix I.4); the mixing property (Assumption 1) allows a stationary trajectory to rapidly reach to a steady distribution (Appendix I.6). We highlight some unique terms in our analysis.

*Gradient heterogeneity.* This term accounts for the local update heterogeneity, which scales with the environmental heterogeneity. The effect of time-varying policies coupled with multiple local updates accentuates the effect of such heterogeneity. Thus, particular care is needed to ensure that the bias introduced by heterogeneity does not compound over iterations (Appendix I.3).

*Backtracking.* FedSARSA possesses nonstationary transition kernels. To deal with this challenge and use the mixing property of stationary MDPs, we *virtually backtrack* a period $\tau$: starting at time step $t-\tau$, we fix the policy $\Gamma(\theta_{t-\tau}^{(i)})$ for agent $i$, and consider a subsequent virtual trajectory following this fixed policy. The divergence between the updates computed on real and virtual observations is controlled by the step-size $\alpha_t$ and backtracking period $\tau$ (Appendix I.7).

*Gradient progress.* Note that the steady distribution in the mixing term corresponds to an *old* policy. Since the backtracking period is small, the discrepancy (progress) between this old policy and the current one is small (Appendix I.5).

*Gradient variance.* While one can directly use the projection radius to bound the semi-gradient variance, such an approach would fall short of establishing the desired linear speedup effect. To achieve the latter, we need a more refined argument that shows how one can obtain a "variance-reduction" effect by combining data generated from non-identical time-varying Markov chains (Appendix I.8).

## 6 SIMULATIONS

We create a finite state space of size $|\mathcal{S}| = 100$, an action space of $|\mathcal{A}| = 100$, a feature space of dimension $d = 25$, and set $\gamma = 0.2$ and $R = 10$. The actions determine the transition matrices by shifting the columns of a reference matrix. The synchronization period is set to $K = 10$, and the step-size of $\alpha_0 = 0.01$. For the full experiment setup, please refer to Appendix C. In Figure 1, we plot the mean squared error averaged over ten runs for different heterogeneity levels and numbers of agents. The simulation results are consistent with Corollary 2.1 and demonstrate the robustness of our method towards environmental heterogeneity. Additional simulations, including federated TD(0) and on-policy federated Q-learning covered by our algorithm, can be found in Appendix C.

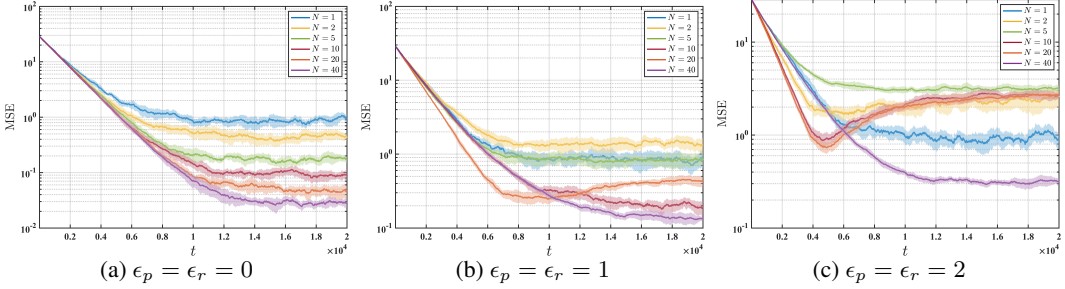

(a) $\epsilon_p = \epsilon_r = 0$        (b) $\epsilon_p = \epsilon_r = 1$        (c) $\epsilon_p = \epsilon_r = 2$

Figure 1: Performance of FedSARSA under Markovian sampling.

## 7 CONCLUSION

We proposed a straightforward yet powerful on-policy federated reinforcement learning method: FedSARSA. Our finite-time analysis of FedSARSA provides the first theoretically conformation of the statement: an agent can expedite the process of learning its own near-optimal policy by leveraging information from other agents with potentially different environments.

ACKNOWLEDGMENTS

JA is partially supported by Columbia Data Science Institute and NSF grants ECCS 2144634 & 2231350.

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

# Appendix

## Table of Contents

## A    ORGANIZATION OF APPENDIX

The appendix is organized as follows. First, we present an additional comparison of our results with other finite-time results in Appendix B, and additional simulation results in Appendix C. In Appendices D and E, we introduce the concept of central MDP and some notation that will assist our analysis. In Appendix F, to aid readability, we list all the constants that appear in the paper for readers' convenience. In Appendix G, we provide several preliminary lemmas that will be used throughout the analysis. Before presenting lemmas for Theorem 2, we first prove Theorem 1 in Appendix H, for it will be used by later lemmas. In Appendix I, we first decompose the mean squared error and then present seven lemmas, each bounding one term in the decomposition. Then, we provide the proof of Theorem 2 and Corollaries 2.1 and 2.2 in Appendix J and Appendix K, respectively. To provide insights into our results, we discuss the dependencies of constants in Appendix L. Finally, we reduce FedSARSA to the tabular case in Appendix M, demonstrating the flexibility and efficiency of our algorithm.

## B    FINITE-TIME RESULTS COMPARISON

A comparison of finite-time results on temporal difference methods is provided in Table 2.

Table 2: Comparison of finite-time results. Results with green background are first provided by our work; results with blue background are covered by our work. "Linear" indicates the usage of linear function approximation, and "Hetero" indicates the presence of environmental heterogeneity. All constants are defined in Section 5 and Appendix I. We show the squared $\ell_2$ error for linear settings and squared $\ell_\infty$ error for tabular settings. Asymptotic notations are omitted for simplicity.

| | Federated | | | | | | Single-Agent | |
| --- | --- | --- | --- | --- | --- | --- | --- | --- |
| | Linear | | Tabular | | | | Linear | Tabular |
| | Hetero | Homog | Hetero | | Homog | | | |
| TD Learning | $\frac{H^2}{(1-\gamma)^2NT}+\frac{\Lambda^2}{(1-\gamma)}$ [†] | $\frac{H^2}{(1-\gamma)^2NT}$ [‡] | $\frac{SA}{\lambda^2(1-\gamma)^4NT}+\frac{\Lambda^2}{\lambda^2(1-\gamma)^2}$ [**] | | $\frac{S^2}{\lambda^5(1-\gamma)^9NT}$ [‡] | | $\frac{H^2}{(1-\gamma)^2T}$ [¶] | $\frac{SA}{\lambda^2(1-\gamma)^4T}$ [**] |
| Q-Learning | $-$ | | $\frac{H^2}{(1-\gamma)^2NT}$ [‡] | $\frac{1}{(1-\gamma)^6T^2}+\frac{\Lambda^2}{(1-\gamma)^4}$ [§] | $\frac{S^2}{\lambda^5(1-\gamma)^9NT}$ [‡] | | $\frac{H^2}{(1-\gamma)^2T}$ [¶] | $\frac{SA}{\lambda(1-\gamma)^5T}$ [‖] |
| SARSA | $\frac{H^2}{(1-\gamma)^2NT}+\frac{\Lambda^2}{(1-\gamma)^2}$ [*] | $\frac{H^2}{(1-\gamma)^2NT}$ [*] | $\frac{SA}{\lambda^2(1-\gamma)^4NT}+\frac{\Lambda^2}{\lambda^2(1-\gamma)^2}$ [**] | | $\frac{SA}{\lambda^2(1-\gamma)^4NT}$ [**] | | $\frac{H^2}{(1-\gamma)^2T}$ [#] | $\frac{SA}{\lambda^2(1-\gamma)^4T}$ [**] |

[†] (Wang et al., 2023a)    [‡] (Khodadadian et al., 2022)    [§] (Jin et al., 2022)    [¶] (Bhandari et al., 2018)
[‖] (Qu & Wierman, 2020)    [#] (Zou et al., 2019)    [*] Corollary 2.2    [**] Appendix M

## C    ADDITIONAL SIMULATIONS

### C.1    ADDITIONAL SIMULATIONS FOR FEDSARSA

We first restate the simulation setup in more detail. We index a finite state space by $\mathcal{S} = [100]$ and an action space by $\mathcal{A} = [100]$, where the actions determine the transition matrices by shifting the columns of a reference matrix $P_0$:

$$P_a = \texttt{circ\_shift}(P_0, \texttt{columns} = a),$$

where `circ_shift` denotes a circular shift operator. We construct the feature extractor as

$$\phi(s, a) = e_{(s \bmod d_1) \cdot d_2 + a \bmod d_2} \in \mathbb{R}^{d_1 \times d_2},$$

where $e_i$ is the indicator vector with the $i$-th entry being 1 and the rest being 0. We set $d_1 = 5$ and $d_2 = 5$. For the policy improvement operator, we employ the softmax function with a temperature of 100:

$$\pi_\theta(a|s) = \frac{\exp(\theta^T\phi(s,a)/100)}{\sum_{a'\in\mathcal{A}}\exp(\theta^T\phi(s,a')/100)}.$$

Other parameters are set as follows: the reward cap $R = 10$, the discount factor $\gamma = 0.2$, the synchronization period $K = 10$, and the step-size $\alpha_0 = 0.01$.

To construct heterogeneous MDPs, we first generate a nominal MDP $\mathcal{M}_1$ and obtain the remaining MDPs by adding the perturbations to $\mathcal{M}_1$. Unlike in FedTD (Wang et al., 2023a), where the optimal parameters can be obtained by solving the linear projected Bellman equation directly, here we get a *reference* parameter $\theta_{\mathrm{ref}}^{(1)}$ by running a single-agent linear SARSA on $\mathcal{M}_1$ with decaying step-size. As suggested in Corollary 2.2, the reference parameter converges to the optimal parameter corresponding to $\mathcal{M}_1$. Then, we calculate the mean squared error with respect to the reference parameter: $\left\|\bar{\theta}_t - \theta_{\mathrm{ref}}^{(1)}\right\|_2^2$. All of our simulations are averaged over ten runs and all graphs are plotted with 95% confidence region.

In Figure 1, both kernel heterogeneity and reward heterogeneity are set at the same level. In Figure 2, we fix the kernel heterogeneity as $1.0$ and vary the reward heterogeneity. In contrast, we fix the reward heterogeneity as zero and vary the kernel heterogeneity in Figure 3. Again, these results affirm the robustness of our method towards environmental heterogeneity. Furthermore, they seemingly suggest that the algorithm is more sensitive to reward heterogeneity than kernel heterogeneity. However, it is important to note that $\epsilon_p$ and $\epsilon_r$ represent upper bounds and may be much larger than the actual heterogeneity level.

Further exploring the effect of heterogeneity on federated collaboration, Figures 4 and 5 illustrate the effect of different reward and kernel heterogeneity levels on the performance of FedSARSA respectively. Generally, higher levels of heterogeneity result in larger mean squared error, which aligns with our theoretical results in Section 5.

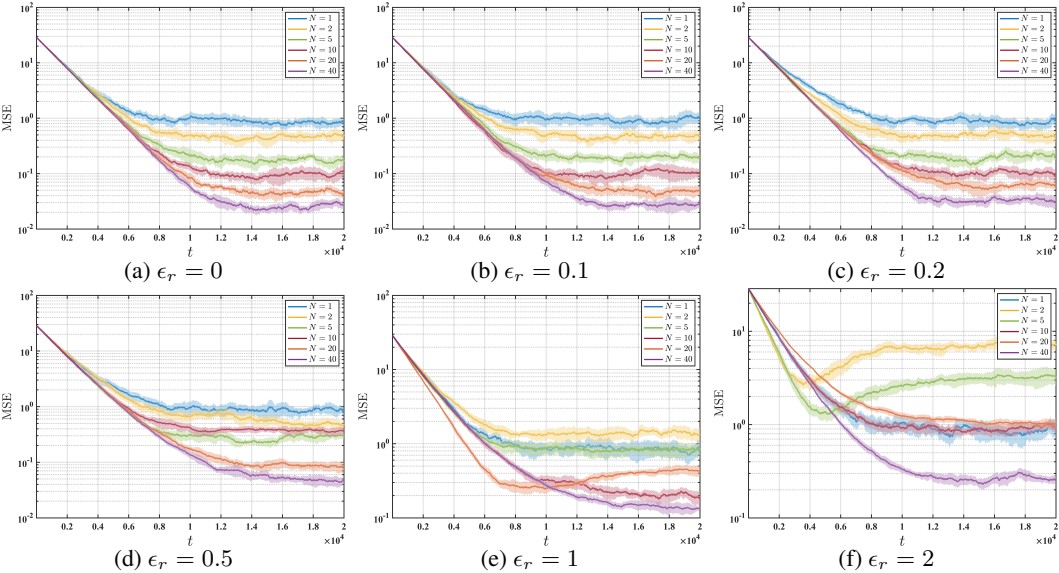

Figure 2: Performance of FedSARSA under Markovian sampling for varying reward heterogeneity and numbers of agents with fixed kernel heterogeneity ($\epsilon_p = 1$).

## C.2 SIMULATIONS FOR FEDERATED TD(0)

As discussed in Section 4, FedSARSA reduces to federated TD(0) (Wang et al., 2023a) when the policy improvement operator maps any parameter to a fixed policy $\pi$. This corresponds to a fixed transition kernel. Therefore, we conduct simulations for federated TD(0) to demonstrate the adaptability of FedSARSA. We inherit the simulation setup from the previous subsection (Appendix C.1), which matches the setup in Wang et al. (2023a). We fix the behavior policy by fix the transition matrix as the reference matrix $P_0$. The results are presented in Figure 6, which are similar to the results in Section 6, again validating our theoretical results.

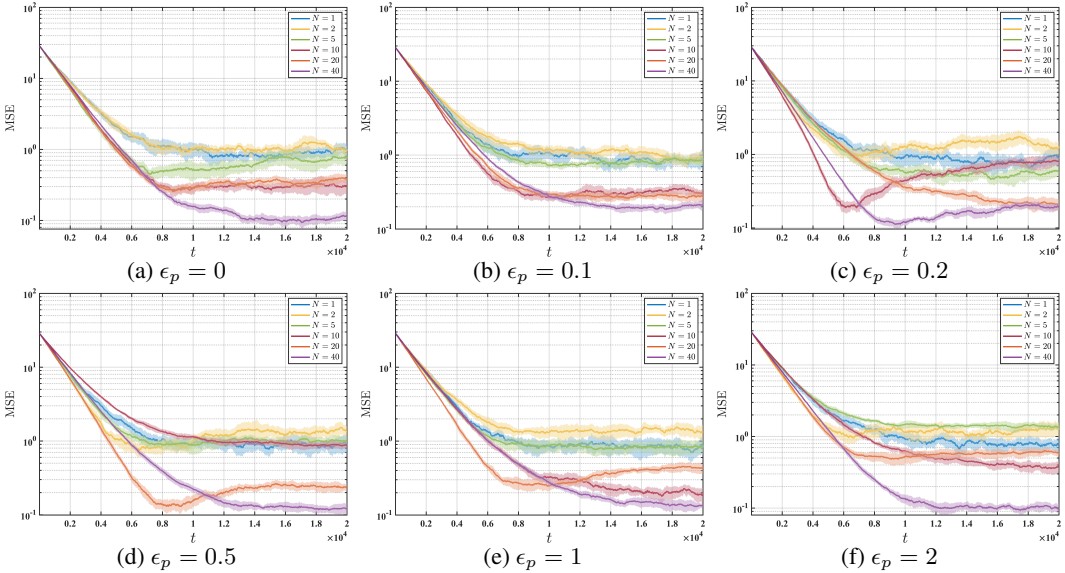

Figure 3: Performance of FedSARSA under Markovian sampling for varying kernel heterogeneity and numbers of agents with fixed reward heterogeneity ($\epsilon_r = 1$).

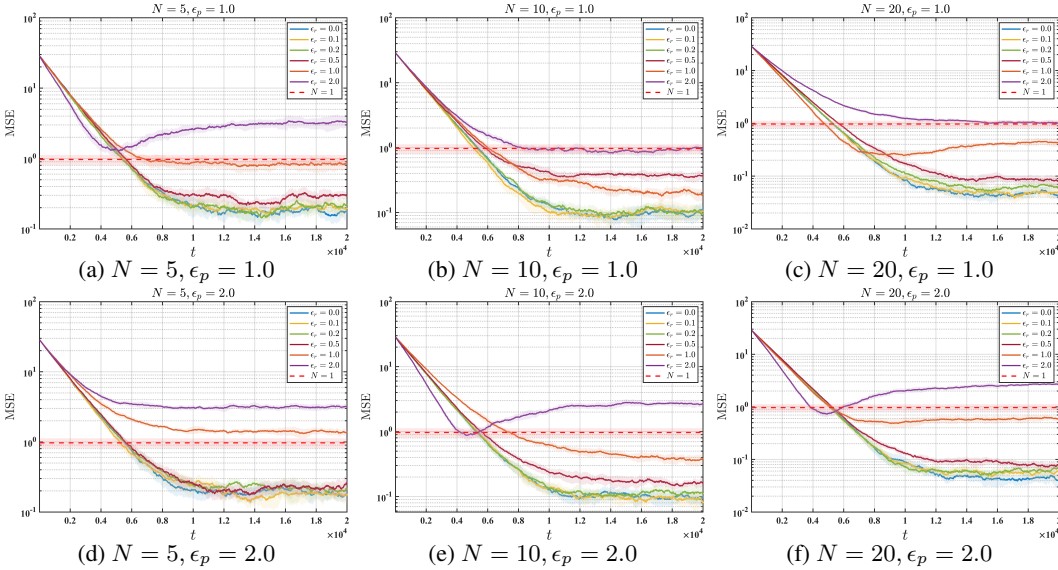

Figure 4: Effect of the reward heterogeneity on the performance of FedSARSA.

## C.3 SIMULATIONS FOR ON-POLICY FEDERATED Q-LEARNING

When equipped with a greedy policy improvement operator, FedSARSA reduces to on-policy federated Q-Learning. Specifically, we employ the greedy policy improvement operator:

$$\pi_\theta(a|s) = \mathbb{1}\{a = \operatorname*{argmax}_{a' \in \mathcal{A}} \theta^T \phi(s, a')\},$$

where $\mathbb{1}$ is the indicator function. For the other part of the simulation setup, we inherit the setup from the previous subsection (Appendix C.1). The results are presented in Figure 7, which resemble the results in Section 6 and Appendix C.2.

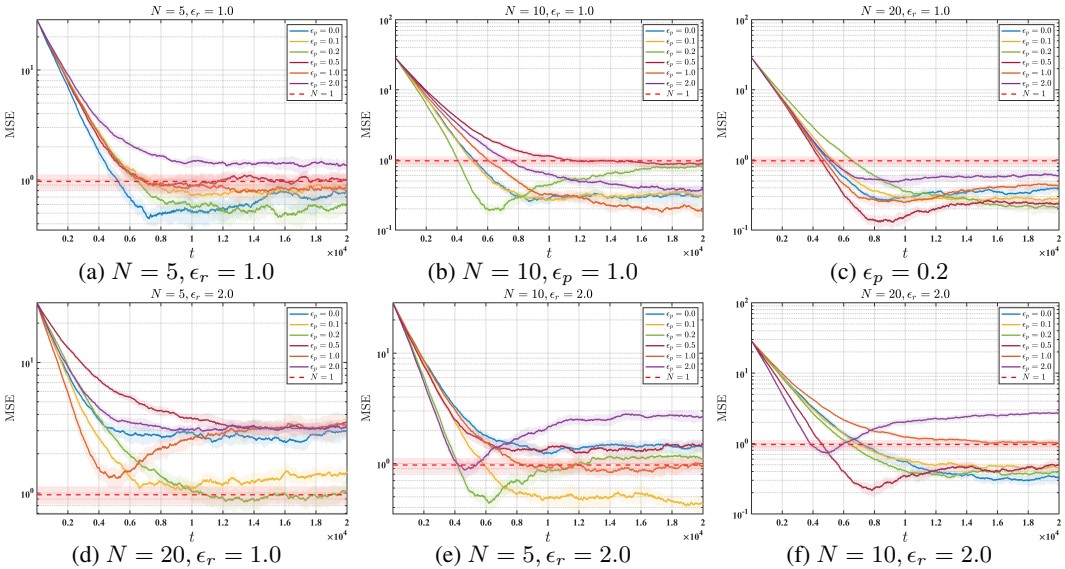

Figure 5: Effect of the kernel heterogeneity on the performance of FedSARSA.

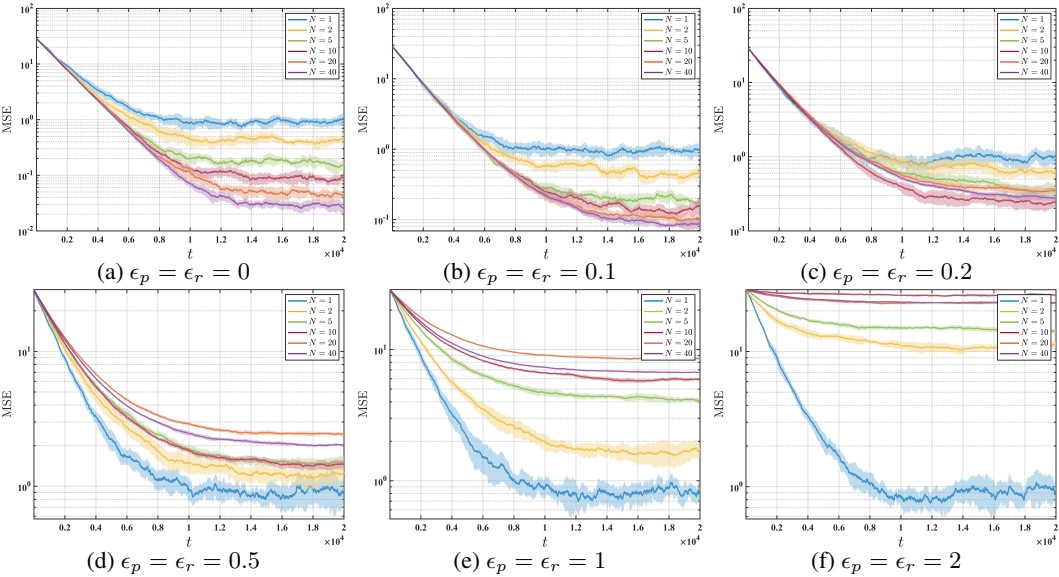

Figure 6: Performance of FedSARSA with a fixed-point policy improvement operator, covering federated TD(0).

## D  CENTRAL MDP

To facilitate our analysis, we introduce a virtual MDP: $\bar{\mathcal{M}} := \frac{1}{N}\sum_{i=1}^{N}\mathcal{M}^{(i)}$. Specifically, $\bar{\mathcal{M}} = (\mathcal{S}, \mathcal{A}, \bar{r}, \bar{P}, \gamma)$, where $\bar{r} = \frac{1}{N}\sum_{i=1}^{N}r^{(i)}$, $\bar{P} = \frac{1}{N}\sum_{i=1}^{N}P^{(i)}$. We refer to this virtual MDP as the *central MDP*. The following proposition shows that $\bar{\mathcal{M}}$ is indistinguishable from the collection of actual MDPs and also satisfies Assumption 1.

**Proposition 1.** *If MDPs* $\{\mathcal{M}^{(i)}\}$ *are ergodic (aperiodic and irreducible) under a fixed policy* $\pi$, *the central MDP* $\bar{\mathcal{M}}$ *is also ergodic under* $\pi$.

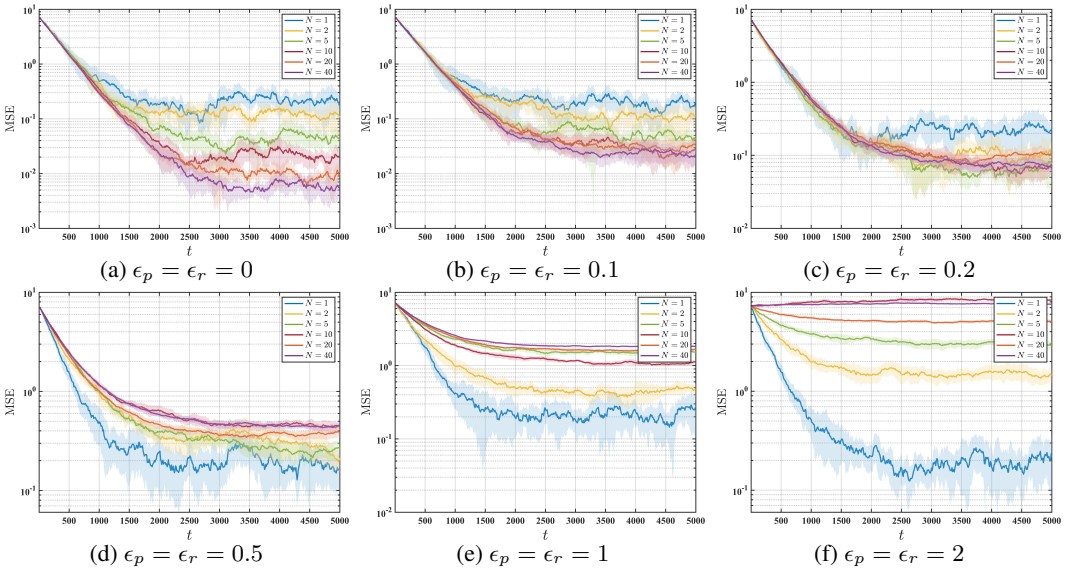

Figure 7: Performance of FedSARSA with a greedy policy improvement operator, covering on-policy federated Q-learning.

*Proof.* Suppose $\pi$ is given. We first show that $\mathcal{M}$ is also aperiodic. If not, by the definition of aperiodicity (Meyn & Tweedie, 2012, Page 121), there exists $s \in \mathcal{S}$ such that

$$\bar{d}(s) := \gcd\{n : \bar{P}^n(s,s) > 0\} > 1,$$

where $\gcd$ returns the greatest common divisor and we omit the subscript $\pi$ of $\bar{P}_\pi$ since we consider a fixed policy. The above inequality indicates that, for any $n \in \mathbb{N} \setminus \{k\bar{d}(s)\}_{k\in\mathbb{N}}$, it holds that

$$0 = \bar{P}^n(s,s) = \left(\frac{1}{N}\sum_{i=1}^{N} P^{(i)}\right)^n (s,s) \geq \frac{1}{N^n}\left(P^{(i)}\right)^n (s,s), \quad \forall i \in [N]. \tag{6}$$

Now since $\mathcal{M}^{(i)}$ is aperiodic, $d^{(i)}(s) := \gcd\{n : (P^{(i)})^n(s,s) > 0\} = 1$. Thus there exists $n \in \mathbb{N} \setminus \{k\bar{d}(s)\}_{k\in\mathbb{N}}$ such that $(P^{(i)})^n(s,s) > 0$ (otherwise $d^{(i)}(s) \geq \bar{d}(s)$), which contradicts to (6). Therefore, we conclude that $\bar{d}(s) = 1$ for any $s \in \mathcal{S}$, and thus $\mathcal{M}$ is aperiodic. We now show that $\bar{\mathcal{M}}$ is irreducible given $\{\mathcal{M}^{(i)}\}$ are irreducible (Meyn & Tweedie, 2012, Page 93). For any $A \subset \mathcal{B}(\mathcal{S})$ with positive measure, where $\mathcal{B}(\mathcal{S})$ is the Borel $\sigma$-field on $\mathcal{S}$, we have $\min\{n : (P^{(i)})^n(s,A) > 0\} < +\infty$ for any $s \in \mathcal{S}$ and $i \in [N]$. Then again by (6), we get

$$\min\{n : \bar{P}^n(s,A) > 0\} \leq \min_{i\in[N]} \min\left\{n : \frac{1}{N^n}\left(P^{(i)}\right)^n (s,A) > 0\right\} < +\infty.$$

Therefore, $\bar{\mathcal{M}}$ is irreducible. $\square$

When the state-action space is finite, Proposition 1 covers Wang et al. (2023a, Proposition 1).

By Proposition 1, we can confidently regard $\bar{\mathcal{M}}$ as the MDP of a virtual agent, which does not exhibit any distinctive properties in comparison to the actual agents. Therefore, we denote $\mathcal{M}^{(0)} := \bar{\mathcal{M}}$ and define the extended number set $[\bar{N}] := [N] \cup \{0\} = \{0, 1, \ldots, N\}$. When we drop the superscript $(i)$, it should be clear from the context if we are talking about the central MDP $\bar{\mathcal{M}}$ or an arbitrary MDP $\mathcal{M}^{(i)}$. Clearly, the extended MDP set $\{\mathcal{M}^{(i)}\}_{i\in[\bar{N}]}$ still satisfies Assumption 1 and Definitions 1 and 2, and thus Theorem 1. Now we can specify the special parameter $\theta_*$ in Theorem 2: it is the unique solution to (3) for $\bar{\mathcal{M}}$. In other words, Theorem 2 asserts that the algorithm converges to the central optimal parameter.

# E    NOTATION

Before presenting lemmas and proofs of our main theorems, we introduce some notation that will aid in our analysis. We introduce a notation for the unprojected central parameter:

$$\breve{\theta}_{t+1} = \frac{1}{N} \sum_{i=1}^{N} \left( \theta_t^{(i)} + \alpha_t g_t^{(i)} \right), \quad \text{when } t+1 \equiv 0 \pmod{K}.$$

Then, $\theta_{t+1}^{(i)} = \bar{\theta}_{t+1} = \Pi_{\bar{G}}(\breve{\theta}_{t+1})$ when $t+1 \equiv 0 \pmod{K}$. It's easy to verify that for any $\|\theta\| \le \bar{G}$, we have

$$\|\bar{\theta}_t - \theta\| \le \|\breve{\theta}_t - \theta\|. \tag{7}$$

Then we define some notations on the MDPs. Note that all these definitions apply to the extended MDP set $\{\mathcal{M}^{(i)}\}_{i \in [\bar{N}]}$ that includes the central MDP.

**Definition 3** (Steady distributions). Assumption 1 guarantees the existence of a steady state distribution for any MDP and policy $\pi$. We denote $\eta_\theta^{(i)}$ as the steady state distribution with respect to MDP $\mathcal{M}^{(i)}$ and policy $\pi_\theta$, i.e.,

$$\eta_\theta^{(i)}(s) := \lim_{t \to \infty} P_{\pi_\theta}^{(i)}(S_t = s | S_0 = s_0).$$

Additionally, given a policy $\pi_\theta$, the steady state-action distribution is defined as

$$\mu_\theta^{(i)}(s, a) := \eta_\theta^{(i)}(s) \cdot \pi_\theta(a|s).$$

Then, the two-step steady distribution is defined as

$$\varphi_\theta^{(i)}(s, a, s', a') := \mu_\theta^{(i)}(s, a) P_a^{(i)}(s, s') \pi_\theta(a'|s').$$

For a local parameter $\theta_t^{(i)}$, we simplify the above notations as follows:

$$\eta_t^{(i)} := \eta_{\theta_t^{(i)}}^{(i)}, \quad \mu_t^{(i)} := \mu_{\theta_t^{(i)}}^{(i)}, \quad \varphi_t^{(i)} := \varphi_{\theta_t^{(i)}}^{(i)}.$$

We are now ready to provide the precise definitions of the semi-gradients discussed in Section 5.

**Definition 4** (Semi-gradients). As indicated by (4), a semi-gradient is a function of both the parameter $\theta$ and the observation tuple $O = (s, a, s', a')$, while the observation tuple is dependent on the local Markovian trajectory. Therefore, the general form of a semi-gradient is

$$g_{t-\tau}^{(i)} \left( \theta; O_t^{(i)} \right) := \phi \left( s_t^{(i)}, a_t^{(i)} \right) \left( r^{(i)} \left( s_t^{(i)}, a_t^{(i)} \right) + \gamma \phi^T \left( s_{t+1}^{(i)}, a_{t+1}^{(i)} \right) \theta - \phi^T \left( s_t^{(i)}, a_t^{(i)} \right) \theta \right),$$

where $O_t^{(i)} = \left( s_t^{(i)}, a_t^{(i)}, s_{t+1}^{(i)}, a_{t+1}^{(i)} \right)$ is the observation of agent $i$ at time step $t$, and the subscript $t-\tau$ indicates that the trajectory after time step $t-\tau$ follows a fixed policy $\pi_{\theta_{t-\tau}^{(i)}}$, i.e.,

$$a_{t-\tau}^{(i)}, a_{t-\tau+1}^{(i)}, \ldots, a_{t+1}^{(i)} \sim \pi_{\theta_{t-\tau}^{(i)}}.$$

When $\tau = 0$, the semi-gradient corresponds to an actual SARSA trajectory, and we omit the subscript and the observation argument, i.e.,

$$g^{(i)}(\theta) := g_t^{(i)} \left( \theta; O_t^{(i)} \right).$$

When $\tau > 0$, the semi-gradient corresponds a virtual trajectory, and we use $\tilde{O}_t$ in place of $O_t$ to indicate it is a virtual observation at the current time step.

We add a bar to denote the mean-path semi-gradients, i.e.,

$$\bar{g}_{t-\tau}^{(i)}(\theta) := \mathbb{E}_{\varphi_{t-\tau}^{(i)}} \left[ g_{t-\tau}^{(i)}(\theta, O) \right] = \mathbb{E}_{\varphi_{t-\tau}^{(i)}} \left[ \phi(s, a)(r^{(i)}(s, a) + \gamma \phi^T(s', a')\theta - \phi^T(s, a)\theta) \right],$$

where the expectation is taken over the two-step observation steady distribution:

$$O = (s, a, s', a') \sim \varphi_{t-\tau}^{(i)} := \varphi_{\theta_{t-\tau}^{(i)}}^{(i)}.$$

For mean-path semi-gradients, the randomness of the observation is eliminated, and the parameter $\theta$ is the only argument, and we can substitute the subscript $t-\tau$ with a general parameter $\theta'$; then we define

$$\bar{g}_{\theta'}^{(i)}(\theta) := \mathbb{E}_{\varphi_{\theta'}^{(i)}} \left[ \phi(s, a)(r^{(i)}(s, a) + \gamma \phi^T(s', a')\theta - \phi^T(s, a)\theta) \right].$$

We omit the superscript $(i)$ when referring to the central MDP $\bar{\mathcal{M}}$. For instance,

$$\bar{g}_{t-\tau}(\theta) := \mathbb{E}_{\varphi_{t-\tau}} \left[ \phi(s, a)(\bar{r}(s, a) + \gamma \phi^T(s', a')\theta - \phi^T(s, a)\theta) \right].$$

Finally, for notational simplicity, we use bold symbols to denote the average semi-gradients, e.g.,

$$\boldsymbol{g}_{t-\tau}(\boldsymbol{\theta}_t) = \frac{1}{N} \sum_{i=1}^{N} g_{t-\tau}^{(i)} \left( \theta_t^{(i)} \right).$$

The above notations will be used in combination, e.g.,

$$\bar{\boldsymbol{g}}(\boldsymbol{\theta}_t) = \frac{1}{N} \sum_{i=1}^{N} \bar{g}^{(i)} \left( \theta_t^{(i)} \right) = \frac{1}{N} \sum_{i=1}^{N} \bar{g}_t^{(i)} \left( \theta_t^{(i)} \right).$$

We can further decompose semi-gradients into TD operators.

**Definition 5** (TD operators). A semi-gradient $g_{t-\tau}^{(i)} \left( \theta, O_t^{(i)} \right)$ can be decomposed into the following two two operators:

$$g_{t-\tau}^{(i)} \left( \theta, O_t^{(i)} \right) = A_{t-\tau}^{(i)} \left( O_t^{(i)} \right) \theta + b_{t-\tau}^{(i)} \left( O_t^{(i)} \right).$$

where

$$\begin{cases} A_{t-\tau}^{(i)} \left( O_t^{(i)} \right) & = \phi \left( s_t^{(i)}, a_t^{(i)} \right) \left( \gamma \phi^T \left( s_{t+1}^{(i)}, a_{t+1}^{(i)} \right) - \phi^T \left( s_t^{(i)}, a_t^{(i)} \right) \right), \\ b_{t-\tau}^{(i)} \left( O_t^{(i)} \right) & = \phi \left( s_t^{(i)}, a_t^{(i)} \right) r^{(i)} \left( s_t^{(i)}, a_t^{(i)} \right), \end{cases} \qquad a_t^{(i)}, a_{t+1}^{(i)} \sim \pi_{\theta_{t-\tau}^{(i)}}.$$

Similar to Definition 4, we can define other TD operators for each semi-gradient, e.g., the mean-path TD operators:

$$\begin{cases} \bar{A}_\theta^{(i)} & = \mathbb{E}_{\varphi_\theta^{(i)}}[A^{(i)}(O)], \\ \bar{b}_\theta^{(i)} & = \mathbb{E}_{\mu_\theta^{(i)}}[b^{(i)}(O)]. \end{cases}$$

We summarize the notations defined in this section and other notations used in our analysis in Table 3.

# F CONSTANTS

We first introduce two important constants that serve as base constants throughout the paper. The first one is the upper bound of the norm of the central parameter, denoted by $G \geq \|\bar{\theta}\|$. For this bound to hold, we require the projection radius $\bar{G}$ to be large enough such that $\left\| \theta_*^{(i)} \right\| \leq \bar{G}$ for $i \in [\bar{N}]$. The explicit expression for $G$ will be given in Corollary I.5.3. Then, we define

$$H = R + (1 + \gamma)G. \tag{8}$$

The constant $H$ can be viewed as the scale of the problem, analogous to $|\mathcal{S}||\mathcal{A}|$ for the tabular setting that will be discussed in Appendix M. For local parameters, we define a similar function $h(\theta) := R + (1 + \gamma)\|\theta\|$.

We summarized the constants that appear in our analysis in Table 4. Notice that $\tau, \alpha_0$, and $\alpha_t$ in Table 4 refer to the constants in the case with a linearly decaying step-size. For the case with a constant size, these constants are fixed and specified in Corollary 2.1.

Table 3: Notation

| Notation | Definition |
|----------|------------|
| $[N], [\bar{N}]$ | The set of $N$ numbers and the set of $N + 1$ numbers including $0$ |
| $\mathcal{M}^{(i)}, \bar{\mathcal{M}}$ | Markov decision processes |
| $\mathcal{S}, \mathcal{A}, \Theta$ | State space, action space, and parameter space |
| $r^{(i)}, \bar{r}, P^{(i)}, \bar{P}$ | Reward functions and transition kernels |
| $S_t^{(i)}, U_t^{(i)}, O_t^{(i)}$ | Agent $i$'s state, action, and observation random variable at time step $t$ |
| $s, a, o$ | Instances of the state, action, and observation |
| $\pi, \Gamma$ | A policy and the policy improvement operator |
| $\|\cdot\|_{\mathrm{TV}}$ | Total variation distance and its induced norm for transition kernels |
| $q, Q$ | True Q-value function and estimated Q-value function |
| $\phi, \theta$ | Feature map and feature weight (parameter) |
| $\Pi_\pi, \Pi_{\bar{G}}$ | Orthogonal projection operator |
| $T_\pi$ | Bellman operator |
| $\pi_*^{(i)}, \theta_*^{(i)}$ | Optimal policies and optimal parameters |
| $\eta_\theta^{(i)}, \mu_\theta^{(i)}, \varphi_\theta^{(i)}$ | Steady distributions |
| $g$ | Semi-gradient |
| $A, b, Z$ | Temporal difference operators |
| $h$ | $h(\theta) := R + (1 + \gamma)\|\theta\|$ |
| $\Omega_t, \omega_t$ | Client drift |
| $\mathcal{F}_t$ | Filtration containing all randomness prior to time step $t$ |

## G  PRELIMINARY LEMMAS

In this section, we present two preliminary lemmas that will be used throughout the analysis.

**Lemma G.1** (Steady distribution differences). *For the same MDP, the TV distance between the steady distributions with regard to two different policies is bounded as follows:*

$$\|\eta_{\theta_1} - \eta_{\theta_2}\|_{\mathrm{TV}} \le L\sigma' \|\theta_1 - \theta_2\|_2,$$
$$\|\mu_{\theta_1} - \mu_{\theta_2}\|_{\mathrm{TV}} \le L(1 + \sigma') \|\theta_1 - \theta_2\|_2,$$
$$\|\varphi_{\theta_1} - \varphi_{\theta_2}\|_{\mathrm{TV}} \le L(2 + \sigma) \|\theta_1 - \theta_2\|_2,$$

*where $L$ is the Lipschitz constant of the policy improvement operator specified in Assumption 2 and $\sigma'$ is a constant determined by $m$ and $\rho$ specified in Assumption 1. Letting $\sigma := \sigma' + 2$, all three TV distances above are bounded by $L\sigma\|\theta_1 - \theta_2\|_2$. Next, for a fixed parameter $\theta$, the TV distance between the steady distributions with regard to two MDPs is bounded as follows:*

$$\left\|\eta_\theta^{(i)} - \eta_\theta^{(j)}\right\|_{\mathrm{TV}} \le \sigma'\epsilon_p,$$
$$\left\|\mu_\theta^{(i)} - \mu_\theta^{(j)}\right\|_{\mathrm{TV}} \le \sigma'\epsilon_p,$$
$$\left\|\varphi_\theta^{(i)} - \varphi_\theta^{(j)}\right\|_{\mathrm{TV}} \le (\sigma' + 1)\epsilon_p.$$

*By the above inequalities, for different MDPs and different parameters, we have*

$$\left\|\mu_{\theta^{(i)}}^{(i)} - \mu_{\theta^{(j)}}^{(j)}\right\|_{\mathrm{TV}} \le \sigma'\epsilon_p + L\sigma\|\theta^{(i)} - \theta^{(j)}\|_2.$$

*Proof.* For the same MDP, by (Mitrophanov, 2005, Corollary 3.1), we get

$$\|\eta_{\theta_1} - \eta_{\theta_2}\|_{\mathrm{TV}} \le \sigma'\|P_{\theta_1} - P_{\theta_2}\|_{\mathrm{TV}},$$

Table 4: Constants

| Notation | Meaning | Reference | Range or Order |
|---|---|---|---|
| $N$ | Number of agents | Section 3.2 | $\mathbb{N}$ |
| $R$ | Reward cap | Section 3.2 | $(0, +\infty)$ |
| $S, A$ | Measures of the state space and action space | Section 3.2 | $(0, +\infty]$ |
| $\gamma$ | Discount factor | Section 3.2 | $(0, 1)$ |
| $m_i, m$ | Markov chain mixing constant | Assumption 1 and Lemma G.1 | $[1, +\infty)$ |
| $\rho_i, \rho$ | Markov chain mixing rate | Assumption 1 and Lemma G.1 | $(0, 1)$ |
| $\sigma, \sigma'$ | Steady distribution perturbation constant | Lemma G.1 | $O(\log m/(1 - \rho))$ |
| $\bar{G}$ | Algorithm projection radius | Algorithm 1 | $(0, +\infty)$ |
| $G$ | Parameter norm upper bound | Corollary I.5.3 | $O(\bar{G} + R)$ |
| $H$ | Problem scale | Equation (8) | $O(\bar{G} + R)$ |
| $L$ | Lipschitz constant for the policy improvement operator | Assumption 2 | $[0, w/(H\sigma)]$ |
| $K$ | Local update period | Section 4 | $\mathbb{N}$ |
| $\epsilon_p, \epsilon_r$ | Environmental heterogeneity ratio | Definitions 1 and 2 | $[0, 2]$ |
| $\Lambda$ | Environmental heterogeneity | Theorem 1 | $O(H(\epsilon_p + \epsilon_r))$ |
| $\lambda^{(i)}, \lambda$ | Exploration constant | Equation (18) | $(0, 1)$ |
| $w_i, w$ | Convergence constant | Equation (19) | $[(1 - \gamma)\lambda/2, 1/2]$ |
| $\tau$ | Backtracking period | Lemma I.1 | $O(\log T)$ |
| $\alpha_0$ | Initial step-size | Section 4 | $(0, \min\{1/8K, w/64\}]$ |
| $\alpha_t$ | General step-size | Section 4 | $O(1/t)$ |
| $C_{\text{drift}}$ | Client drift constant | Lemma I.4 | $O(KH)$ |
| $C_{\text{prog}}$ | Parameter progress constant | Lemma I.5 | $O(H\tau)$ |
| $C_{\text{back}}$ | Backtracking constant | Lemma I.7 | $O(\tau^2 w)$ |
| $C_{\text{var}}$ | Gradient variance constant | Lemma I.8 | $O(H^2 w^2 \tau^4))$ |
| $\beta$ | Young's inequality constant | Appendix J | $(0, w/7)$ |
| $H_{\text{drift}}$ | Another drift constant | Appendix J | $O(H)$ |
| $C_\alpha$ | Step-size constant | Appendix J | $O(1)$ |
| $C_1$ | First-order constant | Equation (50) | $O((1 - \gamma)^{-1})$ |
| $C_2$ | Second-order constant | Equation (50) | $O(H^2\tau)$ |
| $C_3$ | Third-order constant | Equation (50) | $O(H^2 w \tau^4)$ |
| $C_4$ | Fourth-order constant | Equation (50) | $O(H^2 w^2 \tau^5)$ |
| $B$ | Square of the convergence region radius for constant step-size | Corollary 2.1 | see Corollary 2.1 |

where $P_\theta(s, s') = \int_{\mathcal{A}} P_a(s, s') \pi_\theta(a|s) \mathrm{d}a$, and

$$\|P_\theta\|_{\text{TV}} = \sup_{\|q\|_{\text{TV}} = 1} \|q P_\theta\|_{\text{TV}} = \sup_{\|q\|_{\text{TV}} = 1} \left\| \int_{\mathcal{S}} q(s) P_\theta(s, \cdot) \mathrm{d}s \right\|_{\text{TV}}.$$

And the constant $\sigma'$ is defined by

$$\sigma' = \hat{n} + \frac{m\rho^{\hat{n}}}{1 - \rho}, \tag{9}$$

where $\hat{n} = \lceil \log_\rho m^{-1} \rceil$, $m := \max_{i \in [N]} m_i$, and $\rho := \max_{i \in [N]} \rho_i$ with $m_i, \rho_i$ specified in Assumption 1. Note that in the above inequalities, we actually should use $\sigma'_i$ defined by $m_i$ and $\rho_i$; but $\sigma'_i$ is bounded by $\sigma'$ for all $i \in [N]$, so we use this possibly looser bound for notational simplicity.

Then, by Assumption 2, we have

$$
\begin{aligned}
\|P_{\theta_1} - P_{\theta_2}\|_{\mathrm{TV}} &= \sup_{\|q\|_{\mathrm{TV}}=1} \int_{\mathcal{S}} \left| \int_{\mathcal{S}} q(s)(P_{\theta_1}(s,s') - P_{\theta_2}(s,s'))\mathrm{d}s' \right| \mathrm{d}s \\
&= \sup_{\|q\|_{\mathrm{TV}}=1} \int_{\mathcal{S}} \left| \int_{\mathcal{S}\times\mathcal{A}} q(s)(P_a(s,s')\pi_{\theta_1}(a|s) - P_a(s,s')\pi_{\theta_2}(a|s))\mathrm{d}a\mathrm{d}s' \right| \mathrm{d}s \\
&\leq \sup_{\|q\|_{\mathrm{TV}}=1} \int_{\mathcal{S}^2\times\mathcal{A}} |q(s)| \, P_a(s,s') \, |\pi_{\theta_1}(a|s) - \pi_{\theta_2}(a|s)| \, \mathrm{d}a\mathrm{d}s'\mathrm{d}s \\
&= \sup_{\|q\|_{\mathrm{TV}}=1} \int_{\mathcal{S}\times\mathcal{A}} |q(s)| \, |\pi_{\theta_1}(a|s) - \pi_{\theta_2}(a|s)| \, \mathrm{d}a\mathrm{d}s \\
&= \sup_{\|q\|_{\mathrm{TV}}=1} \int_{\mathcal{S}} |q(s)| \, \|\pi_{\theta_1}(\cdot|s) - \pi_{\theta_2}(\cdot|s)\|_{\mathrm{TV}} \, \mathrm{d}s \\
&\leq L\|\theta_1 - \theta_2\|_2 \sup_{\|q\|_{\mathrm{TV}}=1} \int_{\mathcal{S}} |q(s)| \, \mathrm{d}s \\
&= L\|\theta_1 - \theta_2\|_2.
\end{aligned}
$$

Therefore, we get

$$
\|\eta_{\theta_1} - \eta_{\theta_2}\|_{\mathrm{TV}} \leq L\sigma'\|\theta_1 - \theta_2\|_2.
$$

Next, for the state-action distribution, we have

$$
\begin{aligned}
\|\mu_{\theta_1} - \mu_{\theta_2}\|_{\mathrm{TV}} &= \int_{\mathcal{S}\times\mathcal{A}} |\eta_{\theta_1}(s)\pi_{\theta_1}(a|s) - \eta_{\theta_2}(s)\pi_{\theta_2}(a|s)| \, \mathrm{d}s\mathrm{d}a \\
&\leq \int_{\mathcal{S}\times\mathcal{A}} \eta_{\theta_1}(s) |\pi_{\theta_1}(a|s) - \pi_{\theta_2}(a|s)| \, \mathrm{d}s\mathrm{d}a + \int_{\mathcal{S}\times\mathcal{A}} |\eta_{\theta_1}(s) - \eta_{\theta_2}(s)| \, \pi_{\theta_2}(a|s)\mathrm{d}a\mathrm{d}s \\
&\leq L\|\theta_1 - \theta_2\|_2 + \|\eta_{\theta_1} - \eta_{\theta_2}\|_{\mathrm{TV}} \\
&\leq L(1 + \sigma')\|\theta_1 - \theta_2\|_2.
\end{aligned}
$$

Similarly, we have

$$
\|\varphi_{\theta_1} - \varphi_{\theta_2}\|_{\mathrm{TV}} \leq L(2 + \sigma')\|\theta_1 - \theta_2\|_2.
$$

Also by (Mitrophanov, 2005, Corollary 3.1), we get

$$
\left\|\eta_\theta^{(i)} - \eta_\theta^{(j)}\right\|_{\mathrm{TV}} \leq \sigma'\|P_\theta^{(i)} - P_\theta^{(j)}\|_{\mathrm{TV}} \leq \sigma'\epsilon_p,
$$

where $\epsilon_p$ is defined in Definition 1. Then, for the state-action distribution, we have

$$
\left\|\mu_\theta^{(i)} - \mu_\theta^{(j)}\right\|_{\mathrm{TV}} = \left\|\eta_\theta^{(i)} \cdot \pi_\theta - \eta_\theta^{(j)} \cdot \pi_\theta\right\|_{\mathrm{TV}} = \left\|\eta_\theta^{(i)} - \eta_\theta^{(j)}\right\|_{\mathrm{TV}} \leq \sigma'\epsilon_p.
$$

And similarly, we have

$$
\begin{aligned}
&\left\|\varphi_\theta^{(i)} - \varphi_\theta^{(j)}\right\|_{\mathrm{TV}} \\
&= \int_{S^2\times A^2} \left| \mu_\theta^{(i)}(s,a)\pi_\theta(a|s)P_a^{(i)}(s,s')\pi_\theta(a'|s') - \mu_\theta^{(j)}(s,a)\pi_\theta(a|s)P_a^{(j)}(s,s')\pi_\theta(a'|s') \right| \mathrm{d}s\mathrm{d}s'\mathrm{d}a\mathrm{d}a' \\
&\leq \int_{S^2\times A^2} \left| \mu_\theta^{(i)}(s,a)\pi_\theta(a|s)P_a^{(i)}(s,s')\pi_\theta(a'|s') - \mu_\theta^{(j)}(s,a)\pi_\theta(a|s)P_a^{(i)}(s,s')\pi_\theta(a'|s') \right| \mathrm{d}s\mathrm{d}s'\mathrm{d}a\mathrm{d}a' \\
&\quad + \int_{S^2\times A^2} \left| \mu_\theta^{(j)}(s,a)\pi_\theta(a|s)P_a^{(i)}(s,s')\pi_\theta(a'|s') - \mu_\theta^{(j)}(s,a)\pi_\theta(a|s)P_a^{(j)}(s,s')\pi_\theta(a'|s') \right| \mathrm{d}s\mathrm{d}s'\mathrm{d}a\mathrm{d}a' \\
&\leq \left\|\mu_\theta^{(i)} - \mu_\theta^{(j)}\right\|_{\mathrm{TV}} + \|P^{(i)} - P^{(j)}\|_{\mathrm{TV}} \\
&\leq (\sigma' + 1)\epsilon_p \leq \sigma\epsilon_p
\end{aligned}
$$

Finally, by the triangle inequality, we get

$$
\left\|\mu_{\theta^{(i)}}^{(i)} - \mu_{\theta^{(j)}}^{(j)}\right\|_{\mathrm{TV}} \leq \sigma'\epsilon_p + L\sigma\|\theta^{(i)} - \theta^{(j)}\|_2.
$$

$\square$

Similarly, we can bound the differences between TD operators defined in Definition 5.

**Lemma G.2** (TD operator differences). *For the same MDP, the difference between the mean-path TD operators with regard to different parameters is bounded as follows:*

$$
\begin{cases}
\left\| \bar{A}_{\theta_1} - \bar{A}_{\theta_2} \right\| & \leq (1+\gamma) L \sigma \left\| \theta_1 - \theta_2 \right\|_2, \\
\left\| \bar{b}_{\theta_1} - \bar{b}_{\theta_2} \right\| & \leq R L \sigma \left\| \theta_1 - \theta_2 \right\|_2.
\end{cases}
$$

*Next, for a fixed parameter $\theta$, the difference between the mean-path TD operators with regard to different MDPs is bounded as follows:*

$$
\begin{cases}
\left\| \bar{A}_{\theta}^{(i)} - \bar{A}_{\theta}^{(j)} \right\| & \leq (1+\gamma) \sigma \epsilon_p, \\
\left\| \bar{b}_{\theta}^{(i)} - \bar{b}_{\theta}^{(j)} \right\| & \leq R(\epsilon_r + \sigma \epsilon_p).
\end{cases}
$$

*Then, by the triangle inequality, we get*

$$
\begin{cases}
\left\| \bar{A}_{\theta^{(i)}}^{(i)} - \bar{A}_{\theta^{(j)}}^{(j)} \right\| & \leq (1+\gamma) \sigma \left( L \left\| \theta^{(i)} - \theta^{(j)} \right\|_2 + \epsilon_p \right), \\
\left\| \bar{b}_{\theta^{(i)}}^{(i)} - \bar{b}_{\theta^{(j)}}^{(j)} \right\| & \leq R(\epsilon_r + \sigma \epsilon_p) + R L \sigma \left\| \theta^{(i)} - \theta^{(j)} \right\|_2.
\end{cases}
$$

*Proof.* For the same MDP, by Definition 5, we have

$$
\begin{aligned}
\left\| \bar{A}_{\theta_1} - \bar{A}_{\theta_2} \right\| &= \left\| \int_{\mathcal{S}^2 \times \mathcal{A}^2} \phi(s,a)(\gamma \phi^T(s',a') - \phi^T(a,s))(\mathrm{d}\varphi_{\theta_1}(s,a,s',a') - \mathrm{d}\varphi_{\theta_2}(s,a,s',a')) \right\| \\
&\leq (1+\gamma) \left\| \varphi_{\theta_1} - \varphi_{\theta_2} \right\|_{\mathrm{TV}} \\
&\leq (1+\gamma) L \sigma \left\| \theta_1 - \theta_2 \right\|_2,
\end{aligned}
$$

where the last inequality comes from Lemma G.1. Similarly, we have

$$
\begin{aligned}
\left\| \bar{b}_{\theta_1} - \bar{b}_{\theta_2} \right\| &= \left\| \int_{\mathcal{S} \times \mathcal{A}} \phi(s,a) r(s,a)(\mathrm{d}\mu_{\theta_1}(s,a) - \mathrm{d}\mu_{\theta_2}(s,a)) \right\| \\
&\leq R \left\| \mu_{\theta_1} - \mu_{\theta_2} \right\|_{\mathrm{TV}} \\
&\leq R L \sigma \left\| \theta_1 - \theta_2 \right\|_2.
\end{aligned}
$$

Then for the same parameter $\theta$, we have

$$
\begin{aligned}
\left\| \bar{A}_{\theta}^{(i)} - \bar{A}_{\theta}^{(j)} \right\| &= \left\| \int_{\mathcal{S}^2 \times \mathcal{A}^2} \phi(s,a)(\gamma \phi^T(s',a') - \phi^T(a,s))(\mathrm{d}\varphi_{\theta}^{(i)}(s,a,s',a') - \mathrm{d}\varphi_{\theta}^{(j)}(s,a,s',a')) \right\| \\
&\leq (1+\gamma) \left\| \varphi_{\theta}^{(i)} - \varphi_{\theta}^{(j)} \right\|_{\mathrm{TV}} \\
&\leq (1+\gamma) \sigma \epsilon_p,
\end{aligned}
$$

where the last inequality comes from Lemma G.1. Similarly, we have

$$
\begin{aligned}
\left\| \bar{b}_{\theta}^{(i)} - \bar{b}_{\theta}^{(j)} \right\| &= \left\| \int_{\mathcal{S} \times \mathcal{A}} \phi(s,a) \left( r^{(i)}(s,a) \mathrm{d}\mu_{\theta}^{(i)}(s,a) - r^{(j)}(s,a) \mathrm{d}\mu_{\theta}^{(j)}(s,a) \right) \right\| \\
&\leq \int_{\mathcal{S} \times \mathcal{A}} \left| r^{(i)}(s,a) - r^{(j)}(s,a) \right| \mathrm{d}\mu_{\theta}^{(i)}(s,a) + \int_{\mathcal{S} \times \mathcal{A}} r^{(j)}(s,a) \left| \mathrm{d}\mu_{\theta}^{(i)}(s,a) - \mathrm{d}\mu_{\theta}^{(j)}(s,a) \right| \\
&\leq R \epsilon_r + R \sigma' \epsilon_p,
\end{aligned}
$$

where the last inequality comes from Definition 2 and Lemma G.1. $\qquad\square$

## H PROOF OF THEOREM 1

**Theorem 1.** *For any $i, j \in [\bar{N}]$, we have*

$$
\left\| \theta_*^{(j)} - \theta_*^{(i)} \right\|_2 \leq \frac{1}{w_j} \left( R \epsilon_r + H \sigma \epsilon_p \right) \leq \frac{\Lambda(\epsilon_p, \epsilon_r)}{w},
$$

*where $w := \min_{i \in [\bar{N}]} w_i$; $w_i$ is defined in Lemma I.2 and $\Lambda(\epsilon_p, \epsilon_r)$ is defined in Lemma I.3.*

*Proof.* First, we formulate the Bellman optimal equation in terms of TD operators defined in Definition 5:

$$\bar{A}_*^{(i)}\theta_*^{(i)} + \bar{b}_*^{(i)} = 0,$$

for any $i \in [\bar{N}]$, where

$$\bar{A}_*^{(i)} := \bar{A}_{\theta_*^{(i)}}^{(i)}, \quad \bar{b}_*^{(i)} := \bar{b}_{\theta_*^{(i)}}^{(i)}.$$

Then for any $i, j \in [\bar{N}]$, we have

$$\left(\bar{A}_*^{(j)} - \bar{A}_*^{(i)}\right)\theta_*^{(i)} + \bar{A}_*^{(j)}\left(\theta_*^{(j)} - \theta_*^{(i)}\right) = \bar{b}_*^{(i)} - \bar{b}_*^{(j)}.$$

By Tsitsiklis & Van Roy (1996, Theorem 2), $\bar{A}_*^{(j)}$ is negative definite Therefore, $\bar{A}_*^{(j)}$ is non-singular, and we get

$$\left\|\theta_*^{(j)} - \theta_*^{(i)}\right\|_2 \le \left\|\left(\bar{A}_*^{(j)}\right)^{-1}\right\|\left\|\left(\bar{A}_*^{(i)} - \bar{A}_*^{(j)}\right)\theta_*^{(i)} + \left(\bar{b}_*^{(i)} - \bar{b}_*^{(j)}\right)\right\|_2.$$

And we have

$$\left\|\left(\bar{A}_*^{(j)}\right)^{-1}\right\| = \sigma_{\min}^{-1}\left(\bar{A}_*^{(j)}\right) \tag{10}$$

$$= \frac{1}{\left|\lambda_{\max}\left(\bar{A}_*^{(j)}\right)\right|} \tag{11}$$

$$\le \frac{1}{-\Re\lambda_{\max}\left(\bar{A}_*^{(j)}\right)} \tag{12}$$

$$\le \frac{1}{-\lambda_{\max}\left(\operatorname{sym}\left(\bar{A}_*^{(j)}\right)\right)} \tag{13}$$

$$= \frac{1}{2w_j}, \tag{14}$$

where (10) uses the spectrum norm equality and $\sigma_{\min}$ returns the smallest singular value of a matrix; (11) and (12) use the fact that $\bar{A}_*^{(j)}$ is negative definite; (13) is by (Zhang, 2011, Theorem 10.28); and lastly, (14) is the definition of $w_j$ (see Lemma I.2).

Therefore, letting $G$ be large enough to contain $\{\theta_*^{(i)}\}_{i \in [\bar{N}]}$, we get

$$\left\|\theta_*^{(j)} - \theta_*^{(i)}\right\|_2 \le \frac{1}{2w_j}\left(\left\|A_*^{(i)} - A_*^{(j)}\right\|G + \left\|b_*^{(i)} - b_*^{(j)}\right\|\right).$$

By Lemma G.2, we get

$$\left\|\theta_*^{(j)} - \theta_*^{(i)}\right\|_2 \le \frac{1}{2w_j}\left((1+\gamma)\sigma G\left(\epsilon_p + L\left\|\theta_*^{(i)} - \theta_*^{(j)}\right\|_2\right) + R(\epsilon_r + \sigma\epsilon_p) + RL\sigma\left\|\theta_*^{(i)} - \theta_*^{(j)}\right\|_2\right)$$

$$\le \frac{1}{2w_j}\left(R\epsilon_r + H\sigma\epsilon_p + LH\sigma\left\|\theta_*^{(i)} - \theta_*^{(j)}\right\|_2\right).$$

We require that $LH\sigma \le w_j$ (the same restriction (20) in Lemma I.2); then we get

$$\left\|\theta_*^{(j)} - \theta_*^{(i)}\right\|_2 \le \frac{1}{2w_j}(R\epsilon_r + H\sigma\epsilon_p) + \frac{w_j}{2w_j}\left\|\theta_*^{(i)} - \theta_*^{(j)}\right\|_2,$$

which gives

$$\left\|\theta_*^{(j)} - \theta_*^{(i)}\right\|_2 \le \frac{1}{w_j}(R\epsilon_r + H\sigma\epsilon_p) \le \frac{\Lambda(\epsilon_p, \epsilon_r)}{w},$$

where $w := \min_{i \in [\bar{N}]} w_i$ and $\Lambda(\epsilon_p, \epsilon_r) := R\epsilon_r + H\sigma\epsilon_p$ (the same definition in Lemma I.3).

$\square$

# I  KEY LEMMAS

In this section, we first decompose the mean squared error and then present seven lemmas, each bounding one term in the decomposition.

## I.1  ERROR DECOMPOSITION

**Lemma I.1** (Error decomposition). *The one-step mean squared error can be decomposed recursively as follows:*

$$
\begin{aligned}
\mathbb{E}\left\|\bar{\theta}_{t+1}-\theta_*\right\|^2 &\leq \mathbb{E}\|\breve{\theta}_{t+1}-\theta_*\|^2 = \mathbb{E}\left\|\bar{\theta}_t-\theta_*\right\|^2 \\
&+ 2\alpha_t \mathbb{E}\left\langle \bar{\theta}_t-\theta_*, \bar{g}\left(\bar{\theta}_t\right)-\bar{g}\left(\theta_*\right)\right\rangle && \text{(descent direction)} \\
&+ \frac{2\alpha_t}{N}\sum_{i=1}^N \mathbb{E}\left\langle \bar{\theta}_t-\theta_*, \bar{g}^{(i)}\left(\bar{\theta}_t\right)-\bar{g}\left(\bar{\theta}_t\right)\right\rangle && \text{(gradient heterogeneity)} \\
&+ \frac{2\alpha_t}{N}\sum_{i=1}^N \mathbb{E}\left\langle \bar{\theta}_t-\theta_*, \left(\bar{g}^{(i)}\left(\theta_t^{(i)}\right)-\bar{g}^{(i)}\left(\bar{\theta}_t\right)\right)\right\rangle && \text{(client drift)} \\
&+ \frac{2\alpha_t}{N}\sum_{i=1}^N \mathbb{E}\left\langle \bar{\theta}_t-\theta_*, \bar{g}_{t-\tau}^{(i)}\left(\theta_t^{(i)}\right)-\bar{g}^{(i)}\left(\theta_t^{(i)}\right)\right\rangle && \text{(gradient progress)} \\
&+ \frac{2\alpha_t}{N}\sum_{i=1}^N \mathbb{E}\left\langle \bar{\theta}_t-\theta_*, g_{t-\tau}^{(i)}\left(\theta_t^{(i)}, \tilde{O}_t^{(i)}\right)-\bar{g}_{t-\tau}^{(i)}\left(\theta_t^{(i)}\right)\right\rangle && \text{(mixing)} \\
&+ \frac{2\alpha_t}{N}\sum_{i=1}^N \mathbb{E}\left\langle \bar{\theta}_t-\theta_*, g_t^{(i)}\left(\theta_t^{(i)}, O_t^{(i)}\right)-g_{t-\tau}^{(i)}\left(\theta_t^{(i)}, \tilde{O}_t^{(i)}\right)\right\rangle && \text{(backtracking)} \\
&+ \alpha_t^2 \mathbb{E}\left\|\frac{1}{N}\sum_{i=1}^N g_t^{(i)}\left(\theta_t^{(i)}\right)\right\|^2. && \text{(gradient variance)}
\end{aligned}
$$

One can verify the above decomposition given $\bar{g}(\theta_*)=0$.

## I.2  DESCENT DIRECTION

**Lemma I.2** (Descent direction). *There exist positive constants $\{w_i\}_{i\in[\bar{N}]}$ such that for any $\|\theta\| \leq G$, we have*

$$
\left\langle \theta-\theta_*^{(i)}, \bar{g}^{(i)}(\theta)-\bar{g}^{(i)}(\theta_*^{(i)})\right\rangle \leq -w_i\left\|\theta-\theta_*^{(i)}\right\|^2, \quad \forall i\in[\bar{N}].
$$

*Proof.* We drop the subscript $(i)$ in this lemma since the following derivation holds for all MDPs. We first denote $\Delta\theta = \theta - \theta_*$. Then, we have

$$
\begin{aligned}
\langle \theta-\theta_*, \bar{g}(\theta)-\bar{g}(\theta_*)\rangle &= \Delta\theta^T\left(\left(\bar{A}_\theta\theta+\bar{b}_\theta\right)-\left(\bar{A}_{\theta_*}\theta_*+\bar{b}_{\theta_*}\right)\right) \\
&= \Delta\theta^T\bar{A}_{\theta_*}\Delta\theta + \Delta\theta^T\left(\bar{A}_\theta-\bar{A}_{\theta_*}\right)\theta + \Delta\theta^T(\bar{b}_\theta-\bar{b}_{\theta_*}) \\
&\leq \Delta\theta^T\bar{A}_{\theta_*}\Delta\theta + \|\Delta\theta\|\left\|\bar{A}_\theta-\bar{A}_{\theta_*}\right\|\|\theta\| + \|\Delta\theta\|\left\|\bar{b}_\theta-\bar{b}_{\theta_*}\right\| \\
&\leq \Delta\theta^T\bar{A}_{\theta_*}\Delta\theta + (1+\gamma)L\sigma\|\theta\|\|\Delta\theta\|^2 + RL\sigma\|\Delta\theta\|^2 && (15) \\
&= \Delta\theta^T(\bar{A}_{\theta_*}+L\sigma(R+(1+\gamma)\|\theta\|)I)\Delta\theta \\
&\leq \Delta\theta^T\left(\bar{A}_{\theta_*}+L\sigma H\cdot I\right)\Delta\theta && (16) \\
&=: \Delta\theta^T\widetilde{A}_{\theta_*}\Delta\theta,
\end{aligned}
$$

where (15) uses Lemma G.2, and (16) uses the fact that $\|\theta\| \leq G$ and $H := R+(1+\gamma)G$. By (Tsitsiklis & Van Roy, 1996, Theorem 2), $\bar{A}_{\theta_*}$ is negative definite in the sense that $x^*Ax < 0$

for any vector $x \in \mathbb{R}^d$. Specifically, for any nonzero $x \in \mathbb{R}^d$, we denote $u = x^T \phi(S, A)$ and $u' = x^T \phi(S', A')$. Then, for any $x \neq 0$, by Definition 5, we have

$$x^T \bar{A}_\theta x = \mathbb{E}_{\varphi_\theta} \left[ \gamma u u' - u^2 \right] = \gamma \mathbb{E}[uu'] - \mathbb{E}[u^2] \leq \frac{\gamma}{2} \left( \mathbb{E}[u^2] + \mathbb{E}[u'^2] \right) - \mathbb{E}[u^2] = (\gamma - 1)\mathbb{E}[u^2] < 0, \tag{17}$$

where we use the fact that $\mathbb{E}[u^2] = \mathbb{E}[u'^2]$ under a steady distribution. Let $\Phi_\theta^{(i)} := \mathbb{E}_{\mu_\theta^{(i)}}[\phi(S,A)\phi^T(S,A)] \succ 0$. We define

$$\lambda^{(i)} := \lambda_{\min}\left( \Phi_{\theta^{(i)}}^{(i)} \right), \quad \lambda := \min_{i \in [\bar{N}]} \lambda^{(i)}. \tag{18}$$

By (17), we have

$$-w_i := \frac{1}{2}\lambda_{\max}\left( \mathrm{sym}\left( \bar{A}_{\theta^{(i)}}^{(i)} \right) \right) \leq \frac{1}{2}(\gamma - 1)\lambda_{\min}\left( \Phi_{\theta^{(i)}}^{(i)} \right) = \frac{\gamma - 1}{2}\lambda^{(i)}. \tag{19}$$

where $\mathrm{sym}(A) := \frac{1}{2}(A + A^*)$ maps general matrices to Hermitian matrices. Due to the positive definiteness of $\Phi_{\theta^{(i)}}^{(i)}$, We know $w_i > 0$ for any $i \in [\bar{N}]$. By Zhang (2011, Theorem 10.21) and the linearity of the $\mathrm{sym}$ function, we know

$$\lambda_{\max}\left( \mathrm{sym}\left( \widetilde{A}_{\theta^{(i)}}^{(i)} \right) \right) \leq \lambda_{\max}\left( \mathrm{sym}\left( \bar{A}_{\theta^{(i)}}^{(i)} \right) \right) + \lambda_{\max}(\mathrm{sym}(L\sigma H \cdot I)) = -2w_i + L\sigma H.$$

Let $w = \min_{i \in [\bar{N}]}\{w_i\}$. Then, we can choose $L$ to be small enough such that

$$L \leq \frac{w}{\sigma H}, \tag{20}$$

which gives

$$-w_i \geq \lambda_{\max}\left( \mathrm{sym}\left( \widetilde{A}_{\theta^{(i)}}^{(i)} \right) \right).$$

Therefore, for any $i \in [\bar{N}]$, we have

$$\left\langle \theta - \theta_*^{(i)}, \bar{g}^{(i)}(\theta) - \bar{g}^{(i)}(\theta_*^{(i)}) \right\rangle \leq \lambda_{\max}\left( \mathrm{sym}\left( \widetilde{A}_{\theta^{(i)}}^{(i)} \right) \right) \left\| \theta - \theta_*^{(i)} \right\|^2 \leq -w_i \left\| \theta - \theta_*^{(i)} \right\|^2. \tag{21}$$

$\square$

*Remark* 1 (Convergence constant). Equation (21) mirrors the result of stochastic gradient descent (SGD) (Bottou et al., 2018), with $w$ being analogous to the Lipschitz constant of a function's gradient. Therefore, similar to SGD, $w$ controls the convergence rate of our algorithm.

*Remark* 2 (Exploration constant). The value of $w$ depends on $\lambda$, a constant that reflects the *exploration difficulty* of the environment. We can see this by considering a simple tabular setting, where the feature map $\phi$ is simply the indicator function (see Appendix M for detailed definitions). Then $\mathbb{E}_\mu[\phi(S,A)\phi^T(S,A)]$ reduces to $\mathrm{diag}\{\mu(s,a)\}_{(s,a)\in\mathcal{S}\times\mathcal{A}}$. In this case, the minimal eigenvalue of $\Phi$ is $\min_{(s,a)\in\mathcal{S}\times\mathcal{A}} \mu(s,a)$, i.e., the probability of visiting the least probable state-action pair under the steady distribution.

We say an environment is *hard to explore* if some state-action pairs have a very small probability of being visited under the steady distribution, then $\lambda$ is small. Conversely, $\lambda$ is large when the environment is easy to explore. Intuitively, an environment that is hard to explore requires more samples to learn an optimal policy.

In the context of LFA, the value of $\lambda$, and consequently $w$, is determined by the conditions of both the MDPs and the feature map $\phi$. If the environments in the feature space are easy to explore under the MDPs, $\lambda$ and $w$ will take on larger values, and the algorithm converges faster.

### I.3 GRADIENT HETEROGENEITY

**Lemma I.3** (Gradient heterogeneity). *For $\|\theta\| \leq G$, we have*

$$\left\| \bar{g}(\theta) - \frac{1}{N}\sum_{i=1}^N \bar{g}^{(i)}(\theta) \right\| \leq H\sigma\epsilon_p + R\epsilon_r =: \Lambda(\epsilon_p, \epsilon_r)$$

*Proof.* Directly applying the decomposition in Definition 5 and Lemma G.2 gives

$$\left\| \bar{g}(\theta) - \frac{1}{N}\sum_{i=1}^{N}\bar{g}^{(i)}(\theta) \right\| = \left\| (\bar{A}_\theta \theta + \bar{b}_\theta) - \frac{1}{N}\sum_{i=1}^{N}(\bar{A}_\theta^{(i)}\theta + \bar{b}_\theta^{(i)}) \right\|$$

$$\leq \frac{1}{N}\sum_{i=1}^{N}\left( \left\| \bar{b}_\theta - \bar{b}_\theta^{(i)} \right\| + \|\bar{A}_\theta - \bar{A}_\theta^{(i)}\|\|\theta\| \right)$$

$$\leq \sigma\epsilon_p \left( R + (1+\gamma)\|\theta\| \right) + R\epsilon_r,$$

$\square$

### I.4 CLIENT DRIFT

Before bounding the gradient progress, we first bound the client drift.

**Lemma I.4** (Client drift). *If $\|\bar{\theta}_t\| \leq G$ holds for all $t \in \mathbb{N}$, then*

$$\frac{1}{N}\sum_{i=1}^{N}\left\| \bar{g}^{(i)}(\theta_t^{(i)}) - \bar{g}^{(i)}(\bar{\theta}_t) \right\|^2 \leq \alpha_{t-k}^2 (1+\gamma+\sigma LH)^2 C_{\text{drift}}^2,$$

*where $k$ is the smallest integer such that $t - k \equiv 0 \pmod{K}$, and*

$$C_{\text{drift}}^2 = 4K^2 H^2.$$

*Proof.* Similar to (15) in the proof of Lemma I.2, we have

$$\left\| \bar{g}^{(i)}(\theta_t^{(i)}) - \bar{g}^{(i)}(\bar{\theta}_t) \right\| \leq \left( 1 + \gamma + L\sigma \left( R + (1+\gamma)\|\bar{\theta}_t\| \right) \right) \left\| \theta_t^{(i)} - \bar{\theta}_t \right\| \tag{22}$$

Then, since $\|\bar{\theta}_t\| \leq G$, we have

$$\frac{1}{N}\sum_{i=1}^{N}\left\| \bar{g}^{(i)}(\theta_t^{(i)}) - \bar{g}^{(i)}(\bar{\theta}_t) \right\|^2 \leq (1+\gamma+\sigma LH)^2 \cdot \frac{1}{N}\sum_{i=1}^{N}\left\| \theta_t^{(i)} - \bar{\theta}_t \right\|^2. \tag{23}$$

Let $\Omega_t := \frac{1}{N}\sum_{i=1}^{N}\left\| \theta_t^{(i)} - \bar{\theta}_t \right\|^2$. We then need to bound $\Omega_t$. First, if $t \equiv 0 \pmod{K}$, we have $\Omega_t = 0$. Now suppose $t \not\equiv 0 \pmod{K}$. Let $k$ be the smallest integer such that $t - k \equiv 0 \pmod{K}$. Then we know that there is no aggregation step between time step $t - k$ and $t$, and $\bar{\theta}_{t-l} = 1/N\sum_{i=1}^{N}\theta_{t-l}^{(i)}$ for $0 \leq l \leq k$. Therefore, we have

$$\left\| \theta_t^{(i)} - \bar{\theta}_t \right\|^2 = \left\| \theta_{t-k}^{(i)} - \bar{\theta}_{t-k} + \sum_{l=1}^{k}\alpha_{t-l}\left( g_{t-l}^{(i)}(\theta_{t-l}^{(i)}) - \boldsymbol{g}_{t-l}(\boldsymbol{\theta}_{t-l}) \right) \right\|^2$$

$$\leq k\alpha_{t-k}^2\sum_{l=1}^{k}\left\| g_{t-l}^{(i)}(\theta_{t-l}^{(i)}) - \boldsymbol{g}_{t-l}(\boldsymbol{\theta}_{t-l}) \right\|^2,$$

where $\boldsymbol{g}_t(\boldsymbol{\theta}_t) = \frac{1}{N}\sum_{i=1}^{N}g_t^{(i)}(\theta_t^{(i)})$, and we choose $\alpha$ to be non-increasing. Since for a random vector $X$, $\text{Var}(X) \leq \mathbb{E}\|X\|^2$, we have

$$\Omega_t \leq k\alpha_{t-k}^2\sum_{l=1}^{k}\frac{1}{N}\sum_{i=1}^{N}\left\| g_{t-l}^{(i)}(\theta_{t-l}^{(i)}) - \boldsymbol{g}_{t-l}(\boldsymbol{\theta}_{t-l}) \right\|^2$$

$$\leq k\alpha_{t-k}^2\sum_{l=1}^{k}\frac{1}{N}\sum_{i=1}^{N}\left\| g_{t-l}^{(i)}(\theta_{t-l}^{(i)}) \right\|^2$$

$$\leq k\alpha_{t-k}^2\sum_{l=1}^{k}\frac{1}{N}\sum_{i=1}^{N}2\left( \left\| g_{t-l}^{(i)}(\theta_{t-l}^{(i)}) - g_{t-l}^{(i)}(\bar{\theta}_{t-l}) \right\|^2 + \left\| g_{t-l}^{(i)}(\bar{\theta}_{t-l}) \right\|^2 \right)$$

where we also used Jensen's inequality. By the definition of the Markovian semi-gradients (see Definition 4), they are linear and Lipschitz continuous with the Lipschitz constant bounded by $\|A_{t-l}^{(i)}\| \le 1 + \gamma$. However, it is worth emphasizing that the mean-path semi-gradients are non-linear and non-Lipschitz continuous (unless $\|\bar{\theta}_t\|$ is bounded; see (22)). Given the Lipschitz continuity, we have

$$\Omega_t \le 2k\alpha_{t-k}^2 \sum_{l=1}^{k} \frac{1}{N} \sum_{i=1}^{N} \left( (1+\gamma)^2 \left\| \theta_{t-l}^{(i)} - \bar{\theta}_{t-l} \right\|^2 + H^2 \right)$$

$$= 2k\alpha_{t-k}^2 \left( kH^2 + (1+\gamma)^2 \sum_{l=1}^{k} \Omega_{t-l} \right). \tag{24}$$

Recursively applying (24) gives

$$\Omega_t \le 2k\alpha_{t-k}^2 \left( kH^2 + (1+\gamma)^2 \sum_{l=2}^{k} \Omega_{t-l} \right) + 2k\alpha_{t-k}^2(1+\gamma)^2 \cdot 2(k-1)\alpha_{t-k}^2 \left( kH^2 + (1+\gamma)^2 \sum_{l=2}^{k} (\Omega_{t-l}) \right)$$

$$\le 2k\alpha_{t-k}^2 \left( 1 + 8(k-1)\alpha_{t-k}^2 \right) \left( kH^2 + (1+\gamma)^2 \sum_{l=2}^{k} \Omega_{t-l} \right)$$

$$\le 2k\alpha_{t-k}^2 \prod_{j=1}^{k} \left( 1 + 8(k-j)\alpha_{t-k}^2 \right) \left( kH^2 + (1+\gamma)^2 \Omega_{t-k} \right)$$

$$\le 2k\alpha_{t-k}^2 \left( 1 + 8k\alpha_{t-k}^2 \right)^k \cdot kH^2,$$

where we use the fact that $\Omega_{t-k} = 0$. To continue, we impose a constraint on the initial step-size by requiring $4K\alpha_0 \le 1$, which gives $16k^2\alpha_{t-k}^2 \le 1$. Then, we have

$$(1 + 8k\alpha_{t-k}^2)^k \le 1 + \sum_{l=1}^{k} k^l (8k\alpha_{t-k}^2)^l \le 1 + \frac{8k^2\alpha_{t-k}^2}{1 - 8k^2\alpha_{t-k}^2} \le 1 + 16k^2\alpha_{t-k}^2 \le 2. \tag{25}$$

Therefore, we get

$$\Omega_t \le 2k^2H^2\alpha_{t-k}^2(1 + 16k^2\alpha_{t-k}^2) \le 4k^2H^2\alpha_{t-k}^2 \le \alpha_{t-k}^2 C_{\text{drift}}^2. \tag{26}$$

Plugging (26) back into (23) gives the final result.

$\square$

**Corollary I.4.1.** *For future reference, we extract two bounds on the client drift from the proof of Lemma I.4:*

$$\Omega_t \le \alpha_{t-k}^2 C_{\text{drift}}^2, \quad \omega_t \le \alpha_{t-k} C_{\text{drift}},$$

*where $\omega_t := \frac{1}{N} \sum_{i=1}^{N} \|\theta_t^{(i)} - \bar{\theta}_t\|$.*

## I.5 GRADIENT PROGRESS

To bound the gradient progress, we first need to bound the parameter progress. Instead of directly bounding the client parameter progress, we bound the central parameter progress, which then gives the client parameter progress combining Lemma I.4.

**Lemma I.5** (Central parameter progress). *If $\left\| \bar{\theta}_t \right\| \le G$ for any $l \le t$, then we have*

$$\left\| \bar{\theta}_t - \bar{\theta}_{t-\tau} \right\| \le \alpha_{sK} C_{\text{prog}}(\tau),$$

*where $s$ is the largest integer such that $sK \le t - \tau$ and*

$$C_{\text{prog}}(\tau) = 2(\tau + 2K)(H + 2\alpha_0 C_{\text{drift}}) = O(\tau).$$

*Proof.* Bounding the central parameter progress is harder than bounding the client parameter progress since

$$\|\bar{\theta}_t - \bar{\theta}_{t-1}\| \not\lesssim \alpha_{t-1}(R + (1+\gamma)\|\bar{\theta}_{t-1}\|).$$

Therefore we need to introduce the client drift, and then bound the parameter progress using Lemma I.4. First, for any $t$, we have

$$\|\bar{\theta}_t\| \le \|\breve{\theta}_t\| = \left\| \bar{\theta}_{t-1} + \frac{\alpha_{t-1}}{N} \sum_{i=1}^{N} g_{t-1}^{(i)} \left( \theta_{t-1}^{(i)} \right) \right\|$$

$$= \left\| \bar{\theta}_{t-1} + \frac{\alpha_{t-1}}{N} \sum_{i=1}^{N} \left( A^{(i)} \left( O_{t-1}^{(i)} \right) \bar{\theta}_{t-1} + A^{(i)} \left( O_{t-1}^{(i)} \right) \left( \theta_{t-1}^{(i)} - \bar{\theta}_{t-1} \right) + b^{(i)} \left( O_{t-1}^{(i)} \right) \right) \right\|$$

$$\le (1 + \alpha_{t-1}(1+\gamma)) \|\bar{\theta}_{t-1}\| + \alpha_{t-1}(R + 2\omega_{t-1}), \tag{27}$$

where $\omega_{t-1}$ is defined in Corollary I.4.1. Let $k$ be the smallest positive integer such that $t - k \equiv 0 \pmod{K}$ (if $t \equiv 0 \pmod{K}$, then $k = K$). Recursively applying (27) gives

$$\|\bar{\theta}_t\| \le \prod_{l=t-k}^{t-1} (1 + 2\alpha_l) \|\bar{\theta}_{t-k}\| + \sum_{j=0}^{k-1} (1 + 2\alpha_{t-j})^j \alpha_{t-j-1}(R + 2\omega_{t-1})$$

$$\le (1 + 2\alpha_{t-k})^k \|\bar{\theta}_{t-k}\| + \alpha_{t-k}(R + 2\alpha_{t-k}C_{\text{drift}}) \frac{(1 + 2\alpha_{t-k})^k - 1}{2\alpha_{t-k}} \tag{28}$$

$$\le (1 + 4k\alpha_{t-k}) \|\bar{\theta}_{t-k}\| + 2k\alpha_{t-k}(R + 2\alpha_{t-k}C_{\text{drift}}) \tag{29}$$

$$\le 2\|\bar{\theta}_{t-k}\| + 2k\alpha_{t-k}(R + 2\alpha_{t-k}C_{\text{drift}}), \tag{30}$$

where (28) uses Corollary I.4.1 and we require $\alpha$ to be non-increasing; and in (29) and (30), we require that $4\alpha_0 K \le 1$, which gives $(1 + 2\alpha_{t-k})^k \le 1 + 4k\alpha_{t-k}$ with the similar reasoning in (25).

Now we are ready to bound any central parameter progress between two aggregation steps. Since $t - k \equiv 0 \pmod{K}$, we have $\|\bar{\theta}_{t-k}\| \le \bar{G}$. Then by (7), we get

$$\|\bar{\theta}_t - \bar{\theta}_{t-k}\| \le \left\| \breve{\theta}_t - \bar{\theta}_{t-k} \right\| \le \sum_{l=t-k}^{t-1} \|\bar{\theta}_{l+1} - \bar{\theta}_l\|$$

$$\overset{(27)}{\le} \sum_{l=t-k}^{t-1} \alpha_l \left( R + (1+\gamma)\|\bar{\theta}_l\| + 2\omega_l \right)$$

$$\overset{(30)}{\le} k\alpha_{t-k}(R + 2(1+\gamma)\|\bar{\theta}_{t-k}\| + 4k\alpha_{t-k}(R + 2\alpha_{t-k}C_{\text{drift}}) + 2\alpha_{t-k}C_{\text{drift}})$$

$$\le 2k\alpha_{t-k} \left( R + (1+\gamma)\|\bar{\theta}_{t-k}\| + 2\alpha_{t-k}C_{\text{drift}} \right), \tag{31}$$

where we use the fact that $\gamma < 1$ and $4k\alpha_{t-k} \le 1$.

Finally, we need to bound the central parameter progress for general time period $\tau$. For any $t > \tau > 1$, let $s$ be the largest integer such that $sK \le t - \tau$. And let $s'$ be the largest integer such that $s'K \le t$. Then we have

$$\|\bar{\theta}_t - \bar{\theta}_{t-\tau}\| \le \sum_{j=1}^{s'-s} \|\bar{\theta}_{(s+j)K} - \bar{\theta}_{(s+j-1)K}\| + \|\bar{\theta}_t - \bar{\theta}_{s'K}\| + \|\bar{\theta}_{t-\tau} - \bar{\theta}_{sK}\|$$

$$\overset{(31)}{\le} 2(\tau + 2K)\alpha_{sK}(R + 2\alpha_{sK}C_{\text{drift}})$$

$$+ 2(1+\gamma)\alpha_{sK} \left( \sum_{j=1}^{s'-s} K\|\bar{\theta}_{(s+j-1)K}\| + (t - s'K)\|\bar{\theta}_{s'K}\| + (t - \tau - sK)\|\bar{\theta}_{sK}\| \right)$$

$$\le 2\alpha_{sK}(\tau + 2K)(R + 2\alpha_{sK}C_{\text{drift}}) + 2\alpha_{sK}(\tau + 2K)(1+\gamma)G$$

$$\le 2\alpha_{sK}(\tau + 2K)(R + (1+\gamma)G + 2\alpha_{sK}C_{\text{drift}})$$

$$\le \alpha_{sK}C_{\text{prog}}(\tau), \tag{32}$$

where $C_{\text{prog}}(\tau) := 2(\tau + 2K)(H + 2\alpha_0 C_{\text{drift}}) = O(\tau)$.

$$\square$$

**Corollary I.5.1** (Client parameter progress). *If $\left\|\bar{\theta}_l\right\| \leq G$ holds for all $l \leq t$, we also have*

$$\left\|\theta_t^{(i)} - \theta_{t-\tau}^{(i)}\right\| \leq \alpha_{sK} C_{\text{prog}}(\tau),$$

*where $s$ is the largest integer such that $sK \leq t - \tau$.*

*Proof.* If $t \equiv 0 \pmod{K}$ and $t - \tau \equiv 0 \pmod{K}$, then $\theta_t^{(i)} = \bar{\theta}_t$ and $\theta_{t-\tau}^{(i)} = \bar{\theta}_{t-\tau}$, and the result directly follows Lemma I.5. Without loss of generality, we assume $t \not\equiv 0 \pmod{K}$ and $t - \tau \not\equiv 0 \pmod{K}$. Let $s$ be the largest integer such that $sK < t - \tau$. And let $s'$ be the largest integer such that $s'K < t$. Similar to (32), we have

$$\left\|\theta_t^{(i)} - \theta_{t-\tau}^{(i)}\right\| \leq \|\bar{\theta}_{s'K} - \bar{\theta}_{sK}\| + \left\|\theta_t^{(i)} - \bar{\theta}_{s'K}\right\| + \left\|\theta_{t-\tau}^{(i)} - \bar{\theta}_{sK}\right\|.$$

By Lemma I.5, we have

$$\left\|\bar{\theta}_{s'K} - \bar{\theta}_{sK}\right\| \leq \alpha_{sK} C_{\text{prog}}(s'K - sK - 2K),$$

where we subtract $2K$ to offset the addition of $2K$ in Lemma I.5 for general $t$ and $\tau$.

Then to bound the client parameter progress after a synchronization, we first notice that when $t \not\equiv 0 \pmod{K}$, we have

$$\left\|\theta_t^{(i)} - \theta_{t-1}^{(i)}\right\| \leq 2\alpha_{t-1} \left\|\theta_{t-1}^{(i)}\right\| + \alpha_{t-1} R.$$

Similar to (27)-(30), we have

$$\begin{aligned}
\left\|\theta_t^{(i)}\right\| &\leq (1 + 2\alpha_{t-1}) \left\|\theta_t^{(i)}\right\| + \alpha_{t-1} R \\
&\leq \prod_{l=t-k}^{t-1} (1 + 2\alpha_l) \|\bar{\theta}_{t-k}\| + R \sum_{j=0}^{k-1} (1 + 2\alpha_{t-j})^j \alpha_{t-j-1} \\
&\leq 2 \left(\|\bar{\theta}_{t-k}\| + k\alpha_{t-k} R\right),
\end{aligned}$$

where $k$ is the smallest integer such that $t - k \equiv 0 \pmod{K}$. Then, we get

$$\begin{aligned}
\left\|\theta_t^{(i)} - \theta_{t-k}^{(i)}\right\| &\leq \sum_{l=t-k}^{t-1} \left\|\theta_{l+1}^{(i)} - \theta_l^{(i)}\right\| \\
&\leq \sum_{l=t-k}^{k-1} \alpha_l \left((1 + \gamma) \left\|\theta_l^{(i)}\right\| + R\right) \\
&\leq k\alpha_{t-k} \left(2(1 + \gamma) \left(\|\bar{\theta}_{t-k}\| + k\alpha_{t-k} R\right) + R\right) \\
&\leq 2k\alpha_{t-k} \left(R + (1 + \gamma)\|\bar{\theta}_{t-k}\|\right),
\end{aligned}$$

where we use the fact that $4k\alpha_{t-k} \leq 1$. Therefore, we have

$$\begin{aligned}
\left\|\theta_t^{(i)} - \theta_{s'K}\right\| &\leq 2(t - s'K)\alpha_{s'K} H \leq 2\alpha_{sK} KH, \\
\left\|\theta_{t-\tau}^{(i)} - \theta_{sK}\right\| &\leq 2(t - \tau - sK)\alpha_{sK} H \leq 2\alpha_{sK} KH.
\end{aligned}$$

Putting all together gives

$$\left\|\theta_t^{(i)} - \theta_{t-\tau}^{(i)}\right\| \leq \alpha_{sK}(C_{\text{prog}}(s'K - sK - 2K) + 4KH) \leq \alpha_{sK} C_{\text{prog}}(\tau).$$

$\square$

With the above corollary, we are ready to bound the gradient progress.

**Corollary I.5.2** (Graident progress). *If $\left\|\bar{\theta}_l\right\| \leq G$ holds for all $l \leq t$, then for any $\theta$, we have*

$$\left\|\bar{g}_{t-\tau}^{(i)}(\theta) - \bar{g}_t^{(i)}(\theta)\right\| \leq L\sigma h(\theta)\alpha_{sK} C_{\text{prog}}(\tau),$$

*where $s$ is the largest integer such that $sK \leq t - \tau$.*

*Proof.*
$$\left\| \bar{g}_{t-\tau}^{(i)}(\theta) - \bar{g}_t^{(i)}(\theta) \right\| = \left\| \left( \bar{A}_{t-\tau}^{(i)} - \bar{A}_t^{(i)} \right) \theta + \bar{b}_{t-\tau}^{(i)} - \bar{b}_t^{(i)} \right\| \le L\sigma \left( R + (1+\gamma)\|\theta\| \right) \left\| \theta_{t-\tau}^{(i)} - \theta_t^{(i)} \right\|,$$
where the inequality uses Lemma G.2. Then we get the desired result by plugging in Corollary I.5.1.
□

The third corollary of Lemma I.5 is the expression of $G$, which was stated as an assumption in previous lemmas.

**Corollary I.5.3** (Parameter bound). *Given the explicit projection $\Pi_{\bar{G}}$, for any $t \in \mathbb{N}$, we have*
$$\left\| \bar{\theta}_t \right\| \le G := \frac{2(2\bar{G} + R)}{1 - 16\alpha_0^2 K^2 \gamma} \le \frac{2(2\bar{G} + R)}{1 - \gamma}.$$

*Proof.* For any $t \in \mathbb{N}$, by (30) in Lemma I.5, we have
$$\left\| \bar{\theta}_t \right\| \le 2 \left( \bar{G} + K\alpha_0(R + 2\alpha_0 C_{\text{drift}}) \right).$$
Plugging the expression of $C_{\text{drift}}$ in Lemma I.4 into the above inequality gives the recursive definition:
$$G = 2(\bar{G} + \alpha_0 K(R + 2\alpha_0 \cdot 2K(R + (1+\gamma)G))).$$
Note that we require $4K\alpha_0 \le 1$ in Lemma I.5. Thus, we have
$$G \le 2\bar{G} + R + 8\alpha_0^2 K^2(1+\gamma)G,$$
which gives
$$G \le \frac{2(2\bar{G} + R)}{1 - 16\alpha_0^2 K^2 \gamma}.$$
Therefore, we let $G := 2(2\bar{G} + R)/(1 - 16\alpha_0^2 K^2 \gamma)$; and then, we have
$$\left\| \bar{\theta}_t \right\| \le 2\bar{G} + R + 8\alpha_0^2 K^2(1+\gamma)G \le G.$$
□

## I.6 MIXING

Unlike stationary MDPs in TD(0) and off-policy Q-learning, the mixing process in our algorithm is a virtual process. After backtracking, we fixed the policy as $\Gamma(\theta_{t-\tau}^{(i)})$, which then introduces a virtual stationary MDP. We denote $\tilde{O}_t^{(i)} = (\tilde{S}_t^{(i)}, \tilde{U}_t^{(i)}, \tilde{S}_{t+1}^{(i)}, \tilde{U}_{t+1}^{(i)})$ the observation of this virtual MDP at time step $t$.

**Lemma I.6** (Mixing). *Let $\mathcal{F}_{t-\tau}$ denote the filtration containing all preceding randomness up to time step $t - \tau$. For any deterministic $\theta$ conditioned on $\mathcal{F}_{t-\tau}$—such as a constant parameter or a parameter determined by $\mathcal{F}_{t-\tau}$—we have*
$$\left\| \mathbb{E}\left[ g_{t-\tau}^{(i)}(\theta, \tilde{O}_t^{(i)}) - \bar{g}_{t-\tau}^{(i)}(\theta) \,\Big|\, \mathcal{F}_{t-\tau} \right] \right\| \le m_i \rho_i^\tau h(\theta)$$

*Proof.* We define a new TD operator:
$$Z_{t-\tau}^{(i)}\left( \theta, \tilde{O}_t^{(i)} \right) := g_{t-\tau}^{(i)}\left( \theta, \tilde{O}_t^{(i)} \right) - \bar{g}_{t-\tau}^{(i)}(\theta).$$
Then, we have
$$\left\| \mathbb{E}\left[ Z_{t-\tau}^{(i)}\left( \theta, \tilde{O}_t^{(i)} \right) \,\Big|\, \mathcal{F}_{t-\tau} \right] \right\|$$
$$= \left\| \mathbb{E}\left[ g_{t-\tau}^{(i)}\left( \theta, \tilde{O}_t^{(i)} \right) \,\Big|\, \mathcal{F}_{t-\tau} \right] - \bar{g}_{t-\tau}^{(i)}(\theta) \right\|$$
$$= \left\| \int_{\mathcal{S}^2 \times \mathcal{A}^2} \phi(s,a)\left( r^{(i)}(s,a) + \gamma\phi^T(s',a')\theta - \phi^T(s,a)\theta \right)\left( P_{t-\tau}^{(i)}\left( \tilde{O}_t^{(i)} = O \,\Big|\, \mathcal{F}_{t-\tau} \right) - \varphi_{t-\tau}^{(i)}(O) \right) dO \right\|$$
$$\le (R + (1+\gamma)\|\theta\|) \cdot \left\| P_{t-\tau}^{(i)}(\tilde{S}_t^{(i)} = \cdot \,\Big|\, \mathcal{F}_{t-\tau}) - \eta_{t-\tau}^{(i)} \right\|_{\text{TV}}$$
$$\le m_i \rho_i^\tau (R + (1+\gamma)\|\theta\|),$$
where the last inequality is by Assumption 1.
□

I.7 BACKTRACKING

**Lemma I.7** (Backtracking). *If $\|\bar{\theta}_l\| \leq G$ for all $l \leq t$, then for any deterministic $\theta$ conditioned on $\mathcal{F}_{t-\tau}$, we have*

$$\left\| \mathbb{E}\left[ g_t^{(i)}(\theta, O_t^{(i)}) - g_{t-\tau}^{(i)}(\theta, \tilde{O}_t^{(i)}) \,\Big|\, \mathcal{F}_{t-\tau} \right] \right\| \leq \alpha_{sK} C_{\text{back}}(\tau) h(\theta),$$

*where*

$$C_{\text{back}}(\tau) = \tau L C_{\text{prog}}(\tau) = O(\tau^2).$$

*Proof.* First, we have

$$\left\| \mathbb{E}\left[ g_t^{(i)}(\theta, O_t^{(i)}) - g_{t-\tau}^{(i)}(\theta, \tilde{O}_t^{(i)}) \,\Big|\, \mathcal{F}_{t-\tau} \right] \right\|$$

$$\leq (R + (1+\gamma)\|\theta\|) \left\| P_{\theta_t^{(i)}}^{(i)}(O_t^{(i)} = \cdot \mid \mathcal{F}_{t-\tau}) - P_{\theta_{t-\tau}^{(i)}}^{(i)}(\tilde{O}_t^{(i)} = \cdot \mid \mathcal{F}_{t-\tau}) \right\|_{\text{TV}}.$$

For a specific client, for notation simplicity, we omit the superscript $(i)$ and denote $P_{\theta_t}$ by $P_t$. Let $O = (s, a, s', a')$; then we have

$$P_t(O_t = O|\mathcal{F}_{t-\tau}) = \int_{\Theta^2} P_t(S_t = s, U_t = a, S_{t+1} = s', U_{t+1} = a', \theta_{t-1} = \theta, \theta_t = \theta'|\mathcal{F}_{t-\tau})\mathrm{d}\theta\mathrm{d}\theta'$$

$$= \int_{\Theta^2} P_t(S_t = s|\mathcal{F}_{t-\tau}) \cdot P_t(\theta_{t-1} = \theta|\mathcal{F}_{t-\tau}, S_t = s)$$

$$\cdot P_t(U_t = a|\mathcal{F}_{t-\tau}, S_t = s, \theta_{t-1} = \theta)$$

$$\cdot P_t(S_{t+1} = s'|\mathcal{F}_{t-\tau}, S_t = s, \theta_{t-1} = \theta, a_t = a)$$

$$\cdot P_t(\theta_t = \theta'|\mathcal{F}_{t-\tau}, S_t = s, \theta_{t-1} = \theta, U_t = a, S_{t+1} = s')$$

$$\cdot P_t(U_{t+1} = a'|\mathcal{F}_{t-\tau}, S_t = s, \theta_{t-1} = \theta, U_t = a, S_{t+1} = s', \theta_t = \theta')\mathrm{d}\theta\mathrm{d}\theta'$$

$$= \int_{\Theta^2} P_t(S_t = s|\theta_{t-\tau}, S_{t-\tau}) \cdot P_t(\theta_{t-1} = \theta|\theta_{t-\tau}, S_{t-\tau}, S_t = s) \cdot \pi_\theta(a|s)$$

$$\cdot P_a(s, s') \cdot P_t(\theta_t = \theta'|\theta_{t-\tau}, S_{t-\tau}, \theta_{t-1} = \theta, S_t = s, U_t = a) \cdot \pi_{\theta'}(a'|s')\mathrm{d}\theta\mathrm{d}\theta',$$

where we use that fact that $U_t$ is dependent on $\theta_{t-1}$ instead of $\theta_t$; and when $\theta_{t-1}$ is determined, $\theta_t$ is not dependent on $S_{t+1}$. Notice that for any $(s, s', a) \in \mathcal{S}^2 \times \mathcal{A}$, we have

$$\int_{\Theta^2} P_t(\theta_{t-1} = \theta|\mathcal{F}_{t-\tau}, S_t = s) \cdot P_t(\theta_t = \theta'|\mathcal{F}_{t-\tau}, \theta_{t-1} = \theta, S_t = s, U_t = a)\mathrm{d}\theta\mathrm{d}\theta' = 1.$$

Thus, for $P_{t-\tau}(\tilde{O}|\mathcal{F}_{t-\tau})$, we have a similar expression:

$$P_{t-\tau}(\tilde{O}_t = O|\mathcal{F}_{t-\tau}) = \int_{\Theta^2} P_{t-\tau}(\tilde{S}_t = s, \tilde{U}_t = a, \tilde{S}_{t+1} = s', \tilde{U}_{t+1} = a'|\mathcal{F}_{t-\tau}) \cdot P_t(\theta_{t-1} = \theta|\mathcal{F}_{t-\tau}, S_t = s)$$

$$\cdot P_t(\theta_t = \theta'|\mathcal{F}_{t-\tau}, \theta_{t-1} = \theta, S_t = s, U_t = a)\mathrm{d}\theta\mathrm{d}\theta'$$

$$= \int_{\Theta^2} P_{t-\tau}(\tilde{S}_t = s|\theta_{t-\tau}, S_{t-\tau}) \cdot \pi_{\theta_{t-\tau}}(a|s) \cdot P_a(s, s') \cdot \pi_{\theta_{t-\tau}}(a'|s')$$

$$\cdot P_t(\theta_{t-1} = \theta|\theta_{t-\tau}, S_{t-\tau}, S_t = s) \cdot P_t(\theta_t = \theta'|\theta_{t-\tau}, S_{t-\tau}, \theta_{t-1} = \theta, S_t = s, U_t = a)\mathrm{d}\theta\mathrm{d}\theta'$$

Therefore, we decompose the observation distribution discrepancy as follows:

$$\left\| P_t(O_t|\mathcal{F}_{t-\tau}) - P_{t-\tau}(\tilde{O}_t|\mathcal{F}_{t-\tau}) \right\|_{\text{TV}} \leq \int_{\mathcal{S}^2 \times \mathcal{A}^2} \left( \underbrace{\left| P_t(O_t = O|\mathcal{F}_{t-\tau}) - Q_t(O) \right|}_{S_1} + \underbrace{\left| Q_t(O) - P_{t-\tau}(\tilde{O}_t = O|\mathcal{F}_{t-\tau}) \right|}_{S_2} \right)\mathrm{d}O,$$

where

$$Q_l(O) := \int_{\Theta^2} P_{t-\tau}(\tilde{S}_l = s|\theta_{t-\tau}, S_{t-\tau}) \cdot \pi_{\theta_{t-\tau}}(a|s) \cdot P_a(s, s') \cdot \pi_{\theta'}(a'|s')$$

$$\cdot P_l(\theta_{l-1} = \theta|\theta_{t-\tau}, S_{t-\tau}, S_l = s) \cdot P_l(\theta_l = \theta'|\theta_{t-\tau}, S_{t-\tau}, \theta_{l-1} = \theta, S_l = s, U_l = a)\mathrm{d}\theta\mathrm{d}\theta'.$$

For $S_1$, we have

$$\int_{\mathcal{S}^2 \times \mathcal{A}^2} \left| P_{t-\tau}(\tilde{O}_t = O | \mathcal{F}_{t-\tau}) - Q_t(O) \right| \mathrm{d}O$$

$$\leq \int_{\mathcal{S}^2 \times \mathcal{A}^2 \times \Theta^2} P_{t-\tau}(\tilde{S}_t = s | \theta_{t-\tau}, S_{t-\tau}) \pi_{\theta_{t-\tau}}(a|s) P_a(s, s') P_t(\theta_{t-1} = \theta | \theta_{t-\tau}, S_{t-\tau}, S_t = s)$$

$$\cdot P_t(\theta_t = \theta' | \theta_{t-\tau}, S_{t-\tau}, \theta_{t-1} = \theta, S_t = s, U_t = a) \left| \pi_{\theta_{t-\tau}}(a'|s') - \pi_{\theta'}(a'|s') \right| \mathrm{d}O \mathrm{d}\theta \mathrm{d}\theta'$$

$$= \int_{\mathcal{S}^2 \times \mathcal{A} \times \Theta^2} P_{t-\tau}(\tilde{S}_t = s | \theta_{t-\tau}, S_{t-\tau}) \pi_{\theta_{t-\tau}}(a|s) P_a(s, s') P_t(\theta_{t-1} = \theta | \theta_{t-\tau}, S_{t-\tau}, S_t = s)$$

$$\cdot P_t(\theta_t = \theta' | \theta_{t-\tau}, S_{t-\tau}, \theta_{t-1} = \theta, S_t = s, U_t = a) \cdot \left\| \pi_{\theta_{t-\tau}}(\cdot|s') - \pi_{\theta'}(\cdot|s') \right\|_{\mathrm{TV}} \mathrm{d}s \mathrm{d}s' \mathrm{d}a \mathrm{d}\theta \mathrm{d}\theta'.$$

By the Lipschitzness of the policy improvement operator (see Assumption 2), we know

$$\sup_{s' \in \mathcal{S}} \left\| \pi_{\theta_{t-\tau}}(\cdot|s') - \pi_{\theta'}(\cdot|s') \right\|_{\mathrm{TV}} \leq L \left\| \theta_{t-\tau} - \theta' \right\|.$$

Then for any $\theta' \in \Theta$ for which $P_t(\theta_t = \cdot | \mathcal{F}_{t-\tau})$ has non-zero density, meaning that $\theta'$ is reachable at time step $t$, Corollary I.5.1 implies

$$\left\| \theta_{t-\tau}^{(i)} - \theta' \right\| \leq \alpha_{sK} C_{\mathrm{prog}}(\tau),$$

where $s$ is the largest integer such that $sK \leq t - \tau$.

Therefore, we have

$$\int_{\mathcal{S}^2 \times \mathcal{A}^2} \left| P_{t-\tau}(\tilde{O}_t = O | \mathcal{F}_{t-\tau}) - Q_t(O) \right| \mathrm{d}O \leq \alpha_{sK} L C_{\mathrm{prog}}(\tau).$$

For $S_2$, we have

$$\int_{\mathcal{S}^2 \times \mathcal{A}^2} |P_t(O_t = O | \mathcal{F}_{t-\tau}) - Q_t(O)| \mathrm{d}O$$

$$\leq \int_{\mathcal{S}^2 \times \mathcal{A}^2 \times \Theta^2} P_t(\theta_{t-1} = \theta | \theta_{t-\tau}, S_{t-\tau}, S_t = s) P_t(\theta_t = \theta' | \theta_{t-\tau}, S_{t-\tau}, S_t = s, U_t = a, \theta_{t-1} = \theta) \pi_{\theta'}(a'|s') P_a(s, s')$$

$$\cdot \left| P_{t-\tau}(\tilde{S}_t = s | \theta_{t-\tau}, S_{t-\tau}) \pi_{\theta_{t-\tau}}(a|s) - P_t(S_t = s | \theta_{t-\tau}, S_{t-\tau}) \pi_\theta(a|s) \right| \mathrm{d}O \mathrm{d}\theta \mathrm{d}\theta'$$

$$= \int_{\mathcal{S} \times \mathcal{A} \times \Theta} P_t(\theta_{t-1} = \theta | \theta_{t-\tau}, S_{t-\tau}, S_t = s)$$

$$\cdot \left| P_{t-\tau}(\tilde{S}_t = s | \theta_{t-\tau}, S_{t-\tau}) \pi_{\theta_{t-\tau}}(a|s) - P_t(S_t = s | \theta_{t-\tau}, S_{t-\tau}) \pi_\theta(a|s) \right| \mathrm{d}s \mathrm{d}a \mathrm{d}\theta$$

$$\leq \int_{\mathcal{S} \times \mathcal{A} \times \Theta} P_t(\theta_{t-1} = \theta | \theta_{t-\tau}, S_{t-\tau}, S_t = s)$$

$$\cdot \left( \left| P_{t-\tau}(\tilde{S}_t = s | \theta_{t-\tau}, S_{t-\tau}) \pi_{\theta_{t-\tau}}(a|s) - P_{t-\tau}(\tilde{S}_t = s | \theta_{t-\tau}, S_{t-\tau}) \pi_\theta(a|s) \right| \right.$$

$$\left. + \left| P_{t-\tau}(\tilde{S}_t = s | \theta_{t-\tau}, S_{t-\tau}) \pi_\theta(a|s) - P_t(S_t = s | \theta_{t-\tau}, S_{t-\tau}) \pi_\theta(a|s) \right| \right) \mathrm{d}s \mathrm{d}a \mathrm{d}\theta$$

$$\leq \sup_{s \in \mathcal{S}} \left\| \pi_{t-\tau}(\cdot|s) - \pi_\theta(\cdot|s) \right\|_{\mathrm{TV}} + \left\| P_{t-\tau}(\tilde{S}_t = \cdot | \mathcal{F}_{t-\tau}) - P_t(S_t = \cdot | \mathcal{F}_{t-\tau}) \right\|_{\mathrm{TV}}$$

$$\leq \alpha_{sK} L C_{\mathrm{prog}}(\tau - 1) + \left\| P_{t-\tau}(\tilde{S}_t = \cdot | \mathcal{F}_{t-\tau}) - P_t(S_t = \cdot | \mathcal{F}_{t-\tau}) \right\|_{\mathrm{TV}}.$$

Substituting $S_1$ and $S_2$ with the above bounds gives

$$\left\| P_t(O_t | \mathcal{F}_{t-\tau}) - P_{t-\tau}(\tilde{O}_t | \mathcal{F}_{t-\tau}) \right\|_{\mathrm{TV}} \leq \left\| P_{t-\tau}(\tilde{S}_t = \cdot | \mathcal{F}_{t-\tau}) - P_t(S_t = \cdot | \mathcal{F}_{t-\tau}) \right\|_{\mathrm{TV}} + \alpha_{sK} L \sum_{l=\tau-1}^{\tau} C_{\mathrm{prog}}(l). \tag{33}$$

Applying a similar decomposition as $S_1$ and $S_2$, we can obtain an analogous bound to (33) for the state distribution discrepancy:

$$\left\| P_{t-\tau}(\tilde{S}_t = \cdot | \mathcal{F}_{t-\tau}) - P_t(S_t = \cdot | \mathcal{F}_{t-\tau}) \right\|_{\text{TV}}$$

$$\leq \left\| P_{t-\tau}(\tilde{S}_{t-1} = \cdot | \mathcal{F}_{t-\tau}) - P_{t-1}(S_{t-1} = \cdot | \mathcal{F}_{t-\tau}) \right\|_{\text{TV}} + \alpha_{sK} L C_{\text{prog}}(\tau - 2)$$

$$\leq \left\| P_{t-\tau}(\tilde{S}_{t-\tau} = \cdot | \mathcal{F}_{t-\tau}) - P_{t-\tau}(S_{t-\tau} = \cdot | \mathcal{F}_{t-\tau}) \right\|_{\text{TV}} + \sum_{l=1}^{\tau-2} \alpha_{sK} L C_{\text{prog}}(l)$$

$$\leq (\tau - 2) \alpha_{sK} L C_{\text{prog}}(\tau).$$

Putting this bound back into (33) gives

$$\left\| P_t(O_t | \mathcal{F}_{t-\tau}) - P_{t-\tau}(\tilde{O}_t | \mathcal{F}_{t-\tau}) \right\|_{\text{TV}} \leq \tau a_{sK} L C_{\text{prog}}(\tau).$$

Finally, we get

$$\left\| \mathbb{E} \left[ g_t^{(i)}(\theta, O_t^{(i)}) - g_{t-\tau}^{(i)}(\theta, \tilde{O}_t^{(i)}) \,\Big|\, \mathcal{F}_{t-\tau} \right] \right\| \leq \tau \alpha_{sK} L C_{\text{prog}}(\tau) \left( R + (1 + \gamma) \|\theta\| \right).$$

$\square$

## I.8 GRADIENT VARIANCE

**Lemma I.8** (Gradient variance).

$$\mathbb{E} \|g_t(\boldsymbol{\theta}_t)\|^2 \leq 64 \left( \mathbb{E} \|\bar{\theta}_t - \theta_*\|^2 + \frac{\Lambda^2(\epsilon_p, \epsilon_r)}{w^2} \right) + \alpha_{sK}^2 C_{\text{var}}(\tau) + 4m^2 \rho^{2\tau} H^2 + \frac{32H^2}{N},$$

*where*

$$C_{\text{var}}(\tau) = 4 \left( 4(3 + H^2 L^2 \sigma^2) C_{\text{drift}}^2 + 4H^2 L^2 \sigma^2 C_{\text{prog}}^2(\tau) + H^2 C_{\text{back}}^2(\tau) \right).$$

*Proof.* Similar to Lemma I.1, we first decompose the gradient variance and establish the linear speedups for the backtracking and mixing terms.

$$\|\boldsymbol{g}_t(\boldsymbol{\theta}_t)\|^2 = \|\boldsymbol{g}_t(\boldsymbol{\theta}_t) - \bar{\boldsymbol{g}}(\boldsymbol{\theta}_*)\|^2 \tag{34}$$

$$= \Big\| \boldsymbol{g}_t(\boldsymbol{\theta}_t) - \boldsymbol{g}_t(\boldsymbol{\theta}_*, \boldsymbol{O}_t) + \boldsymbol{g}_t(\boldsymbol{\theta}_*, \boldsymbol{O}_t) - \boldsymbol{g}_{t-\tau}(\boldsymbol{\theta}_*, \tilde{\boldsymbol{O}}_t)$$

$$+ \boldsymbol{g}_{t-\tau}(\boldsymbol{\theta}_*, \tilde{\boldsymbol{O}}_t) - \bar{\boldsymbol{g}}_{t-\tau}(\boldsymbol{\theta}_*) + \bar{\boldsymbol{g}}_{t-\tau}(\boldsymbol{\theta}_*) - \bar{\boldsymbol{g}}(\boldsymbol{\theta}_*) \Big\|^2 \tag{35}$$

$$\leq \frac{4}{N} \sum_{i=1}^{N} \left( \underbrace{\left\| g_t^{(i)}\left(\theta_t^{(i)}\right) - g_t^{(i)}\left(\theta_*^{(i)}\right) \right\|^2}_{G_1} + \underbrace{\left\| \bar{g}_{t-\tau}^{(i)}\left(\theta_*^{(i)}\right) - \bar{g}\left(\theta_*^{(i)}\right) \right\|^2}_{G_2, \text{ gradient progress}} \right)$$

$$+ 4 \underbrace{\left\| \boldsymbol{g}_t\left(\boldsymbol{\theta}_*, \boldsymbol{O}_t\right) - \boldsymbol{g}_{t-\tau}\left(\boldsymbol{\theta}_*, \tilde{\boldsymbol{O}}_t\right) \right\|^2}_{G_3, \text{ backtracking}} + 4 \underbrace{\left\| \boldsymbol{g}_{t-\tau}\left(\boldsymbol{\theta}_*, \tilde{\boldsymbol{O}}_t\right) - \bar{\boldsymbol{g}}_{t-\tau}(\boldsymbol{\theta}_*) \right\|^2}_{G_4, \text{ mixing}},$$

where (34) uses the fact that $\bar{\boldsymbol{g}}(\boldsymbol{\theta}_*) = \frac{1}{N} \sum_{i=1}^{N} \bar{g}^{(i)}\left(\theta_*^{(i)}\right) = 0$; and in (35), we denote $\boldsymbol{g}_{t-\tau}(\boldsymbol{\theta}_*, \tilde{\boldsymbol{O}}_t) = \frac{1}{N} \sum_{i=1}^{N} g_{t-\tau}^{(i)}\left(\theta_*^{(i)}, \tilde{O}_t^{(i)}\right)$, and the same notation applies to other semi-gradients.

By the Lipschitzness of semi-gradient $g_t^{(i)}$, $G_1$ is bounded by

$$\left\| g_t^{(i)}\left(\theta_t^{(i)}\right) - g_t^{(i)}\left(\theta_*^{(i)}\right) \right\|^2 \leq 4 \left\| \theta_t^{(i)} - \theta_*^{(i)} \right\|^2$$

$$\leq 12 \left( \left\| \theta_t^{(i)} - \bar{\theta}_t \right\|^2 + \left\| \bar{\theta}_t - \theta_* \right\|^2 + \left\| \theta_*^{(i)} - \theta_* \right\|^2 \right).$$

By Lemma G.1, $G_2$ is bounded by

$$\left\| \bar{g}_{t-\tau}^{(i)}\left(\theta_*^{(i)}\right) - \bar{g}\left(\theta_*^{(i)}\right) \right\|^2 \le \left(\left(R + (1+\gamma)\left\|\theta_*^{(i)}\right\|\right)\left\|\mu_{t-\tau}^{(i)} - \mu_*^{(i)}\right\|_{\text{TV}}\right)^2$$
$$\le H^2 L^2 \sigma^2 \left\|\theta_{t-\tau}^{(i)} - \theta_*^{(i)}\right\|^2$$
$$\le 4 H^2 L^2 \sigma^2 \left(\left\|\theta_{t-\tau}^{(i)} - \bar{\theta}_{t-\tau}\right\|^2 + \left\|\bar{\theta}_{t-\tau} - \bar{\theta}_t\right\|^2 + \left\|\bar{\theta}_t - \theta_*\right\|^2 + \left\|\theta_* - \theta_*^{(i)}\right\|^2\right).$$

Now we are left with $G_3$ and $G_4$. However, we only have the bound of their first moment by Lemma I.7 and I.6. We first note that for a set of functions $\{g_i\}_{i=1}^N$ such that $\|g_i\|_\infty \le a$ and independent random variables $\{x_i\}_{i=1}^N$ such that $\|\mathbb{E}g_i(x_i)\| \le b$, we have

$$\mathbb{E}\|\boldsymbol{g}(\boldsymbol{x})\|^2 = \mathbb{E}\left\langle \frac{1}{N}\sum_{i=1}^N g_i(x_i), \frac{1}{N}\sum_{i=1}^N g_i(x_i)\right\rangle$$
$$= \frac{1}{N^2}\sum_{i=1}^N \mathbb{E}\|g_i(x_i)\|^2 + \frac{1}{N^2}\sum_{i\neq j}\langle \mathbb{E}g_i(x_i), \mathbb{E}g_j(x_j)\rangle$$
$$\le \frac{a^2}{N} + \frac{1}{N^2}\sum_{i=1}^N\sum_{j=1}^N \|\mathbb{E}g_i(x_i)\|\|\mathbb{E}g_j(x_j)\|$$
$$\le \frac{a^2}{N} + b^2. \tag{36}$$

By (36) and Lemma I.7, the expectation of $G_3$ is bounded by

$$\mathbb{E}\left[\left\|\boldsymbol{g}_t\left(\boldsymbol{\theta}_*, \boldsymbol{O}_t\right) - \boldsymbol{g}_{t-\tau}\left(\boldsymbol{\theta}_*, \tilde{\boldsymbol{O}}_t\right)\right\|^2 \,\middle|\, \mathcal{F}_{t-\tau}\right] \le \frac{4H^2}{N} + \alpha_{sK}^2 C_{\text{back}}^2 H^2. \tag{37}$$

By (36) and Lemma I.6, the expectation of $G_4$ is bounded by

$$\mathbb{E}\left[\left\|\boldsymbol{g}_{t-\tau}\left(\boldsymbol{\theta}_*, \tilde{\boldsymbol{O}}_t\right) - \bar{\boldsymbol{g}}_{t-\tau}(\boldsymbol{\theta}_*)\right\|^2 \,\middle|\, \mathcal{F}_{t-\tau}\right] = \mathbb{E}\left[\left\|\boldsymbol{Z}_{t-\tau}(\boldsymbol{\theta}_*, \tilde{\boldsymbol{O}}_t)\right\|^2 \,\middle|\, \mathcal{F}_{t-\tau}\right] \le \frac{4H^2}{N} + m^2\rho^{2\tau}H^2.$$

Combining all together with Lemma I.4, I.5, and Theorem 1, we get

$$\mathbb{E}\left[\|\boldsymbol{g}_t(\boldsymbol{\theta}_t)\|^2|\mathcal{F}_{t-\tau}\right] \le 4\left(4(3 + H^2 L^2\sigma^2)\left(\mathbb{E}\left[\left\|\bar{\theta}_t - \theta_*\right\|^2|\mathcal{F}_{t-\tau}\right] + \alpha_{sK}^2 C_{\text{drift}}^2 + \frac{\Lambda^2(\epsilon_p, \epsilon_r)}{w^2}\right)\right.$$
$$\left. + 4H^2 L^2\sigma^2 \alpha_{sK}^2 C_{\text{prog}}^2 + \frac{8H^2}{N} + \left(\alpha_{sK}^2 C_{\text{back}}^2 + m^2\rho^{2\tau}\right)H^2\right)$$
$$\le 64\left(\mathbb{E}\left[\left\|\bar{\theta}_t - \theta_*\right\|^2|\mathcal{F}_{t-\tau}\right] + \frac{\Lambda^2(\epsilon_p, \epsilon_r)}{w^2}\right) + \alpha_{sK}^2 C_{\text{var}} + 4m^2\rho^{2\tau}H^2 + \frac{32H^2}{N},$$

where we use the fact that $LH\sigma \le w \le 1$ required by (20), and

$$C_{\text{var}} = 4\left(4(3 + H^2 L^2\sigma^2)C_{\text{drift}}^2 + 4H^2 L^2\sigma^2 C_{\text{prog}}^2 + H^2 C_{\text{back}}^2\right).$$

Finally, we get

$$\mathbb{E}\|\boldsymbol{g}_t(\boldsymbol{\theta}_t)\|^2 = \mathbb{E}\left[\mathbb{E}\left[\|\boldsymbol{g}_t(\boldsymbol{\theta}_t)\|^2|\mathcal{F}_{t-\tau}\right]\right] \le 64\left(\mathbb{E}\left\|\bar{\theta}_t - \theta_*\right\|^2 + \frac{\Lambda^2(\epsilon_p, \epsilon_r)}{w^2}\right) + \alpha_{sK}^2 C_{\text{var}} + 4m^2\rho^{2\tau}H^2 + \frac{32H^2}{N}.$$

$$\square$$

Recall that Lemma I.5 bounds the central parameter progress, which gives a *naive* bound of the mean square central parameter progress $\mathbb{E}\|\bar{\theta}_t - \bar{\theta}_{t-\tau}\|^2 \le \alpha_{sK}^2 C_{\text{prog}}^2(\tau)$ for any $\tau \le t$, where $s$ is the largest integer such that $sK \le t - \tau$. However, with the help of Lemma I.8, we can derive a tighter bound of the mean square central parameter progress, which is essential for proving Theorem 2 later.

**Corollary I.8.1** (Mean square central parameter progress). *For any $\tau \leq t$, we have*

$$\mathbb{E}\left\|\bar{\theta}_t - \bar{\theta}_{t-\tau}\right\|^2 \leq 4(\tau+K)(\tau+3K)\alpha_{sK}^2 \left(64\mathbb{E}\left\|\bar{\theta}_{sK} - \theta_*\right\|^2 + V(\tau)\right),$$

*where $s$ is the largest integer such that $sK \leq t - \tau$ and*

$$V(\tau) := \frac{64\Lambda^2(\epsilon_p, \epsilon_r)}{w^2} + \alpha_{sK}^2 C_{\text{var}}(\tau) + 4m^2\rho^{2\tau}H^2 + \frac{32H^2}{N}$$

*is part of the gradient variance bound in Lemma I.8.*

*Proof.* Recall in Lemma I.5, we utilize a *naive* bound of $\|\boldsymbol{g}_t(\boldsymbol{\theta}_t)\|$ by $(R + (1+\gamma)\|\bar{\theta}_t\| + 2\omega_l)$; the key difference in this proof is that we will bound $\mathbb{E}\|\boldsymbol{g}_t(\boldsymbol{\theta}_t)\|^2$ using Lemma I.8. Therefore, similar to (32), let $s$ and $s'$ be the largest integer such that $sK \leq t - \tau$ and $s'K \leq t$ respectively. Then we have

$$\mathbb{E}\|\bar{\theta}_t - \bar{\theta}_{t-\tau}\|^2 \leq (s' - s + 2)\left(\mathbb{E}\|\bar{\theta}_t - \bar{\theta}_{s'K}\|^2 + \sum_{j=1}^{s'-s}\mathbb{E}\|\bar{\theta}_{(s+j)K} - \bar{\theta}_{(s+j-1)K}\|^2 + \mathbb{E}\|\bar{\theta}_{t-\tau} - \bar{\theta}_{sK}\|^2\right)$$

$$\leq (s' - s + 2)\left(\mathbb{E}\|\breve{\theta}_t - \bar{\theta}_{s'K}\|^2 + \sum_{j=1}^{s'-s}\mathbb{E}\|\breve{\theta}_{(s+j)K} - \bar{\theta}_{(s+j-1)K}\|^2 + \mathbb{E}\|\breve{\theta}_{t-\tau} - \bar{\theta}_{sK}\|^2\right)$$

$$\leq 2(s' - s + 2)K \sum_{l=sK}^{t-1} \alpha_l^2 \mathbb{E}\|\boldsymbol{g}_l(\boldsymbol{\theta}_l)\|^2$$

$$\leq 2(\tau + 3K)\alpha_{sK}^2 \sum_{l=sK}^{t-1} \mathbb{E}\|\boldsymbol{g}_l(\boldsymbol{\theta}_l)\|^2.$$

By Lemma I.8, we get

$$\mathbb{E}\|\bar{\theta}_t - \bar{\theta}_{t-\tau}\|^2 \leq 2(\tau + 3K)\alpha_{sK}^2 \sum_{l=sK}^{t-1}\left(64\mathbb{E}\|\bar{\theta}_l - \theta_*\|^2 + V(l - sK)\right). \tag{38}$$

Then, similar to (30), we want to bound $\mathbb{E}\|\bar{\theta}_l - \theta_*\|^2$ by $\mathbb{E}\|\bar{\theta}_{sK} - \theta_*\|^2$ for $sK < l \leq t - 1$. We have

$$\mathbb{E}\left\|\bar{\theta}_l - \theta_*\right\|^2 \leq \mathbb{E}\left\|\breve{\theta}_l - \theta_*\right\|^2$$

$$= \mathbb{E}\|\bar{\theta}_{l-1} - \theta_* + \alpha_{l-1}\boldsymbol{g}_{l-1}(\boldsymbol{\theta}_{l-1})\|^2$$

$$= \mathbb{E}\|\bar{\theta}_{l-1} - \theta_*\|^2 + 2\alpha_{l-1}\mathbb{E}\langle\bar{\theta}_{l-1} - \theta_*, \boldsymbol{g}_{l-1}(\boldsymbol{\theta}_{l-1})\rangle + \alpha_{l-1}^2\mathbb{E}\|\boldsymbol{g}_{l-1}(\boldsymbol{\theta}_{l-1})\|^2$$

$$\leq (1 + \alpha_{l-1})\mathbb{E}\left\|\bar{\theta}_{l-1} - \theta_*\right\|^2 + \alpha_{l-1}(1 + \alpha_{l-1})\mathbb{E}\left\|\boldsymbol{g}_{l-1}(\boldsymbol{\theta}_{l-1})\right\|^2 \tag{39}$$

$$\leq (1 + \alpha_{l-1})(1 + 64\alpha_{l-1})\mathbb{E}\|\bar{\theta}_{l-1} - \theta_*\|^2 + \alpha_{l-1}(1 + \alpha_{l-1})V(l - 1 - sK), \tag{40}$$

where (39) uses Young's inequality and (40) uses Lemma I.8. We require $64\alpha_{sK} \leq 1$, which gives $(1 + \alpha_{l-1})(1 + 64\alpha_{l-1}) \leq (1 + 66\alpha_{l-1})$. Recursively applying (40) gives

$$\mathbb{E}\left\|\bar{\theta}_l - \theta_*\right\|^2 \leq (1 + 66\alpha_{sK})^{l-sK}\mathbb{E}\|\bar{\theta}_{sK} - \theta_*\|^2 + \alpha_{sK}(1 + \alpha_{sK})V(\tau)\sum_{j=0}^{l-1-sK}(1 + 66\alpha_{sK})^j,$$

where we use the fact that $V$ is monotonically increasing. We further requires that $132(\tau + K)\alpha_{sK} \leq 1$. Then, similar to (25), we get

$$\mathbb{E}\left\|\bar{\theta}_l - \theta_*\right\|^2 \leq 2\mathbb{E}\|\bar{\theta}_{sK} - \theta_*\|^2 + 2\alpha_{sK}(1 + \alpha_{sK})(\tau + K)V(\tau). \tag{41}$$

Combining (38) and (41) gives

$$\mathbb{E}\|\bar{\theta}_t - \bar{\theta}_{t-\tau}\|^2 \leq 2(\tau + 3K)\alpha_{sK}^2 \sum_{l=sK}^{t-1} \left(128\mathbb{E}\|\bar{\theta}_{sK} - \theta_*\|^2 + 128\alpha_{sK}(1 + \alpha_{sK})(\tau + K)V(\tau) + V(\tau)\right)$$

$$\leq 2(\tau + 3K)\alpha_{sK}^2 \sum_{l=sK}^{t-1} \left(128\mathbb{E}\|\bar{\theta}_{sK} - \theta_*\|^2 + \left(\frac{128}{132} \cdot \frac{133}{132} + 1\right)V(\tau)\right) \quad (42)$$

$$\leq 4(\tau + K)(\tau + 3K)\alpha_{sK}^2 \left(64\mathbb{E}\|\bar{\theta}_{sK} - \theta_*\|^2 + V(\tau)\right),$$

where (42) uses our requirement that $\alpha_{t-\tau} \leq (\tau + K)\alpha_{sK} \leq 1/132$. □

## J  PROOF OF THEOREM 2

**Theorem 2.** *If $\left\|\bar{\theta}_l\right\| \leq G$ holds for all $l \leq t$, then*

$$\mathbb{E}\left\|\bar{\theta}_{t+1} - \theta_*\right\|^2 \leq (1 - \alpha_t w)\mathbb{E}\left\|\bar{\theta}_t - \theta_*\right\|^2 + \alpha_t C_1 \Lambda^2(\epsilon_p, \epsilon_r) + \alpha_t^2 \frac{C_2}{N} + \alpha_t^3 C_3 + \alpha_t^4 C_4,$$

*where $C_1, C_2, C_3, C_4$ are constants defined in (50).*

*Proof.* We need to pre-process the results from Lemmas I.3 to I.7 before plugging them back into Lemma I.1. Throughout this proof, let $s$ and $s'$ be the largest integers such that $sK \leq t - \tau$ and $s'K \leq t$. First, for Lemma I.3, by Young's inequality $ab \leq \frac{1}{2}\left(\beta a^2 + \frac{1}{\beta}b^2\right)$, for any positive $\beta$, we have

$$2\mathbb{E}\left\langle \bar{\theta}_t - \theta_*, \frac{1}{N}\sum_{i=1}^N \bar{g}^{(i)}\left(\bar{\theta}_t\right) - \bar{g}\left(\bar{\theta}_t\right)\right\rangle \leq \beta\mathbb{E}\left\|\bar{\theta}_t - \theta_*\right\|^2 + \frac{\Lambda^2(\epsilon_p, \epsilon_r)}{\beta}. \quad (43)$$

Similarly, for Lemma I.4 and I.5.2, we have

$$2\mathbb{E}\left\langle \bar{\theta}_t - \theta_*, \frac{1}{N}\sum_{i=1}^N \left(\bar{g}^{(i)}(\theta_t^{(i)}) - \bar{g}^{(i)}(\bar{\theta}_t)\right)\right\rangle \leq \beta\mathbb{E}\left\|\bar{\theta}_t - \theta_*\right\|^2 + \frac{1}{\beta}\alpha_{s'K}^2(1 + \gamma + \sigma LH)^2 C_{\mathrm{drift}}^2, \quad (44)$$

$$\frac{1}{N}\sum_{i=1}^N 2\mathbb{E}\left\langle \bar{\theta}_t - \theta_*, \bar{g}_{t-\tau}^{(i)}(\theta_t^{(i)}) - \bar{g}^{(i)}(\theta_t^{(i)})\right\rangle \leq \beta\mathbb{E}\left\|\bar{\theta}_t - \theta_*\right\|^2 + \frac{1}{\beta}\alpha_{sK}^2 C_{\mathrm{prog}}^2 L^2\sigma^2\mathbb{E}h^2\left(\boldsymbol{\theta}_t\right), \quad (45)$$

where $h^2(\boldsymbol{\theta}_t) = \frac{1}{N}\sum_{i=1}^N h^2\left(\theta_t^{(i)}\right)$, and

$$\mathbb{E}h^2\left(\boldsymbol{\theta}_t\right) = 2H^2 + 2(1 + \gamma)^2\mathbb{E}\left[\Omega_t\right] \leq 2H^2 + 8\alpha_{s'K}^2 C_{\mathrm{drift}}^2 \leq H_{\mathrm{drift}}^2,$$

where we define $H_{\mathrm{drift}} := \sqrt{2H^2 + 8\alpha_0^2 C_{\mathrm{drift}}^2}$.

Then, for Lemma I.6, we have

$$\frac{1}{N}\sum_{i=1}^N \mathbb{E}\left\langle \bar{\theta}_t - \theta_*, g_{t-\tau}^{(i)}\left(\theta_t^{(i)}\right) - \bar{g}_{t-\tau}^{(i)}\left(\theta_t^{(i)}\right)\right\rangle$$

$$= \frac{1}{N}\sum_{i=1}^N \mathbb{E}\left\langle \bar{\theta}_t - \bar{\theta}_{t-\tau}, g_{t-\tau}^{(i)}\left(\theta_t^{(i)}\right) - \bar{g}_{t-\tau}^{(i)}\left(\theta_t^{(i)}\right)\right\rangle + \frac{1}{N}\sum_{i=1}^N \mathbb{E}\left\langle \bar{\theta}_{t-\tau} - \theta_*, g_{t-\tau}^{(i)}\left(\theta_t^{(i)}\right) - \bar{g}_{t-\tau}^{(i)}\left(\theta_t^{(i)}\right)\right\rangle$$

$$\leq \underbrace{\mathbb{E}\left[\frac{1}{N}\sum_{i=1}^N \mathbb{E}\left[\left\langle \bar{\theta}_t - \bar{\theta}_{t-\tau}, g_{t-\tau}^{(i)}\left(\theta_t^{(i)}\right) - \bar{g}_{t-\tau}^{(i)}\left(\theta_t^{(i)}\right)\right\rangle \Big| \mathcal{F}_{t-\tau}\right]\right]}_{H_1}$$

$$+ \underbrace{\mathbb{E}\left[\frac{1}{N}\sum_{i=1}^N \left\|\bar{\theta}_{t-\tau} - \theta_*\right\| \left\|\mathbb{E}\left[g_{t-\tau}^{(i)}\left(\theta_t^{(i)}\right) - \bar{g}_{t-\tau}^{(i)}\left(\theta_t^{(i)}\right) \Big| \mathcal{F}_{t-\tau}\right]\right\|\right]}_{H_2}.$$

For $H_1$, since both $g_{t-\tau}^{(i)}$ and $\bar{g}_{t-\tau}^{(i)}$ are independent of $\theta_t^{(i)}$ conditioned on $\mathcal{F}_{t-\tau}$, Lemma I.5 and I.6 give

$$
\begin{aligned}
H_1 =& \mathbb{E}\left[\frac{1}{N}\sum_{i=1}^{N}\left\langle \mathbb{E}[\bar{\theta}_t - \bar{\theta}_{t-\tau} \mid \mathcal{F}_{t-\tau}], \mathbb{E}\left[g_{t-\tau}^{(i)}\left(\theta_t^{(i)}\right) - \bar{g}_{t-\tau}^{(i)}\left(\theta_t^{(i)}\right) \,\Big|\, \mathcal{F}_{t-\tau}\right]\right\rangle\right] \\
\leq& \mathbb{E}\left[\frac{1}{N}\sum_{i=1}^{N}\left\| \mathbb{E}[\bar{\theta}_t - \bar{\theta}_{t-\tau} \mid \mathcal{F}_{t-\tau}]\right\| \left\| \mathbb{E}\left[Z_{t-\tau}^{(i)}\left(\theta_t^{(i)}\right) \,\Big|\, \mathcal{F}_{t-\tau}\right]\right\|\right] \\
\leq& \alpha_{sK} C_{\text{prog}} \cdot m\rho^{\tau}\mathbb{E}h(\boldsymbol{\theta}_t) \\
\leq& \alpha_{sK} m\rho^{\tau} C_{\text{prog}} H_{\text{drift}}.
\end{aligned}
\tag{46}
$$

Similarly, for $H_2$, we have

$$
\begin{aligned}
H_2 =& \mathbb{E}\left[\|\bar{\theta}_{t-\tau} - \theta_*\| \frac{1}{N}\sum_{i=1}^{N}\left\| \mathbb{E}\left[g_{t-\tau}^{(i)}\left(\theta_t^{(i)}\right) - \bar{g}_{t-\tau}^{(i)}\left(\theta_t^{(i)}\right) \,\Big|\, \mathcal{F}_{t-\tau}\right]\right\|\right] \\
\leq& \mathbb{E}\left[m\rho^{\tau}\mathbb{E}h\left(\boldsymbol{\theta}_t\right)\left(\|\bar{\theta}_{t-\tau} - \bar{\theta}_t\| + \|\bar{\theta}_t - \theta_*\|\right)\right] \\
\leq& m\rho^{\tau} H_{\text{drift}}\left(\mathbb{E}\|\bar{\theta}_t - \theta_*\| + \alpha_{sK} C_{\text{prog}}\right) \\
\leq& \frac{1}{2}\left(\beta\mathbb{E}\|\bar{\theta}_t - \theta_*\|^2 + \frac{1}{\beta}m^2\rho^{2\tau}H_{\text{drift}}^2\right) + \alpha_{sK}m\rho^{\tau}C_{\text{prog}}H_{\text{drift}}.
\end{aligned}
$$

Substituting $H_1$ and $H_2$ with the above bounds gives

$$
\frac{1}{N}\sum_{i=1}^{N}2\mathbb{E}\left\langle\bar{\theta}_t - \theta_*, g_{t-\tau}^{(i)}\left(\theta_t^{(i)}\right) - \bar{g}_{t-\tau}^{(i)}\left(\theta_t^{(i)}\right)\right\rangle \leq \beta\mathbb{E}\|\bar{\theta}_t - \theta_*\|^2 + m\rho^{\tau}H_{\text{drift}}\left(\frac{1}{\beta}m\rho^{\tau}H_{\text{drift}} + 4\alpha_{sK}C_{\text{prog}}\right).
\tag{47}
$$

For Lemma I.7, the trick we applied in (46) is no longer valid because $g_t^{(i)}$ and $\theta_t^{(i)}$ are correlated. Notice that $\theta_{t-\tau}^{(i)}$ is deterministic given $\mathcal{F}_{t-\tau}$, we first apply the following decomposition:

$$
\begin{aligned}
&\frac{1}{N}\sum_{i=1}^{N}2\mathbb{E}\left\langle\bar{\theta}_t - \theta_*, g_t^{(i)}(\theta_t^{(i)}, O_t^{(i)}) - g_{t-\tau}^{(i)}(\theta_t^{(i)}, \tilde{O}_t^{(i)})\right\rangle \\
=& \underbrace{\frac{1}{N}\sum_{i=1}^{N}2\mathbb{E}\left\langle\bar{\theta}_t - \theta_*, \left(g_t^{(i)}(\theta_t^{(i)}, O_t^{(i)}) - g_t^{(i)}(\theta_{t-\tau}^{(i)}, O_t^{(i)})\right) + \left(g_{t-\tau}^{(i)}(\theta_{t-\tau}^{(i)}, \tilde{O}_t^{(i)}) - g_{t-\tau}^{(i)}(\theta_t^{(i)}, \tilde{O}_t^{(i)})\right)\right\rangle}_{H_3} \\
&+ \underbrace{\frac{1}{N}\sum_{i=1}^{N}2\mathbb{E}\left\langle\bar{\theta}_{t-\tau} - \theta_*, g_t^{(i)}(\theta_{t-\tau}^{(i)}, O_t^{(i)}) - g_{t-\tau}^{(i)}(\theta_{t-\tau}^{(i)}, \tilde{O}_t^{(i)})\right\rangle}_{H_4} \\
&+ \underbrace{\frac{1}{N}\sum_{i=1}^{N}2\mathbb{E}\left\langle\bar{\theta}_t - \bar{\theta}_{t-\tau}, g_t^{(i)}(\theta_{t-\tau}^{(i)}, O_t^{(i)}) - g_{t-\tau}^{(i)}(\theta_{t-\tau}^{(i)}, \tilde{O}_t^{(i)})\right\rangle}_{H_5}.
\end{aligned}
$$

By the Lipschitzness of semi-gradient $g_t^{(i)}$ and $g_{t-\tau}^{(i)}$ and Corollary I.5.1, we have

$$
H_3 \leq \frac{1}{N}\sum_{i=1}^{N}2\mathbb{E}\left[\|\bar{\theta}_t - \theta_*\| \cdot 4\left\|\theta_t^{(i)} - \theta_{t-\tau}^{(i)}\right\|\right] \leq \beta\mathbb{E}\|\bar{\theta}_t - \theta_*\|^2 + \frac{4}{\beta}\alpha_{sK}^2 C_{\text{prog}}^2(\tau).
$$

By Lemma I.7, we have

$$
\begin{aligned}
H_4 \leq& \frac{2}{N}\sum_{i=1}^{N}\mathbb{E}\left[\|\bar{\theta}_{t-\tau} - \theta_*\|\mathbb{E}\left[\left\|g_t^{(i)}(\theta_{t-\tau}^{(i)}, O_t^{(i)}) - g_{t-\tau}^{(i)}(\theta_{t-\tau}^{(i)}, \tilde{O}_t^{(i)})\right\| \,\Big|\, \mathcal{F}_{t-\tau}\right]\right] \\
\leq& \beta\mathbb{E}\left\|\bar{\theta}_t - \theta_*\right\|^2 + \frac{1}{\beta}\left(\alpha_{sK}C_{\text{back}}\mathbb{E}h(\boldsymbol{\theta}_{t-\tau})\right)^2 + 2\alpha_{sK}^2 C_{\text{prog}}C_{\text{back}}\mathbb{E}h(\boldsymbol{\theta}_{t-\tau}).
\end{aligned}
$$

Finally, for $H_5$, by Young's inequality, we have

$$H_5 \leq \frac{\tau + 2K}{\alpha_{sK}(\tau + K)(\tau + 3K)} \mathbb{E}\|\bar{\theta}_t - \bar{\theta}_{t-\tau}\|^2$$
$$+ \frac{\alpha_{sK}(\tau + K)(\tau + 3K)}{\tau + 2K} \mathbb{E}\left[\mathbb{E}\left[\|\boldsymbol{g}_t(\boldsymbol{\theta}_{t-\tau}, \boldsymbol{O}_t) - \boldsymbol{g}_{t-\tau}(\boldsymbol{\theta}_{t-\tau}, \tilde{\boldsymbol{O}}_t)\|^2 \,\Big|\, \mathcal{F}_{t-\tau}\right]\right].$$

Since $\boldsymbol{\theta}_{t-\tau}$ is deterministic given $\mathcal{F}_{t-\tau}$, we can apply a similar argument to (37) here, which gives

$$H_5 \leq \frac{(\tau + 2K)}{\alpha_{sK}(\tau + K)(\tau + 3K)} \mathbb{E}\|\bar{\theta}_t - \bar{\theta}_{t-\tau}\|^2 + \alpha_{sK}(\tau + 2K)\left(\frac{4H^2}{N} + \alpha_{sK}^2 C_{\text{back}}^2(\tau)H^2\right).$$

By Corollary I.8.1 and Lemma I.5, we get

$$H_5 \leq \alpha_{sK}(\tau + 2K)\left(256\mathbb{E}\left\|\bar{\theta}_{sK} - \theta_*\right\|^2 + 4V(\tau) + \frac{4H^2}{N} + \alpha_{sK}^2 C_{\text{back}}^2(\tau)H^2\right)$$
$$\leq \alpha_{sK}(\tau + 2K)\Big(256(1 + 1/32)\mathbb{E}\left\|\bar{\theta}_t - \theta_*\right\|^2 + 256(1 + 32)\mathbb{E}\left\|\bar{\theta}_{sK} - \bar{\theta}_t\right\|^2$$
$$+ 4V(\tau) + \frac{4H^2}{N} + \alpha_{sK}^2 C_{\text{back}}^2(\tau)H^2\Big)$$
$$\leq \alpha_{sK}(\tau + 2K)\left(264\mathbb{E}\left\|\bar{\theta}_t - \theta_*\right\|^2 + 8448\alpha_{sK}^2 C_{\text{prog}}^2(\tau) + 4V(\tau) + \frac{4H^2}{N} + \alpha_{sK}^2 C_{\text{back}}^2(\tau)H^2\right)$$

We further require that $132\alpha_{sK}(\tau + 2K) \leq \beta$. Then, plugging $H_3, H_4, H_5$, and $V(\tau)$ back gives

$$\frac{1}{N}\sum_{i=1}^{N} 2\mathbb{E}\left\langle \bar{\theta}_t - \theta_*, g_t^{(i)}(\theta_t^{(i)}, O_t^{(i)}) - g_{t-\tau}^{(i)}(\theta_t^{(i)}, \tilde{O}_t^{(i)}) \right\rangle$$
$$\leq 4\beta\mathbb{E}\|\bar{\theta}_t - \theta_*\|^2 + \frac{1}{\beta}\alpha_{sK}^2\left(4C_{\text{prog}}^2 + 2\beta C_{\text{prog}}C_{\text{back}}H_{\text{drift}} + C_{\text{back}}^2 H_{\text{drift}}^2\right) + \frac{2\beta\Lambda^2}{w^2}$$
$$+ \alpha_{sK}(\tau + 2K)\left(8448\alpha_{sK}^2 C_{\text{prog}}^2 + 4\alpha_{sK}^2 C_{\text{var}} + 16m^2\rho^{2\tau}H^2 + \alpha_{sK}^2 C_{\text{back}}^2 H^2 + \frac{132H^2}{N}\right).$$
$$\tag{48}$$

Putting Equations (43) to (45), (47) and (48) and Lemmas I.2 and I.8 back into Lemma I.1, we get

$$\mathbb{E}\left\|\bar{\theta}_{t+1} - \theta_*\right\|^2 \leq \mathbb{E}\left\|\breve{\theta}_{t+1} - \theta_*\right\|^2$$
$$\leq (1 - 2\alpha_t w)\left\|\bar{\theta}_t - \theta_*\right\|^2 + 8\alpha_t\beta\mathbb{E}\left\|\bar{\theta}_t - \theta_*\right\|^2$$
$$+ \alpha_t\left(\frac{\Lambda^2(\epsilon_p, \epsilon_r)}{\beta} + \frac{2\beta\Lambda^2(\epsilon_p, \epsilon_r)}{w^2} + \frac{1}{\beta}\alpha_{s'K}^2(1 + \gamma + \sigma LH)^2 C_{\text{drift}}^2 + \frac{1}{\beta}\alpha_{sK}^2 C_{\text{prog}}^2 L^2 H_{\text{drift}}^2 \sigma^2\right.$$
$$+ m\rho^\tau H_{\text{drift}}\left(\frac{1}{\beta}m\rho^\tau H_{\text{drift}} + 4\alpha_{sK}C_{\text{prog}}\right) + \frac{1}{\beta}\alpha_{sK}^2\left(4C_{\text{prog}}^2 + 2\beta C_{\text{prog}}C_{\text{back}}H_{\text{drift}} + C_{\text{back}}^2 H_{\text{drift}}^2\right)$$
$$+ \alpha_{sK}(\tau + 2K)\left(8448\alpha_{sK}^2 C_{\text{prog}}^2 + 4\alpha_{sK}^2 C_{\text{var}} + 16m^2\rho^{2\tau}H^2 + \alpha_{sK}^2 C_{\text{back}}^2 H^2 + \frac{132H^2}{N}\right)\Big)$$
$$+ \alpha_t^2\left(64\left(\mathbb{E}\left\|\bar{\theta}_t - \theta_*\right\|^2 + \frac{\Lambda^2(\epsilon_p, \epsilon_r)}{w^2}\right) + \alpha_{sK}^2 C_{\text{var}} + 4m^2\rho^{2\tau}H^2 + \frac{32H^2}{N}\right).$$

Note that $\tau$ is a virtual time range that we backtrack, and we have not determined it yet. Now we require it to be large enough such that $m\rho^\tau \leq \alpha_t$. We also do not want $\tau$ to be too large. Thus, we fix

$$\tau = \lceil(\log\alpha_t - \log m)/\log\rho\rceil \asymp \log\alpha_t^{-1}. \tag{49}$$

We also require that the decay rate of $\alpha_t$ is non-increasing and $\sum_{t=0}^{\infty}\alpha_t = +\infty$. Then, there exists $T_1 > 0$ such that for any $t \geq T_1$, it holds that

$$sK \geq t - \tau - K = t - \left\lceil\frac{\log\alpha_t - \log m}{\log\rho}\right\rceil - K \geq \frac{t}{2}.$$

The requirement on the step-size also gives $\limsup_{t\to\infty} \alpha_{t/2}/\alpha_t < +\infty$. Then, there exists $C'_\alpha, C_\alpha > 0$ such that for any $t \geq 0$, we have

$$\frac{\alpha_{sK}}{\alpha_t} \leq C'_\alpha \cdot \limsup_{t\to\infty} \frac{\alpha_{t/2}}{\alpha_t} = C_\alpha.$$

Thus, after some rearrangement, we get

$$\mathbb{E}\left\|\bar{\theta}_{t+1} - \theta_*\right\|^2$$

$$\leq (1 - 2\alpha_t w + 8\alpha_t \beta + 64\alpha_t^2)\mathbb{E}\left\|\bar{\theta}_t - \theta_*\right\|^2 + 4\alpha_t^2(33C_\alpha(\tau + 2K) + 8)\frac{H^2}{N}$$

$$+ \alpha_t^3 C_\alpha^2 \left(\frac{1}{\beta}\left((1 + \gamma + \sigma LH)^2 C_{\text{drift}}^2 + C_{\text{prog}}^2 L^2 H_{\text{drift}}^2 \sigma^2 + H_{\text{drift}}^2 + 4C_{\text{prog}}^2 + 2\beta C_{\text{prog}} C_{\text{back}} H_{\text{drift}} + C_{\text{back}}^2 H_{\text{drift}}^2\right)\right.$$

$$\left. + 4C_{\text{prog}} H_{\text{drift}}\right) + \alpha_t^4 C_\alpha^3((\tau + 2K)(8448C_{\text{prog}}^2 + 4C_{\text{var}} + 16H^2 + C_{\text{back}}^2 H^2) + C_{\text{var}} + 4H^2)$$

$$+ \alpha_t \left(\frac{1}{\beta} + \frac{2\beta + 64\alpha_t}{w^2}\right)\Lambda^2(\epsilon_p, \epsilon_r).$$

Now we let $\beta$ and $\alpha_0$ small enough such that

$$8\beta + 64\alpha_0 \leq w.$$

Then we get the final form

$$\mathbb{E}\left\|\bar{\theta}_{t+1} - \theta_*\right\|^2 \leq (1 - \alpha_t w)\mathbb{E}\left\|\bar{\theta}_t - \theta_*\right\|^2 + \alpha_t C_1 \Lambda^2(\epsilon_p, \epsilon_r) + \alpha_t^2 \frac{C_2}{N} + \alpha_t^3 C_3 + \alpha_t^4 C_4,$$

where

$$C_1 = \beta^{-1} + (2\beta + 64\alpha_0)w^{-2},$$

$$C_2 = 4(33C_\alpha(\tau + 2K) + 8)H^2,$$

$$C_3 = C_\alpha^2\left(\frac{1}{\beta}\left((1 + \gamma + \sigma LH)^2 C_{\text{drift}}^2 + C_{\text{prog}}^2 L^2 H_{\text{drift}}^2 \sigma^2 + H_{\text{drift}}^2 + 5C_{\text{prog}}^2 + 2C_{\text{back}}^2 H_{\text{drift}}^2\right)\right. \quad (50)$$

$$\left. + 4C_{\text{prog}} H_{\text{drift}}\right)$$

$$C_4 = C_\alpha^3((\tau + 2K)(8448C_{\text{prog}}^2 + 4C_{\text{var}} + 16H^2 + C_{\text{back}}^2 H^2) + C_{\text{var}} + 4H^2).$$

$\square$

## K  PROOF OF CORROLARIES 2.1 AND 2.2

In this section, we provide the proofs of Corollaries 2.1 and 2.2. Combining with the constant dependencies discussed in Appendix L, we get the final results presented in Section 5.

**Corollary 2.1.** *With a constant step-size $\alpha_t \equiv \alpha_0 \leq w/(2120(2K + 8 + \ln(m/(\rho w))))$, for any $T \in \mathbb{N}$, we have*

$$\mathbb{E}\left\|\bar{\theta}_T - \theta_*^{(i)}\right\|^2 \leq 4e^{-\alpha_0 wT}\left\|\theta_0 - \theta_*^{(i)}\right\|^2 + B,$$

*where $B$ is the squared convergence region radius defined by*

$$B := \frac{1}{w}\left(\left(C_1 + \frac{6}{w}\right)\Lambda^2(\epsilon_p, \epsilon_r) + \alpha_0\frac{C_2}{N} + \alpha_0^2 C_3 + \alpha_0^3 C_4\right).$$

*Proof.* Let $\theta_*$ be the central optimal parameter. By Theorem 2, for any $T \in \mathbb{N}$, we have

$$\mathbb{E}\|\bar{\theta}_T - \theta_*\|^2 \leq (1 - \alpha_0 w)^T \mathbb{E}\|\theta_0 - \theta_*\|^2 + \alpha_0 w\left(B - \frac{6\Lambda^2}{w^2}\right)\sum_{t=0}^{T-1}(1 - \alpha_0 w)^t \leq e^{-\alpha_0 wT}\|\theta_0 - \theta_*\|^2 + B - \frac{6\Lambda^2}{w^2},$$

where the last inequality uses the fact that $(1 - \alpha_0 w) \leq e^{-\alpha_0 w}$ and $\sum_{t=0}^\infty (1 - \alpha_0 w)^t = (\alpha_0 w)^{-1}$. Then by Theorem 1, we get

$$\mathbb{E}\left\|\bar{\theta} - \theta_*^{(i)}\right\|^2 \leq 2\mathbb{E}\|\bar{\theta} - \theta_*\|^2 + 2\frac{\Lambda^2}{w^2} \leq 4e^{-\alpha_0 wT}\|\theta_0 - \theta_*^{(i)}\|^2 + B - \frac{6\Lambda^2}{w^2} + \frac{6\Lambda^2}{w^2}.$$

$\square$

**Corollary 2.2.** *With a linearly decaying step-size $\alpha_t = 4/(w(1 + t + a))$, where $a > 0$ is to guarantee that $\alpha_0 \leq \min\{1/(8K), w/64\}$, there exists a convex combination $\widetilde{\theta}_T$ of $\{\bar{\theta}_t\}_{t=0}^T$ such that*

$$\mathbb{E}\|\widetilde{\theta}_T - \theta_*^{(i)}\|^2 \leq \frac{1}{w} O\left(\frac{C_4}{w^3 T^2} + \frac{C_3 \log T}{w^2 T^2} + \frac{C_2}{wNT} + C_1 \Lambda^2(\epsilon_p, \epsilon_r)\right).$$

*Proof.* Let $c_t = a + t$ and $C = \sum_{t=0}^T c_t \geq (T+1)^2/2$. We define

$$\widetilde{\theta}_T = \frac{1}{C} \sum_{t=0}^T c_t \bar{\theta}_t,$$

which is a convex combination of $\{\bar{\theta}_t\}_{t=0}^T$. Then, by Jensen's inequality, we have

$$\mathbb{E}\left\|\widetilde{\theta}_T - \theta_*\right\|^2 \leq \frac{1}{C} \sum_{t=0}^T c_t \mathbb{E}\left\|\bar{\theta}_t - \theta_*\right\|^2. \tag{51}$$

Let $\theta_*$ be the central optimal parameter. By Theorem 2, we have

$$\frac{1}{2}\mathbb{E}\left\|\bar{\theta}_t - \theta_*\right\|^2 \leq \left(\frac{1}{\alpha_t w} - \frac{1}{2}\right) \mathbb{E}\left\|\bar{\theta}_t - \theta_*\right\|^2 - \frac{1}{\alpha_t w}\mathbb{E}\left\|\bar{\theta}_{t+1} - \theta_*\right\|^2 + B(\alpha_t),$$

where $B(\alpha) = (C_1\Lambda^2 + \alpha C_2/N + \alpha^2 C_3 + \alpha^3 C_4)/w$. Recall our choice of the step-size $\alpha_t = 4/(w(a + t + 1))$; then we have $1/(\alpha_t w) = (a + t + 1)/4$. Plugging this back into (51) gives

$$\mathbb{E}\left\|\widetilde{\theta}_T - \theta_*\right\|^2 \leq \frac{1}{C}\sum_{t=0}^T c_t \left(\frac{a+t-1}{2}\mathbb{E}\left\|\bar{\theta}_t - \theta_*\right\|^2 - \frac{a+t+1}{2}\mathbb{E}\left\|\bar{\theta}_{t+1} - \theta_*\right\|^2 + 2B(\alpha_t)\right)$$

$$= \frac{1}{2C}\sum_{t=0}^T \left((a+t-1)(a+t)\mathbb{E}\left\|\bar{\theta}_t - \theta_*\right\|^2 - (a+t)(a+t+1)\mathbb{E}\left\|\bar{\theta}_{t+1} - \theta_*\right\|^2\right)$$

$$+ \frac{2C_1\Lambda^2}{w} + \frac{8C_2}{CNw^2}\sum_{t=0}^T \frac{a+t}{a+t+1} + \frac{32C_3}{Cw^3}\sum_{t=0}^T \frac{a+t}{(a+t+1)^2} + \frac{128C_4}{Cw^4}\sum_{t=0}^T \frac{a+t}{(a+t+1)^3}$$

$$\leq \frac{1}{2C}\left(a(a-1)\left\|\bar{\theta}_0 - \theta_*\right\|^2 - (a+T)(a+T+1)\mathbb{E}\left\|\bar{\theta}_{T+1} - \theta_*\right\|^2\right)$$

$$+ \frac{2C_1\Lambda^2}{w} + \frac{8C_2(T+1)}{CNw^2} + \frac{32C_3}{Cw^3}\sum_{t=0}^T \frac{1}{t+1} + \frac{128C_4}{Cw^4}\sum_{t=0}^T \frac{1}{(t+1)^2}$$

$$\leq \frac{a^2\left\|\theta_0 - \theta_*\right\|^2}{T^2} + \frac{2C_1\Lambda^2}{w} + \frac{8C_2}{w^2 NT} + \frac{32C_3}{w^3 T^2}O(\log(T)) + \frac{256C_4}{w^4 T^2}$$

$$= O\left(\frac{a^2}{T^2} + \frac{C_4}{w^4 T^2} + \frac{C_3 \log T}{w^3 T^2} + \frac{C_2}{w^2 NT} + \frac{C_1\Lambda^2}{w}\right).$$

Then by Theorem 1 and the fact that $1/w \lesssim C_1$ (see Appendix L) and $a \lesssim K/w^2$, we get

$$\mathbb{E}\left\|\widetilde{\theta}_T - \theta_*^{(i)}\right\|^2 \leq 2\mathbb{E}\left\|\widetilde{\theta}_T - \theta_*\right\|^2 + 2\frac{\Lambda^2}{w^2} = O\left(\frac{K^2 + C_4}{w^4 T^2} + \frac{C_3 \log T}{w^3 T^2} + \frac{C_2}{w^2 NT} + \frac{C_1\Lambda^2}{w}\right).$$

$\square$

## L   CONSTANT DEPENDENCIES

In this section, we establish explicit dependencies between the constants. We begin by introducing problem constants that are independent of other parameters: the reward cap $R > 0$, discount factor $\gamma \in (0, 1)$, projection radius $\bar{G} > 0$,[1] local update period $K$, and kernel-related constants, $m \geq 1, \rho \in (0, 1)$, and $\lambda := \min_{i \in [\bar{N}]} \lambda^{(i)} \in (0, 1]$. Throughout this paper, we use asymptotic notation

as $R, \bar{G}, K, m \to \infty$ and $\gamma, \rho \to 1$. We also use the nonasymptotic notation $a \lesssim b$ and $b \gtrsim a$ to indicate that there exists $C \geq 0$ such that $a \leq Cb$, and $a \asymp b$ to indicate that both $a \lesssim b$ and $b \gtrsim a$ hold.

We first give the dependencies of $\sigma'$ defined in (9). By its definition, we have $\sigma' \geq 0$, and

$$\sigma' \leq \frac{\log m}{-\log \rho} + \frac{1}{1 - \rho} \leq \frac{\log m + 1}{1 - \rho} = O\left(\frac{\log m}{1 - \rho}\right),$$

where the asymptotic notation holds as $\rho \to 1$ and $m \to \infty$. We also get $\sigma = \sigma' + 2 = O(\log m/(1 - \rho))$. We will now use $\sigma$ as a base constant.

$w$ is an important MDP constant and plays a critical role in the convergence rate. By its definition (19), we get $w \leq 1/2$ and

$$w = \min_{i \in [\bar{N}]} w_i \geq \frac{1 - \gamma}{2} \min_{i \in [\bar{N}]} \lambda^{(i)} = \frac{1 - \gamma}{2} \lambda,$$

which gives

$$w^{-1} = O((1 - \gamma)^{-1}).$$

We then consider $G$ and $H$. By Corollary I.5.3, we get

$$G = \frac{2(2\bar{G} + R)}{1 - 16\alpha_0^2 K^2 \gamma} = O\left(\frac{\bar{G} + R}{1 - \gamma}\right).$$

When $\gamma$ is near 1, the above bound is undesirable. Thus, when $\gamma$ is large, we can further require $4\alpha_0 K < \sqrt{0.5}$, which gives $G \leq 4(2\bar{G} + R)$. Without loss of generality, we have

$$G \asymp \bar{G} + R$$

And by the definition of $H$, we get

$$H = R + (1 + \gamma)G \asymp \bar{G} + R.$$

We now use $H$ as a base constant and replace $\bar{G} + R$ with $H$ for simplicity. $H$ can be viewed as the scale of the problem. If we choose $\bar{G}$ according to (Zou et al., 2019), then $H = O(R/(1 - \gamma))$.

By (20), we get the dependencies of the policy improvement operator's Lipschitz constant $L$:

$$L \leq \frac{w}{\sigma H}.$$

We now address the constants in Appendix I. By Lemma I.4, we directly have

$$C_{\text{drift}} = O(KH).$$

We now consider $\alpha_0$. There are two requirements on $\alpha_0$ throughout the proof: $4K\alpha_0 < 1$ in Lemmas I.4 and I.5, and $64\alpha_0 \leq w$ in Appendix J. Combining these conditions gives

$$\alpha_0 \leq \min\left\{\frac{1}{4K}, \frac{w}{64}\right\} \lesssim \min\left\{K^{-1}, w\right\}.$$

Therefore, $C_{\text{prog}}$ in Lemma I.5 has the following dependencies:

$$C_{\text{prog}} = O((\tau + K)(H + K^{-1} \cdot KH)) = O((\tau + K)H).$$

And $C_{\text{back}}$ in Lemma I.7 has the following dependencies:

$$C_{\text{back}} = O(\tau^2 LH) = O(\tau^2 w).$$

Then, $C_{\text{var}}$ in Lemma I.8 is controlled by

$$C_{\text{var}} = O(C_{\text{drift}}^2 + w^2 C_{\text{prog}}^2 + H^2 C_{\text{back}}^2) = O(H^2(K^2 + w^2 \tau^4)).$$

---

[1]One can choose $\bar{G} = R/w$ as suggested in Zou et al. (2019). Here, we make it a pre-defined algorithm constant.

Next, we give the dependencies of constants in Appendix J. By definition, we have

$$H_{\text{drift}} = O(H + \alpha_0 C_{\text{drift}}) = O(H).$$

By the requirement of $\beta$, we have

$$\beta \asymp w.$$

And we have $C_\alpha = O(1)$. Therefore, we get

$$
\begin{aligned}
C_1 &= O(w^{-1}) = O((1-\gamma)^{-1}), \\
C_2 &= O(H^2(\tau + K)), \\
C_3 &= O\left(H^2(w^{-1}(\tau^2 + K^2) + w\tau^4)\right), \\
C_4 &= O(H^2(\tau + K)(\tau^2 + K^2 + w^2\tau^4)).
\end{aligned}
$$

Finally, we give the dependencies of constants in Corollaries 2.1 and 2.2. In Corollary 2.1, we choose a constant step-size $\alpha_t = \alpha_0$. There are two requirements on $\alpha_t$ throughout the proof: $132(\tau + K)\alpha_{sK} \leq 1$ in Corollary I.8.1 and $132\alpha_{sK}(\tau + 2K) \leq \beta$ in Appendix J. A concrete condition satisfying these requirements is $\alpha_0 \leq w/(2120\,(2K + \ln(2120m)/(\rho w)))$. Furthermore, if we choose a small enough initial step size such that $\alpha_0^{-1} \asymp \tau \gtrsim \max\left\{K, w^{-1}\right\}$, then $C_2, C_3$, and $C_4$ becomes

$$C_2 = O(H^2\tau) = \widetilde{O}(H^2),\ C_3 = O(H^2 w\tau^4) = \widetilde{O}(H^2 w),\ C_4 = O(H^2 w^2 \tau^5) = \widetilde{O}(H^2 w^2), \quad (52)$$

where $\widetilde{O}$ omits the logarithmic dependencies on $\tau$. Then the convergence region radius in Corollary 2.1 becomes

$$B = O\left(\alpha_0^2 H^2 \tau^4 + \frac{\alpha_0 H^2 \tau}{N(1-\gamma)} + \frac{\Lambda^2}{(1-\gamma)^2}\right) = \widetilde{O}\left(\alpha_0^2 H^2 + \frac{\alpha_0 H^2}{N(1-\gamma)} + \frac{\Lambda^2}{(1-\gamma)^2}\right)$$

With the linearly decaying step-size in Corollary 2.2, (49) gives

$$\tau \asymp \log T$$

as the total number of iterations $T \to \infty$. And the requirements on $\alpha_t$ in previous discussion automatically hold for large enough $t$. Omitting the logarithmic dependencies on $T, C_2, C_3$, and $C_4$ in this case are the same as (52). Therefore, the finite-time error bound in Corollary 2.2 becomes

$$\mathbb{E}\left\|\widetilde{\theta}_T - \theta_*\right\|^2 = \frac{H^2}{(1-\gamma)^2} \cdot O\left(\frac{\tau^5}{T^2} + \frac{\tau}{NT} + \frac{\Lambda^2(\epsilon_p, \epsilon_r)}{H^2}\right) = \frac{H^2}{(1-\gamma)^2} \cdot \widetilde{O}\left(\frac{1}{NT} + \frac{\Lambda^2(\epsilon_p, \epsilon_r)}{H^2}\right).$$

# M  TABULAR FEDSARSA

In this section, we reduce our algorithm and analysis to the tabular setting. Recall that $S$ and $A$ are the measures of the state space $\mathcal{S}$ and action space $\mathcal{A}$, respectively. For the tabular setting, $S$ and $A$ are the numbers of states and actions. Then, we choose the feature map to be an indicator vector function, i.e.,

$$\phi: \mathcal{S} \times \mathcal{A} \to \mathbb{R}^{SA}, \quad [\phi(s,a)]_{(s',a')} \mapsto \mathbb{1}\{(s',a') = (s,a)\},$$

where we treat $\phi(s,a)$ as a vector and use a two-dimensional index such that $[\phi(s,a)]_{(s',a')}$ is the $(s',a')$-th element of $\phi(s,a)$; $\mathbb{1}$ is the indicator function. Using this feature map, the parameter $\theta$ is indeed the estimated value function table:

$$Q_\theta(s,a) = \phi^T(s,a)\theta = [\theta]_{(s,a)}$$

Therefore, the local update rule in Algorithm 1 reduces to the tabular SARSA update rule (2).

We now show a natural bound $G$ for $\|\theta\|_2$ without an explicit projection. First, the true value function (1) is bounded by

$$|q_\pi(s,a)| \leq \sum_{t=0}^{\infty} \gamma^t R = \frac{R}{1-\gamma} =: G_\infty.$$

Suppose current estimated value function satisfies that $|Q_t(s, a)| \leq G_\infty$ for any state-action pair, then we have

$$\begin{aligned}
|Q_{t+1}(s, a)| &= |Q_t(s, a) + \alpha(r(s, a) + \gamma Q_t(s', a') - Q_{s,a})| \\
&= |(1 - \alpha)Q_t(s, a) + \alpha\gamma Q_t(s', a') + \alpha r(s, a)| \\
&\leq (1 - \alpha)G_\infty + \alpha\gamma G_\infty + \alpha R \\
&= (1 - \alpha + \alpha\gamma)\frac{R}{1 - \gamma} + \alpha R \\
&= \frac{R}{1 - \gamma} = G_\infty.
\end{aligned}$$

Therefore, if the bound holds for the initial estimated value function, it holds for all sequential, local or central, estimated value functions. However, $G_\infty$ is a upper bound for $\|\theta\|_\infty$. For 2-norm, we have

$$\|\theta\|_2 \leq \sqrt{SA}\|\theta\|_\infty \leq \frac{\sqrt{SA}R}{1 - \gamma} =: G,$$

which further gives

$$H = O\left(\frac{\sqrt{SA}R}{1 - \gamma}\right).$$

Also, for tabular FedSARSA, Remark 2 tells us that

$$w^{-1} = O\left(\frac{1}{\lambda(1 - \gamma)}\right),$$

$\lambda$ is the probability of visiting the least probable state-action pair under the steady distribution of the optimal policy across all agents. Then, Corollary 2.2 can be translated into the following corollary.

**Corollary 2.3** (Finite-time error bound for tabular FedSARSA with decaying step-size). *With a linearly decaying step-size $\alpha_t = 4/(w(1 + t + a))$, where $a > 0$ is to guarantee that $\alpha_0 \leq \min\{1/(8K), w/64\}$, there exists a convex combination $\widetilde{\theta}_T$ of $\{\bar{\theta}_t\}_{t=0}^T$ such that*

$$\mathbb{E}\big\|\widetilde{\theta}_T - \theta_*^{(i)}\big\|_2^2 \leq \frac{1}{\lambda^2(1 - \gamma)^2} \cdot \widetilde{O}\left(\frac{SAR^2}{\lambda^2(1 - \gamma)^4 T^2} + \frac{SAR^2}{(1 - \gamma)^2 NT} + \Lambda^2\left(\epsilon_p, \epsilon_r\right)\right).$$

*where the asymptotic notation suppresses the logarithmic factors. Since $\|\theta\|_\infty \leq \|\theta\|_2$, we also get the finite-time error bound under the infinity norm.*

