# OpenReview forum: "Finite-Time Analysis of On-Policy Heterogeneous Federated Reinforcement Learning"
_ICLR.cc/2024/Conference — ICLR 2024 poster_

### Official Review · Reviewer_pmTu · 2023-10-30

**Soundness:** 3 good
**Presentation:** 3 good
**Contribution:** 3 good
**Rating:** 6
**Confidence:** 3

**Summary:**

The paper proposes the federated version of SARSA algorithm and analyses its convergent performance with the existence of heterogeneity in both transition dynamics and reward functions.
Different from classical settings of federated reinforcement learning, the paper does not assume that agents have to share the same transition dynamics and reward functions.
The paper demonstrates that its proposed algorithm achieves a linear speedup for the convergence to the optimal answer in each local environment both theoretically and empirically.

**Strengths:**

1. The paper considers heterogeneity in both transition dynamics and reward functions. Moreover, it quantifies the degree of these heterogeneities and discusses their effect in the final convergence of FedSARSA.
2. The paper discusses the convergence region and linear speedup of FedSARSA. It is claimed that smaller learning rates and a larger number of participating agents will help tighten the convergence region, which matches the intuition.
3. The numerical experiment is carried out in settings with different degrees of heterogeneity.

**Weaknesses:**

1. What does MSE in the numerical experiments stand for? Does it mean the averaged MSE of current parameter to optimal parameters in different environments?
2. The explanation of numerical experiments is not enough. For example, why the MSE of FedSARSA with a large number of $N$ ($N=40$) increase along the training process when $\epsilon_p>0,\epsilon_r>0$? And where is the confidence bound for the numerical experiments?
3. The convergence MSE of FedSARSA with different number of agents are different from each other. What makes that difference?

**Questions:**

See Weaknesses.

---

> ### Author Response · Authors · 2023-11-18
> **Response to Reviewer pmTu**
>
> Dear Reviewer pmTu,
>
> We thank your for the detailed review and positive feedback. Below we provide the responses clarifying the questions:
>
> > **W1. What does MSE in the numerical experiments stand for? Does it mean the averaged MSE of current parameter to optimal parameters in different environments?**
>
> We deferred the detailed simulation setup to Appendix C.
> In the second paragraph of Appendix C.1 (ADDITIONAL SIMULATIONS FOR FEDSARSA), we defined our MSE: the mean squared error of the current *averaged* parameter $\bar{\theta}\_t$ to the *reference* parameter $\theta\_{\mathrm{ref}}^{(1)}$, where the latter is an approximate optimal parameter of the *first* MDP.

---

> > ### Author Response · Authors · 2023-11-18
> > **Response to Reviewer pmTu [Continued]**
> >
> > > **W3. The convergence MSE of FedSARSA with different number of agents are different from each other. What makes that difference?**
> >
> > The fact that the MSE of FedSARSA is different with a different number of agents is, in fact, to be expected. The reason for this is as follows. Our simulations are run with a constant step-size. As Corollary 2.1 reveals, in this case, the effect of the linear speedup shows up in tightening the size of the ball (roughly by a factor of $1/N$) to which the iterates converge. Thus, with a larger number of agents, we expect a smaller error-floor/convergence ball. In line with this theory, our simulations do reveal that increasing the number of agents causes the error curve to flat-line at a lower height, indicating a smaller error-floor.

---

> ### Author Response · Authors · 2023-11-18
> **Response to Reviewer pmTu [Continued]**
>
> > **W2. The explanation of numerical experiments is not enough. For example, why the MSE of FedSARSA with a large number of $N$ ($N=40$) increase along the training process when $\epsilon _{p}>0,\epsilon _{r}>0$? And where is the confidence bound for the numerical experiments?**
>
> We thank the Reviewer for their constructive suggestions. In the revised paper, we have now re-plotted all the graphs with 95\% confidence regions.
> Regarding the non-monotonic behavior of the MSE, we have the following explanation.
> Our theoretical analysis considers the expected MSE, whereas the simulation results only approximate this expectation.
> Additionally, our theoretical analysis provides an upper bound of the convergence region radius, while the actual empirical MSE may vary from this upper bound to different extents.
> As such, the fact that the plotted error-curve is not exactly monotone does not violate our theory; it does still reflect what Corollary 2.1 reveals: an initial exponential decay phase, followed by the error trajectory settling down.
> We also want to remark three facts that complement the above explanation: (i) this phenomenon is influenced by the distribution of agents' optimal parameters. Given that we re-generate all agents' MDPs for different numbers of agents, this phenomenon only occurs in certain instances; (ii) the MSE is relative to a reference point, which is returned by an approximation algorithm (since we don't have the closed-form expression of the optimal parameter), and thus it only approximates the actual MSE relative to the optimal parameter; and (iii) this phenomenon only occurs when the heterogeneity level is high, where the performance of `FedSARSA` could be significantly hindered by the disparity among agents.

---

### Official Review · Reviewer_uJdU · 2023-10-31

**Soundness:** 3 good
**Presentation:** 3 good
**Contribution:** 3 good
**Rating:** 8
**Confidence:** 3

**Summary:**

This work performs theoretical studies on federated reinforcement learning with clients facing heterogeneous environments. In particular, the classical SARSA algorithm is extended to the federated version. Theoretical analyses are established to demonstrate the finite-sample convergence of the proposed FedSARSA under linear function approximation. In particular, linear speedups are reported with the established results.

**Strengths:**

- This work follows the interesting line of work of extending federated learning to the domain of decision-making under environmental heterogeneity. This setting is well-motivated and has wide practical implications.

- The established results are solid and novel based on my reading. In particular, no similar results have been reported on FRL with heterogeneous clients in the planning task with both linear function approximation and linear speedup.

- Despite the theoretical nature, the overall presentation is clear and the key intuitions are provided. The listed sketch of the proof especially facilitates the readability.

**Weaknesses:**

- I am overall satisfied with this work. There is just one minor question I have for the authors. As mentioned at the end of page 8, the obtained results from federated RL can be leveraged as initialization points for finetuning with just local data. I imagine the analyses would not be different given existing works, and wonder whether it would be possible to state the finetuning results, which may better highlight the impact of cooperation on accelerating individual learning.

**Questions:**

Please see the weakness section.

---

> ### Author Response · Authors · 2023-11-18
> **Response to Reviewer uJdU**
>
> Dear Reviewer uJdU,
>
> Thank you for your encouraging comments. We are very glad to know that you find our results solid and novel.
>
> Regarding the interesting question you raised, we first note that the finite-time error of local single-agent SARSA is given by:
> $$O\left(\frac{e_0^2}{M^2} + \frac{\sigma^2\log M}{M}\right),$$
> where $e\_0$ is the initial mean-squared error, $\sigma^2$ is the noise variance described by problem constants, and $M$ is the number of iterations performed by the agent. Now consider any agent $i$, and suppose it uses the output of `FedSARSA` as a warm-start. From Corollary 2.2, we then observe that the initial expected squared error $e\_0^2$ would be $\tilde{O}(1/(NT) + \Lambda^2(\epsilon\_p, \epsilon\_r))$, where $\Lambda^2(\epsilon\_p, \epsilon\_r)$ is a measure of the heterogeneity in the agents' MDPs, as defined in Theorem 1. The above discussion suggests that if the agents' MDPs are similar, i.e., if $\Lambda^2(\epsilon\_p, \epsilon\_r)$ is small, then federation yields a small initial error $e\_0$. This is precisely the benefit of collaboration afforded by our approach. At this point, we do not have a better analytical explanation of the benefits of collaboration + fine-tuning; we are happy to add the above discussion to the main paper as the Reviewer sees fit. We should note here that it is possible to potentially provide a sharper analysis of the benefits of fine-tuning in federated RL by drawing on analogous recent studies for federated supervised learning; see, for instance, Collins et al. (2022) and Cheng et al. (2021). This is a ripe direction of research that we are exploring at this point.
>
> ## References
>
> - Collins, L., Hassani, H., Mokhtari, A. and Shakkottai, S., 2022. Fedavg with fine tuning: Local
> updates lead to representation learning. In *Proc. Advances in Neural Information Processing
> Systems (NeurIPS)*.
> - Cheng, G., Chadha, K. and Duchi, J., 2021. Fine-tuning is fine in federated learning. arXiv
> preprint arXiv:2108.07313, 3.

---

> > ### Comment · Reviewer_uJdU · 2023-11-23
> >
> > Thank you for the response! I would like to keep my score for now and discuss with AC and other reviewers.

---

### Official Review · Reviewer_qewk · 2023-11-01

**Soundness:** 3 good
**Presentation:** 1 poor
**Contribution:** 2 fair
**Rating:** 5
**Confidence:** 3

**Summary:**

This paper studies federated on-policy reinforcement learning with linear function approximation. It proposes FedSARSA that is a federated version of SARSA. It proves that FedSARSA converges to the neighborhood of the optimal parameter with a linear speed up.

**Strengths:**

+ FedSARSA is intuitive and reasonable.
+ The theoretical result that proves linear speedup is interesting.

**Weaknesses:**

- The theoretical results do not have the impact of periodic updating.
- The authors do not specify the communication cost and how it trades off with the convergence.
- The FedSARSA is a straightforward of single-agent SARSA -- the only difference is to aggregate the parameter estimation from all agents, which is a straightforward average.
- The authors talked about how heterogeneity is captured in the convergence, but this relationship is not well articulated. In FL, one would first need to define the heterogeneity metric and then express the convergence bound as a function of this metric. Furthermore, such heterogeneity should be defined on the data, not the underlying distribution.
- Paper writing needs some work. It is strange to not have a Conclusion section.

**Questions:**

- The motivation is unclear to me -- why do we want to learn a single universal policy? Each agent may interact with his/her own environment and learn a personalized policy for that environment. Isn't that better to be deployed on that environment than the single averaged policy across all environments?
- I don't see the linear speedup in the simulation results?

---

> ### Author Response · Authors · 2023-11-17
> **Response to Reviewer qewk**
>
> Dear Reviewer qewk,
>
> Thank you for your review and constructive suggestions. Regarding the weaknesses and questions, we provide the following detailed responses:
>
> > **W1. & W2. The theoretical results do not have the impact of periodic updating. The authors do not specify the communication cost and how it trades off with the convergence.**
>
> Thank you for raising the points about the effects of periodic synchronization, and the trade-offs between convergence and communication cost. In what follows, we explain that the effect of periodic synchronization—as captured by the number of local steps $K$—**does, in fact, manifest in our finite-time bounds**. To see this, notice, for instance, that in Corollary 2.2, we need the **initial step-size** to satisfy $\alpha_0 \lesssim 1 /K$. This is achieved by picking the parameter $a$ (in Corollary 2.2) to be large enough such that $\alpha\_0 \leq \min \\{1/(8K), w/64\\}$. The effect of $a$ shows up in only the quadratic $O(K^2/T^2)$ term of our bound. Since for large $T$, this term will eventually get dominated by the $O(1/(NT))$ term, we had initially deferred the exact dependence on $K$ to the appendix. In response to the Reviewer's comment, we have now explicitly shown this dependence in the main paper. The main takeaway here is that a larger $K$ reduces communication, but comes at the expense of increasing the convergence bound via the (eventually negligible) $O(K^2/T^2)$ term. **Thus, our result clearly reveals the trade-off between the convergence speed and communication cost**.

---

> ### Author Response · Authors · 2023-11-17
> **Response to Reviewer qewk [Continued]**
>
> > **W3. The FedSARSA is a straightforward of single-agent SARSA -- the only difference is to aggregate the parameter estimation from all agents, which is a straightforward average.**
>
> We thank the Reviewer for pointing out the straightforwardness of our method, which, in fact, also highlights the practicality of our method. We consider this to be a strength, and not a weakness of our paper. The main point here is that while our approach is "simple", its analysis is not by any means: it takes significant technical effort to clearly establish the benefits - in terms of linear speedup - offered by our approach. We explain these challenges again as is done in the paper in Section 5.3.
>
> ***Challenges and Novelty in our Technical Approach.*** Arriving at our main result Theorem 2 is not simply a matter of tweaking the proofs in existing papers. Why? *The update direction of our algorithm `FedSARSA` may not correspond to the SARSA update direction of any MDP.* This challenge is unique to our setting, and neither shows up in the centralized SARSA analysis, nor in the existing MARL/FRL analyses with homogeneous MDPs. Second, unlike the standard FL optimization setting on i.i.d data, our problem does not correspond to minimizing a static loss function; moreover, we need to contend with *temporally correlated Markovian data from heterogeneous MDPs*. This clearly sets our work apart from existing FL literature. To make things even harder, the policies keep changing at the agents, and they only communicate once in a while. Given these challenges, our work develops a theoretical framework for answering the following fundamental question:
> **What is the long-term effect of combining local SARSA directions with function approximation from heterogeneous non-stationary Markov chains?** As far as we are aware, despite the long list of papers in the MARL area, no work has investigated this important question theoretically.
>
> To answer the above question, our first key innovation is to rigorously quantify the effect of heterogeneity on SARSA fixed points in Theorem 1. This result is significant in that it reveals how heterogeneity in the rewards
> and transition kernels of MDPs can be mapped to differences in the limiting behavior of
> SARSA on such MDPs from a fixed-point perspective. To prove Theorem 2, we need to control seven different terms that appear in the error-decomposition in Section 5.3. In particular, the complex interplay between function approximation, temporal correlations, and non-stationary heterogeneous Markov chains, makes it highly non-trivial to bound the client-drift effect that arises due to performing multiple local steps. Finally, in the single-agent setting, one does not need to worry about establishing a linear speedup. In our case, however, we establish the (a priori non-obvious) fact that combining information from non-stationary heterogeneous Markov chains can in fact lead to a linear speedup in sample-complexity.
>
> We should also note here that the most popular FL algorithm to date—`FedAvg`—also relies on a simple averaging of models. This has not diminished its impact in any way. Given that `FedSARSA` is the first on-policy FRL algorithm, we did not find any reason to study anything more involved, since even our simple approach provides strong guarantees that are challenging to establish.

---

> ### Author Response · Authors · 2023-11-17
> **Response to Reviewer qewk [Continued]**
>
> > **W4. The authors talked about how heterogeneity is captured in the convergence, but this relationship is not well articulated. In FL, one would first need to define the heterogeneity metric and then express the convergence bound as a function of this metric. Furthermore, such heterogeneity should be defined on the data, not the underlying distribution.**
>
> First, we define the heterogeneity metrics in Definitions 1 and 2, and their impact on the convergence bound is indeed expressed as a function of these metrics: $\Lambda(\epsilon _{p},\epsilon _{r})$, defined in Theorem 1. In fact, each of the bounds in Corollaries 2.1 and 2.2. exhibit the effect of $\Lambda(\epsilon _{p},\epsilon _{r})$; we even discuss this point in the paragraph titled "Convergence Region." As for the heterogeneity metric, since the data in our online RL setting are precisely the **Markovian observations**, we defined heterogeneity on the basic **data-generating** components—the transition kernels and reward functions in the MDPs. Since transition kernels and reward functions are the basic elements of MDPs, this seemed like the most natural thing to do. That said, it is important to note that our analysis is **agnostic** to the level of heterogeneity. As such, one could in principle formulate other notions of heterogeneity, and an analysis akin to what we provide will very likely go through.

---

> ### Author Response · Authors · 2023-11-17
> **Response to Reviewer qewk [Continued]**
>
> > **W5. Paper writing needs some work. It is strange to not have a Conclusion section.**
>
> Thanks for the suggestion! We have added a conclusion. We would greatly appreciate any additional input concerning our manuscript's writing issues that we can improve.

---

> ### Author Response · Authors · 2023-11-17
> **Response to Reviewer qewk [Continued]**
>
> > **Q1. The motivation is unclear to me -- why do we want to learn a single universal policy? Each agent may interact with his/her own environment and learn a personalized policy for that environment. Isn't that better to be deployed on that environment than the single averaged policy across all environments?**
>
> We note that the Reviewer's question also applies to federated supervised learning: why learn a common model if the agents' data distributions are very different?  The rationale here - as in our work - is that if the agents' distributions (MDPs) are not too different, then even a common model can serve as a coarse good model that the agents can use to further fine-tune. In fact, this is precisely the approach adopted in recent FL works; see "Fedavg with fine tuning: Local updates lead to representation learning", Collins et al., NeuRIPS 22. In line with this work and others, the main motivation of our paper is to develop a federated method that allows each agent to quickly (i.e., with a linear speedup) identify a common policy that lies in the vicinity of its own optimal policy (as characterized by Theorem 1). Each agent can then use this common policy to warm-start its fine-tuning process (Zeng et al., 2021; Cheng et al., 2021; Beck et al., 2023).
>
> Moreover, learning a common policy for different agents is motivated by real-world needs.
> For instance, Spotify, a leading audio streaming company, intends to design a uniform pricing plan that suits the listening habits of all users. Given the substantial variations in listening habits among users, establishing a pricing strategy that aligns with the preferences of all users is of great importance.
>
> ## References
>
> - Collins, L., Hassani, H., Mokhtari, A. and Shakkottai, S., 2022. Fedavg with fine tuning: Local updates lead to representation learning. In *Proc. Advances in Neural Information Processing Systems (NeurIPS)*.
> - Zeng, S., Anwar, M. A., Doan, T. T., Raychowdhury, A., and Romberg, J., 2021. A decentralized policy gradient approach to multi-task reinforcement learning. In *Uncertainty in Artificial Intelligence*. PMLR.
> - Beck, J., Vuorio, R., Liu, E.Z., Xiong, Z., Zintgraf, L., Finn, C. and Whiteson, S., 2023. A survey of meta-reinforcement learning. arXiv preprint arXiv:2301.08028.
> - Cheng, G., Chadha, K. and Duchi, J., 2021. Fine-tuning is fine in federated learning. arXiv preprint arXiv:2108.07313, 3.

---

> ### Author Response · Authors · 2023-11-17
> **Response to Reviewer qewk [Continued]**
>
> > **Q2. I don't see the linear speedup in the simulation results?**
>
> Our simulations are run with a constant-step size. As Corollary 2.1 reveals, in this case, the effect of the linear speedup shows up in tightening the size of the ball (roughly by a factor of $1/N$) to which the iterates converge. **Figure 1 does in fact show that increasing the number of agents causes the error-floor (i.e., the height of the flat line) to decrease.**  The decrease is only approximately linear since the error-floor is also affected by heterogeneity.

---

> ### Author Response · Authors · 2023-11-21
> **Following-up**
>
> We kindly ask the reviewer if any of their concerns remain, and if so, what in particular?

---

> > ### Comment · Reviewer_qewk · 2023-11-22
> >
> > Thank you for the responses. They provide more insight into this work.

---

> > > ### Author Response · Authors · 2023-11-22
> > > **Reconsider Score?**
> > >
> > > Dear Reviewer qewk,
> > >
> > > We are glad to know that our responses have helped provide more insights. We were wondering if we have managed to address all your concerns. If so, would you be willing to reconsider your score?
> > >
> > > Should there be any remaining concerns, we are more than happy to discuss further. We thank you again for your constructive comments and suggestions.

---

### Official Review · Reviewer_rA2b · 2023-11-07

**Soundness:** 3 good
**Presentation:** 4 excellent
**Contribution:** 3 good
**Rating:** 6
**Confidence:** 3

**Summary:**

This paper studies the problem of on-policy federated RL with agents interacting with potentially different environments. A new algorithm FedSARSA is proposed, and shown to converge to a near-optima policy for all gents. Convergence speed analysis is also provided.

**Strengths:**

- The paper is well written. The formulation and ideas are explained clearly.

**Weaknesses:**

- It would be helpful if the authors could provide more intuition about where the speed up comes from. Specifically, what in the problem formulation/assumptions enable this speedup? Intuitively, this would be possible only when things are homogeneous (or close to that).
- Is it possible to comment on the optimality of the finite-time error? Right now only upper bounds are provided.

**Questions:**

- It would be helpful if the authors could provide more intuition about where the speed up comes from. Specifically, what in the problem formulation/assumptions enable this speedup? Intuitively, this would be possible only when things are homogeneous (or close to that).
- Is it possible to comment on the optimality of the finite-time error? Right now only upper bounds are provided.

---

> ### Author Response · Authors · 2023-11-17
> **Response to Reviewer rA2b**
>
> Dear Reviewer rA2b,
>
> Thank you for the constructive comments and positive feedback on our paper. Below, we provide detailed responses to the questions you raised.
>
> > **W1. & Q1. It would be helpful if the authors could provide more intuition about where the speed up comes from. Specifically, what in the problem formulation/assumptions enable this speedup? Intuitively, this would be possible only when things are homogeneous (or close to that).**
>
> To be precise, the linear speedup effect comes from the fact that the Markovian observation sequences across agents are assumed to be **statistically independent**. This is the key assumption that eventually leads to a "variance-reduction" effect when one merges information from different agents. That said, even when the agents' MDPs are identical, a few important things worth noting are the following: (i) for each agent, its own Markovian observations are temporally correlated over time, and are generated based on time-varying Markov chains; (ii) the agents fuse models, and not raw data (e.g., rewards); and (iii) the realizations of rewards and state transitions can differ across agents. Under these conditions, establishing the linear speedup effect requires significant technical work - which is one of the main contributions of our paper.
>
> The Reviewer's intuition is entirely correct. The linear speedup effect is prominent only when the agents' MDPs are not too different. Indeed, if one inspects the final bound in Corollary 2.2, it is of the form $\tilde{O}(1/(NT) + \Lambda^2(\epsilon\_p, \epsilon\_r))$, where $\Lambda^2(\epsilon\_p, \epsilon\_r)$ is a term that captures the heterogeneity in the agents' MDPs. The implication of this result is as follows: FedSARSA enables each agent to converge to a ball of radius $\Lambda(\epsilon\_p, \epsilon\_r)$ around its own optimal parameter at a rate that gets linearly expedited by the number of agents $N$. Each agent can then use the output of FedSARSA as a *warm-start* to further fine-tune based on its own data. Clearly, if the heterogeneity level is low, i.e., if $\Lambda(\epsilon\_p, \epsilon\_r))$ is small, the agents converge quickly to a smaller ball around their optimal parameter. Conversely, if the heterogeneity is large, the $\Lambda(\epsilon\_p, \epsilon\_r)$ dominates, and the benefit of the speedup gets obscured.
> This intuition is also verified in our simulations, as demonstrated in Figure 4 (and additional experiments in Appendix).

---

> ### Author Response · Authors · 2023-11-17
> **Response to Reviewer rA2b [Continued]**
>
> > **W2. & Q2. Is it possible to comment on the optimality of the finite-time error? Right now only upper bounds are provided.**
>
> This is a great question! To the best of our knowledge, there is no sample complexity lower bound for SARSA or federated reinforcement learning. As such, we cannot claim that our bounds are minimax optimal. That said, we now explain that are bounds are consistent with existing results. For instance, when the number of agents is $1$ (and hence, there is no heterogeneity), our finite-time bound in Corollary 2.2 matches that for SARSA in Zou et al. (2019). Second, when there is no heterogeneity, the final bound we achieve in Corollary 2.2 is $\tilde{O}(1/(NT))$. With $N$ agents, each of whom receives $T$ samples, the total number of samples is $NT$. Given $NT$ samples, we can hope to reduce noise variance by at most a factor of $NT$; so it seems quite unlikely that our bound can be improved in this regard. Thus, it only remains to argue whether the additive $\tilde{O}(\Lambda(\epsilon _{p},\epsilon _{r}))$ term - a measure of heterogeneity defined in Theorem 1 - is an artifact of our analysis, or something unavoidable. Notice that the bound we provide is w.r.t. each agent's own optimal parameter. When the agent's MDPs are non-identical, their optimal parameters need not be identical. Thus, for an algorithm like FedSARSA that combines models from all agents, the average model at the server will - in the general case - not converge to any particular agent's optimal parameter. Thus, it seems unlikely that one can hope to achieve a bound that does not exhibit any bias term at all, unless additional assumptions are made. As to the specific bound of $\tilde{O}(\Lambda(\epsilon _{p},\epsilon _{r}))$, this is consistent with the difference in fixed points established in Theorem 1. Furthermore, a heterogeneity bias term of exactly this form is proven to be **unavoidable** in the recent paper  Wang et al. (2023) that studies federated TD learning. We conjecture that a similar lower bound exemplifying the heterogeneity-induced bias can be established for our setting as well; but this requires more work since unlike the linear operators in TD, SARSA involves nonlinear operators. This is indeed a very promising direction of future work, as identified by the Reviewer.
>
> In response to the Reviewer's comment, and to provide more clarity on our bounds, we have now added a **new section**, Appendix B (FINITE-TIME RESULTS COMPARISON), in our revised manuscript. This section provides a comparison of our finite-time results with other related work, demonstrating that our finite-time error bound is consistent with existing studies.
>
> ## References
>
> - Zou, S., Xu, T., and Liang, Y., 2019. Finite-sample analysis for sarsa with linear function approximation. In *Proc. Advances in Neural Information Processing Systems (NeurIPS)*.
> - Wang, H., Mitra, A., Hassani, H., Pappas, G.J. and Anderson, J., 2023. Federated temporal difference learning with linear function approximation under environmental heterogeneity. arXiv preprint arXiv:2302.02212.

---

> > ### Comment · Reviewer_rA2b · 2023-11-22
> > **Thanks for the responses.**
> >
> > Thanks for the responses. They address my comments. I will keep my score.

---

### Author Response · Authors · 2023-11-21
**General Response**

We would like to thank the reviewers for their valuable and insightful comments on our work.
We are happy with the overall positive feedback regarding clarity of presentation, novel contribution, and intuitive explanations of the attained speedup versus heterogeneity trade-off.

In what follows, we summarize our responses to each of the main comments. We also point out the changes we have made to our paper in response to these comments.

- **Reviewer rA2b**: We have explained the source of the linear speedup. Furthermore, in the revised paper, we have now added a new section, Appendix B (FINITE-TIME RESULTS COMPARISON), to compare our finite-time results with other related work, demonstrating that our finite-time error bound is consistent with existing studies.

- **Reviewer qewk**: In the revised paper, we have now clearly shown how the synchronization period $K$ affects the higher-order term in our finite-time bound for Corollary 2.2. We had previously omitted showing this exact dependence since it showed up as a higher-order effect. We have also added a Conclusion section in our revised paper.

    In our response, we have elaborated on the *myriad of technical challenges pertaining to the interplay between function approximation, temporally correlated Markovian samples, local steps, and time-varying Markov chains across agents*, that makes it particularly challenging to analyze our proposed algorithm `FedSARSA`. In this context, our work is the first to analyze the finite-time effects of combining information from heterogeneous time-varying Markov chains - a challenge that **does not show up in the single-agent case**. The fact that a linear speedup is even possible for the scenario we study is non-obvious a priori - nonetheless, our work overcomes the aforementioned challenges and provides strong theoretical guarantees, highlighting both the advantage and price of federation.

- **Reviewer pmTu**: In the revised paper, we have now re-plotted all the graphs with 95\% confidence region. In our response, we have also explained what the term MSE means in our plots, and also how the linear speedup effect causes the plots to look different for different number of agents.

We hope our responses and changes above address all the main comments that were raised. We would be happy to engage in further discussions if some queries remain.

---

### Meta-Review · Area_Chair_6ZpL · 2023-12-07

**Metareview:**

This paper considers a form of federated SARSA and shows linear speedup under appropriate assumptions in terms of the number of nodes. This is a new result for policy iteration methods of this type and I agree with the majority opinion of the reviewers that it should be accepted.

**Justification For Why Not Higher Score:**

While it is a clear contribution to the area, I am not sure the result yet has significant practical implications, and I would rather reserve spotlight/oral either for breakthrough theory or for results that have both theoretical and practical significance.

**Justification For Why Not Lower Score:**

Paper makes a clear contribution to RL theory.

---

### Decision · Program_Chairs · 2024-01-16

Accept (poster)